# Rethinking Transformer Inputs for Time-Series via Neural Temporal Embedding

## Abstract

Transformer-based models, originally introduced in the field of natural language processing (NLP), have recently demonstrated strong performance in time-series forecasting. Due to the order-agnostic nature of the attention mechanism, these models have relied on positional encoding (PE) to capture temporal information. However, recent studies have reported that simple linear models can outperform complex Transformer architectures, and other works have also shown that modifying the Transformer input design can improve performance. Motivated by this issue, we propose Neural Temporal Embedding (NTE), an embedding mechanism that effectively internalizes temporal dependencies without relying on either value embedding or positional encoding. NTE leverages simple neural modules such as Conv1D and LSTM to independently process each variable's time-series and directly learn temporal patterns. As a result, it removes the need for linear projection for value embedding and positional encoding in the input stage, thereby enabling the model to simultaneously achieve architectural flexibility and competitive performance. Experimental results on standard benchmarks including ETT, ECL, and Weather show that the proposed NTE-based models match or outperform state-of-the-art Transformer variants, particularly maintaining stable accuracy in long-horizon forecasting. These empirical findings show that Transformer-based models for time-series forecasting can achieve performance improvements through simple input enhancements without complex architectural modifications, thereby suggesting new possibilities for simpler and more generalizable input architectures.

## 1 Introduction

The Transformer, first introduced in the field of Natural Language Processing (NLP), has shown remarkably effectiveness in modeling long-term dependencies through its attention mechanismVaswani et al. (2017). However, since the attention operation itself is inherently permutation-invariant, additional positional information must be injected to encode sequence order. To address this, the positional encoding (PE) has become an indispensable component of Transformer models. A standard approach is to employ fixed sinusoidal functions, where sine and cosine waves at multiple frequencies are added to the embeddings, thereby encoding relative and absolute positions into a high-dimensional representation space.

These advancements have rapidly expanded into the field of time-series forecasting, leading to the development of numerous Trasnformer-based models. Such modes typically treat each time step as an individual token and adopt input pipelines that apply absolute PE borrowed from NLP. In contrast, some recent implementations of time-series Transformers either weaken or completely remove explicit PE. For instance, FEDformer leverages temporal embeddings in its frequency-domain decomposition-reconstruction blocks to achieve superior performance Zhou et al. (2022), while iTransformer attains competitive results without explicit PE by adopting feature-wise tokenizationLiu et al. (2024). Furthermore, several recent studies have explored alternative approaches, introducing learnable encodings such as Time2Vec or hybrid variants of PE to further enhance performanceKazemi et al. (2019).

On the other hand, recent studies have raised critical questions regarding the limitations of Transformer based approaches. Surprisingly, simple linear models such as DLinear and RLinear have

been shown to outperform many Trasnformer variants, prompting a reconsideration of whether complex attention mechanisms-and their underlying design choices, including value embeddings and the injection of positional or temporal information-are always the most effective solutionZeng et al. (2023); Li et al. (2023). In particular, time-series data are inherently governed by temporal order and exhibit pronounced characteristics such as irregular interval, seasonality, and structural shifts. This raises the question of whether directly transplanting input pipelines originally designed for NLP is always appropriate. Consequently, there is a growing need for input pipeline designs that better capture the unique complexities of time-series data.

This paper goes beyond merely asking "Is positional information necessary?" and instead aims to reinterpret the entire input stack of Transformers. Traditionally, the input stack of time-series Transformers has been organized as Value Embedding (Linear/Conv1D, hereafter LE) $\rightarrow$ positional or temporal information injection (PE or temporal features). We propose Neural Temporal Embedding (NTE), which unifies this stack into a single learnable temporal layer. NTE is simple neural layers (e.g., Dense, Conv1D, LSTM) that directly encodes the temporal axis of each variable, operates regardless of the presence of LE, and is designed to learn order implicitly without requiring explicit PE. The subsequent Transformer stack remains unchanged.

Traditionally, the input stack of time-series Transformers has been organized as Value Embedding (Linear/Conv1D, hereafter LE) $\rightarrow$ positional or temporal information injection (PE or temporal features). We propose NTE, which unifies this stack into a single learnable temporal layer. NTE is simple neural layers (e.g., Dense, Conv1D, LSTM) that directly encodes the temporal axis of each variable, operates regardless of the presence of LE, and is designed to learn order implicitly without requiring explicit PE. The subsequent Transformer stack remains unchanged.

From this perspective, we systematically evaluate the effect of reinterpreting the input layer across (1) univariate and multivariate settings, and (2) three backbones (Vanilla TransformerVaswani et al. (2017), PatchTSTNie et al. (2022), and iTransformerLiu et al. (2024)). We conduct experiments on nine standard benchmarks (ETTh1, ETTh2, ETTm1, ETTm2, ECL, Exchange, Solar-Energy, Traffic, and Weather). In the main text, we report the averaged results over forecasting horizons (96, 192, 336, 720), while detailed results for each horizon are provided in the Appendix. Moreover, for iTransformer, we include ablation studies that toggle LE and PE, comparisons among NTE variants, and evaluations against diverse PE alternatives. To further interpret why NTE works, we complement our analysis with representation similarity (CKA), token-wise concentration (entropy), and qualitative prediction visualizations.

Our contributions can be summarized as follows:

1. **Input-stack unification:** We replace the conventional value embedding + positional/time encoding input stack with a single learnable NTE layer, which directly encodes per-variable temporal structure and leaves the downstream Transformer stack unchanged.

2. **Simplified input design:** NTE is a shallow neural layer (Dense/Conv1D/LSTM) that learns temporal order without explicit PE, reducing input-design and tuning choices while remaining plug-and-play across tokenization schemes.

3. **Effective under feature-tokenization:** We evaluate on three backbones (Vanilla Transformer, PatchTST, iTransformer) over ETTh1, ECL, and Exchange, reporting horizon-mean results for 96/192/336/512 steps, and then extend to all nine standard datasets with iTransformer $\pm$ NTE. The main observation is that NTE consistently improves performance when the input is inverted into variable-wise tokens (iTransformer-style permute), whereas gains are limited under time-step tokenization.

This paper aims to redefine the input paradigm of Transformers for time-series forecasting and demonstrate that competitive temporal representations can be learned with simple neural layers, even without value embeddings or explicit PE.

## 2 RELATED WORK

### 2.1 ARCHITECTURAL ADVANCES IN TRANSFORMER-BASED TIME-SERIES FORECASTING

Recently, many Transformer-based models have been proposed for long-term sequence forecasting (LTSF), each with distinct mechanisms.

Informer introduces ProbSparse self-attention, reducing complexity to $O(L \log L)$ and processing long sequences efficiently via attention distillationZhou et al. (2021). PatchTST converts time-series into patch-level tokens, reducing input length and applying channel-wise self-attention. It typically uses 1D linear or convolutional projections with positional embeddings, and has shown strong forecasting performanceNie et al. (2022). We adopt PatchTST as a backbone and evaluate replacing its input module with NTE. GAformer partitions variables and temporal dimensions into groups and applies independent attention mechanisms to each group, enabling effective learning of latent temporal structures within multivariate time-seriesXiao et al. (2024). TimXer redefines tokenization and position encoding to build a lightweight Transformer with strong forecasting accuracyWang et al. (2024b).

These Transformer variants have generally maintained fixed or learnable PE. However, criticisms have been raised regarding the structural complexity of Transformer models. LTSF-Linear and DLinear have reported results that match or even surpass Transformer-based approaches in various settings using only simple linear transformations and decomposition-based designs, prompting a reconsideration of whether complex attention structures—and their underlying assumptions of value embedding and positional/temporal information injection—are always optimal. Taken together, recent developments suggest that, in addition to architectural advances, a reevaluation of input design itself is required.

### 2.2 TRENDS IN POSITIONAL ENCODING

The question of whether the fixed PE in Transformer models is truly optimal has been raised in recent studies Irani & Metsis (2025). To address this, various learnable or hybrid forms of PE have been proposed in recent years Kazemi et al. (2019); Li et al. (2021); Ke et al. (2021); Devlin et al. (2019).

One representative approach is Time2Vec, which replaces fixed sinusoidal functions with parameterized vectors that incorporate learned periodicity and linear trends Kazemi et al. (2019). This method allows models to directly learn diverse periodic patterns from data, thereby contributing to improved forecasting performance. Learnable PE assigns a trainable embedding vector to each position, optimizing these parameters during training. This structure enables the dynamic adjustment of positional and semantic information by granting each position in the input sequence a unique, data-driven representation Li et al. (2021). Hybrid PE combines absolute and relative positional information for example, by adding learned relative position biases inside the self-attention computation while still using absolute embeddings for content to leverage the strengths of both schemes Ke et al. (2021). Such methods can provide better generalization to longer sequences and greater flexibility in capturing sequence order. More recently, alternative approaches such as RoPE, which encodes positional information through rotary transformations Su et al. (2021), and ALiBi, which applies linear biases based on input length Press et al. (2022), have also been proposed.

Although these enhanced PE methods contribute to handling long sequences and improving structural flexibility, most Transformer-based time-series models still take the explicit addition of positional or temporal information as a fundamental premise.

### 2.3 REMOVAL OF POSITIONAL ENCODING IN TIME-SERIES FORECASTING

Recently, alongside growing concerns regarding the appropriateness of PE, a few studies have explored training Transformer models without relying on PE Liu et al. (2024); Wu et al. (2021); Zhou et al. (2022); Anh et al. (2024); Li & Anastasiu (2025).

A representative example is iTransformer, which introduced a feature-wise tokenization approach by using variables as tokens, instead of the tokenization using time steps as the basis employed in existing time-series Transformer models Liu et al. (2024). This design enables self-attention across

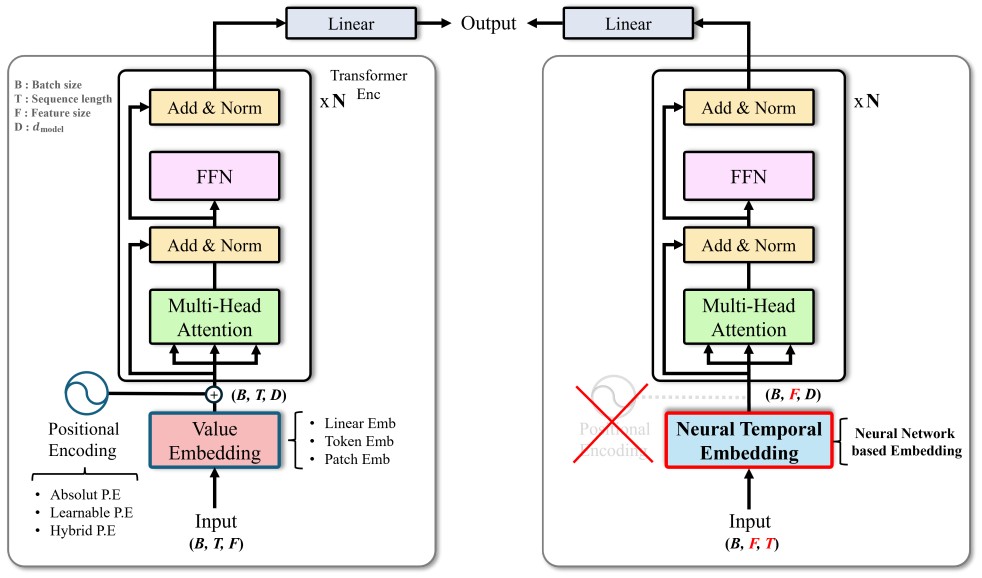

(a) The standard Transformer-based architecture    (b) The proposed Neural Temporal Embedding architecture

Figure 1: **Left**: the standard Transformer-based architecture applies value embedding and explicit PE. **Right**: our proposed architecture replaces these components with NTE, a neural network-based module.

variables without PE and achieves competitive performance on several datasets. Autoformer decomposes series into trend and seasonality via an Auto-Correlation mechanism Wu et al. (2021). FEDformer operates in the frequency domain, capturing global patterns while discarding redundant components to improve efficiency and accuracy Zhou et al. (2022). Additionally, LSPatch-T pretrains a model by utilizing patch-based Transformers and subsequently transfers it to downstream tasks for time-series forecasting. In this model, each variable is treated as a single token, and each token contains the entire time-series corresponding to that variable. As a result, PE is not employed during the prediction phase, since the temporal order information is inherently contained within each token, eliminating the need for separate PE Anh et al. (2024). Furthermore, PFformer builds a position-free encoder–decoder by introducing two learnable embeddings: an Enhanced Feature-based Embedding for the encoder and an AutoEncoder-based Embedding for the decoder Li & Anastasiu (2025).

This trend underscores the question: "Can temporal representations be learned without explicit positional injection?" Building on the feature-wise tokenization paradigm of iTransformer, this paper proposes Neural NTE, which replaces conventional value embedding and positional encoding with simple neural layers such as Conv1D, LSTM, or Dense to implicitly capture temporal structure. By unifying and substituting the input stack with a single learnable temporal layer, NTE demonstrates that competitive forecasting performance can be achieved across diverse backbones without relying on either value embedding or positional encoding.

## 3 METHODOLOGY

### 3.1 OVERALL METHODOLOGY

Figure 1 compares the standard Transformer-based framework for time-series forecasting with the overall architecture of the proposed model. (a) illustrates a conventional encoder-only structure that is commonly used. In this framework, the input time-series data $\mathbf{X} \in \mathbb{R}^{B \times T \times F}$ undergoes a value embedding process, followed by the application of PE. The value embedding typically uses one of various strategies, such as linear projection, token embedding, or patch-based embedding. To reflect sequence order information, PE is added to the embedded sequence, commonly implemented

using absolute, learnable, or hybrid PE methods. The representation constructed in this manner is then processed through the Transformer encoder block, and the final output is generated via a linear projection layer.

Unlike (a), (b) illustrates the overall architecture proposed in this study. This architecture retains the standard Transformer encoder block while exhibiting a fundamental difference in the embedding approach. Unlike conventional approaches that jointly use value embedding and PE, our method completely removes explicit positional information and replaces it with an NTE module.

time-series inputs $X \in \mathbb{R}^{B \times T \times F}$ are inherently ordered, so neural modules can capture sequential structure without explicit PE. A standalone Transformer encoder, however, is permutation-equivariant over tokens and thus order-agnostic by design. By contrast, fully connected layers on lag-ordered windows, recurrent units (e.g., LSTM), and 1D convolutional families are intrinsically order-sensitive—tying parameters to sequential states. Consequently, an NTE front-end built from these operators can internalize temporal structure without any explicit positional mechanism. For example, a 1D convolution

$$(h * x)(t) = \sum_{\tau=0}^{k-1} w(\tau)\, x(t-\tau) \tag{1}$$

satisfies shift-equivariance, i.e., $(h*x)(t+\Delta) = \sum_{\tau} w(\tau)\, x(t+\Delta-\tau)$, thereby preserving temporal order. Likewise, the LSTM recurrence

$$h_t = f(x_t, h_{t-1}; \theta) \tag{2}$$

accumulates all past inputs, embedding order into hidden states. Thus, NTE captures *intra-variable temporal dynamics*, while the Transformer encoder models *inter-variable dependencies*. In contrast, PE corresponds to a fixed expansion $Z_t = X_t W + p_t$, whereas NTE learns an adaptive nonlinear mapping $Z_t = f_\theta(X_{1:t})$, thereby avoiding the need for explicit positional signals.

## 3.2 NTE

Figure 2 illustrates the structure of the proposed NTE, and Algorithm 1 summarizes its overall integration within the iTransformer framework. Given an input tensor $X \in \mathbb{R}^{B \times T \times F}$, where $B$ is

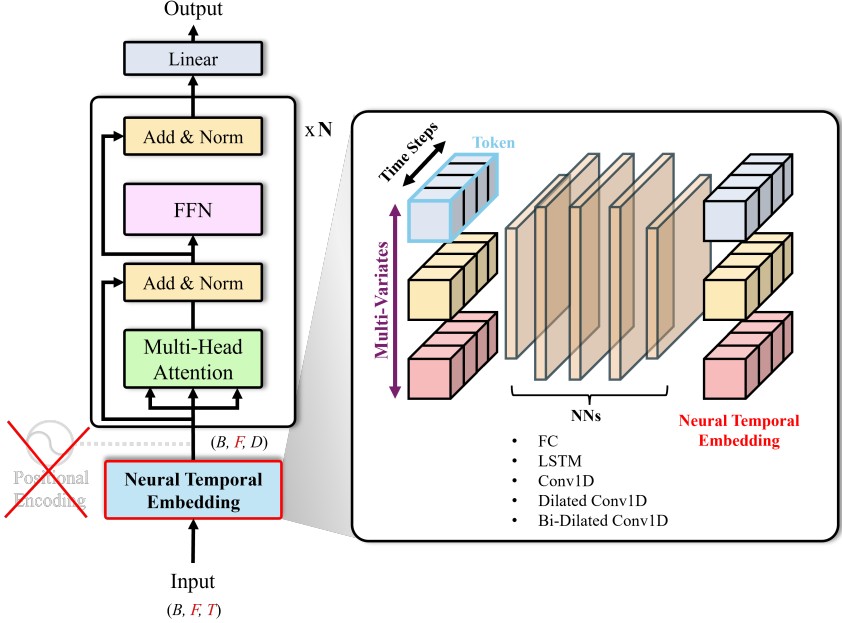

Figure 2: Architecture of NTE. The right panel illustrates the internal structure of NTE, where multivariate tokenized representations are processed across time steps using neural network such as FC layers, LSTM, or Conv1D.

the batch size, $T$ the sequence length, and $F$ the number of variables, the tensor is first transposed to $\mathbb{R}^{B \times F \times T}$ to enable feature-wise tokenization. This step allows each variable to be treated as a distinct token while preserving its temporal dynamics.

Unlike conventional input layers that combine value embedding and PE, our approach directly employs NTE modules to process each variable's temporal sequence. Depending on the desired temporal bias, NTE can be instantiated with different neural modules: Conv1D emphasizes local and multi-scale receptive fields, LSTM accumulates long-range sequential dependencies, and fully connected layers provide a lightweight alternative. Through these modules, NTE implicitly learns temporal dependencies without explicit positional signals. As highlighted in Section 3.1, NTE is primarily responsible for capturing *intra-variable temporal patterns*, while the Transformer encoder models *inter-variable dependencies* via self-attention.

The resulting embeddings are then passed through the Transformer encoder block and projected to match the forecast horizon. Finally, the representation is transposed back to the time-major format to generate the prediction sequence. This process is summarized in Algorithm 1 and visually depicted in Figure 2.

## 4 EXPERIMENTS

**Datasets.** In this study, we utilized publicly available benchmark datasets that are widely used in the field of time-series forecasting, including ETT (Electricity Transformer Temperature; consisting of 4 subsets) Zhou et al. (2021), ECL (Electricity Consumption Load) Li et al. (2019), Weather Zhou et al. (2021), Traffic Wu et al. (2022), and Solar-Energy datasets Lai et al. (2018). These datasets encompass diverse domain-specific time-series characteristics and have been extensively adopted as standard evaluation benchmarks in recent studies. Exchangefor financial time-series forecasting Zhou et al. (2021)and the M4 dataset for short-term forecasting across multiple domainsMakridakis et al. (2018).

**Baselines.** For comparison, we selected forecasting models that have demonstrated superior performance in time-series forecasting tasks. In particular, models such as iTransformer, which recently achieved SOTA performance, were included : iTransformerLiu et al. (2024), RLinearLi et al. (2023), TimesNetWu et al. (2022), FEDformerZhou et al. (2022), PatchTSTNie et al. (2022), DLinearZeng et al. (2023), AutoformerWu et al. (2021), Ada-MSHyperShang et al. (2024), Timemixer++Wang et al. (2024a), S-MambaWang et al. (2025), SDE-MambaWeng et al. (2025), SAMformerIlbert et al. (2024). A comprehensive performance comparison with baseline models is presented in the Appendix.

**Positional encoding baselines.** To evaluate PE and NTE, we compare three backbone models (Vanilla Transformer, PatchTST, and iTransformer) with four representative PE variants: (1) **Sinu-**

---

**Algorithm 1** NTE-based iTransformer architecture for time-series forecasting

---

**Require:** Input sequence $\mathbf{X} \in \mathbb{R}^{B \times T \times F}$, forecast horizon $S$, embedding dimension $D$
1: **Transpose** time and variate axes: $\mathbf{X} \leftarrow \mathbf{X}^{\top}$
2: $\triangleright \mathbf{X} \in \mathbb{R}^{B \times F \times T}$
3: **Embed** variates via NTE: $\mathbf{H} \leftarrow \text{NTE}(\mathbf{X})$
4: $\triangleright \mathbf{H} \in \mathbb{R}^{B \times F \times D}$
5: **Apply** Dropout for regularization: $\mathbf{H} \leftarrow \text{Dropout}(\mathbf{H})$
6: $\triangleright \mathbf{H} \in \mathbb{R}^{B \times F \times D}$
7: **Encode** with Transformer block: $\mathbf{H} \leftarrow \text{TransformerEncoder}(\mathbf{H})$
8: $\triangleright$ Self-Attention + Feed-Forward
9: **Project** embeddings to forecast length: $\mathbf{H} \leftarrow \text{Linear}(\mathbf{H})$
10: $\triangleright \mathbf{H} \in \mathbb{R}^{B \times F \times S}$
11: **Transpose** back to time axis: $\mathbf{H} \leftarrow \mathbf{H}^{\top}$
12: $\triangleright \mathbf{H} \in \mathbb{R}^{B \times S \times F}$
13: **return** Final prediction $\hat{\mathbf{Y}} = \mathbf{H}$
14: $\triangleright \hat{\mathbf{Y}} \in \mathbb{R}^{B \times S \times F}$

---

soidal PE Vaswani et al. (2017), the fixed encoding used in the original Transformer; (2) **Learnable PE**, where positional vectors are trainable parameters; (3) **TUPE** Ke et al. (2020), which unties token and position interactions in attention; and (4) **ConvSPE**, which augments sinusoidal encodings with a convolutional layer.

**Implementation.** All experiments were implemented using the PyTorch framework and conducted on a single NVIDIA RTX 4090 (24GB) GPU environment. The baseline models were reproduced based on their official implementations or publicly available benchmark code.

**Evaluation metric.** Model performance was evaluated using Mean Squared Error (MSE) and Mean Absolute Error (MAE). We additionally measured efficiency (FLOPs, per-sample inference time, end-to-end latency, and peak GPU memory), with detailed results provided in the Appendix.

**Bi-directional Dilated Convolutional Embedding.** In this study, to reflect the bidirectionality of temporal information, we employed a bidirectional Dilated Conv1D structure as the NTE module, as illustrated in Figure 3. This structure consists of Past- and Future-Dilated Conv1D blocks, which independently process past and future information. Similar to the BTCN in BTCSAN modules, the model leverages residual connections and a dilation factor to effectively capture multi-scale temporal patterns Sun et al. (2019). Subsequently, the bidirectional embeddings are integrated through a Feature-Selection Attention module, which dynamically selects the more important information at each time step. The Concat method concatenates the two embeddings, applies soft attention over the entire vector, and generates the final representation through a projection layer. In contrast, the Combine method merges past and future embeddings and applies soft attention to selectively emphasize the more informative direction at each time step.

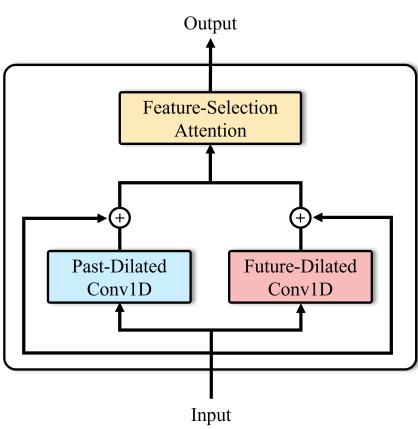

Figure 3: Bi-directional Dilated Conv1D architecture.

## 4.1 MAIN RESULTS

Table 1: MSE comparison of multivariate LTSF backbones

| Model | | Transformer | Transformer +NTE | iTransformer | iTransformer +NTE | PatchTST | PatchTST +NTE |
|---|---|---|---|---|---|---|---|
| ETTh1 | 96 | 0.835 | $0.840_{\Delta \uparrow 0.6\%}$ | 0.386 | $0.380_{\Delta \downarrow 1.6\%}$ | 0.414 | $0.378_{\Delta \downarrow 8.7\%}$ |
| | 192 | 0.933 | $1.082_{\Delta \uparrow 16.0\%}$ | 0.441 | $0.429_{\Delta \downarrow 2.7\%}$ | 0.460 | $0.439_{\Delta \downarrow 4.6\%}$ |
| | 336 | 1.098 | $1.419_{\Delta \uparrow 29.2\%}$ | 0.487 | $0.475_{\Delta \downarrow 2.5\%}$ | 0.501 | $0.476_{\Delta \downarrow 5.0\%}$ |
| | 720 | 1.138 | $1.164_{\Delta \uparrow 2.3\%}$ | 0.503 | $0.484_{\Delta \downarrow 3.8\%}$ | 0.500 | $0.502_{\Delta \uparrow 0.4\%}$ |
| | *Avg* | 1.001 | $1.126_{\Delta \uparrow 12.5\%}$ | 0.454 | $0.442_{\Delta \downarrow 2.6\%}$ | 0.469 | $0.449_{\Delta \downarrow 4.3\%}$ |
| ECL | 96 | 0.298 | $0.480_{\Delta \uparrow 61.1\%}$ | 0.148 | $0.140_{\Delta \downarrow 5.4\%}$ | 0.181 | $0.176_{\Delta \downarrow 2.8\%}$ |
| | 192 | 0.355 | $0.574_{\Delta \uparrow 61.7\%}$ | 0.162 | $0.157_{\Delta \downarrow 3.1\%}$ | 0.188 | $0.184_{\Delta \downarrow 2.1\%}$ |
| | 336 | 0.484 | $0.946_{\Delta \uparrow 95.5\%}$ | 0.178 | $0.170_{\Delta \downarrow 4.5\%}$ | 0.204 | $0.201_{\Delta \downarrow 1.5\%}$ |
| | 720 | 0.414 | $1.032_{\Delta \uparrow 149.0\%}$ | 0.225 | $0.201_{\Delta \downarrow 10.7\%}$ | 0.246 | $0.242_{\Delta \downarrow 1.6\%}$ |
| | *Avg* | 0.388 | $0.758_{\Delta \uparrow 95.4\%}$ | 0.178 | $0.167_{\Delta \downarrow 6.2\%}$ | 0.205 | $0.201_{\Delta \downarrow 2.0\%}$ |
| Exchange | 96 | 0.673 | $1.160_{\Delta \uparrow 72.3\%}$ | 0.086 | $0.087_{\Delta \uparrow 1.2\%}$ | 0.088 | $0.088_{\Delta \approx 0.0\%}$ |
| | 192 | 1.223 | $1.118_{\Delta \downarrow 8.6\%}$ | 0.177 | $0.178_{\Delta \uparrow 0.6\%}$ | 0.176 | $0.188_{\Delta \uparrow 6.8\%}$ |
| | 336 | 1.745 | $1.629_{\Delta \downarrow 6.6\%}$ | 0.331 | $0.336_{\Delta \uparrow 1.5\%}$ | 0.301 | $0.338_{\Delta \uparrow 12.3\%}$ |
| | 720 | 2.369 | $1.450_{\Delta \downarrow 38.8\%}$ | 0.847 | $0.824_{\Delta \downarrow 2.7\%}$ | 0.901 | $0.886_{\Delta \downarrow 1.7\%}$ |
| | *Avg* | 1.503 | $1.339_{\Delta \downarrow 10.9\%}$ | 0.360 | $0.356_{\Delta \downarrow 1.1\%}$ | 0.367 | $0.375_{\Delta \uparrow 2.2\%}$ |

Table 1 presents a comparison of forecasting accuracy (measured by MSE) among three representative backbones (Transformer, iTransformer, and PatchTST) and their NTE-augmented variants.

When applied to the vanilla Transformer, NTE does not yield improvements and even degrades performance on ETTh1 and ECL, indicating that the sequential bias introduced by NTE is insufficient to compensate for the order-agnostic nature of the standard Transformer. In contrast, combining NTE with architectures that already incorporate feature-wise or patch-level tokenization, such as iTransformer and PatchTST, consistently maintains or enhances accuracy across datasets. For instance, iTransformer+NTE reduces MSE by 2.6% on ETTh1 and 6.2% on ECL compared to the baseline iTransformer, while PatchTST+NTE achieves 4.3% and 2.0% lower MSE on ETTh1 and ECL, respectively. The results for MAE on the three representative backbones are provided in Appendix B.1.

## 4.2 ABLATION STUDIES

Table 2: PE variants vs NTE variants on iTransformer

| Setting | ETTh1 | | ECL | | Exchange | |
|---|---|---|---|---|---|---|
| Metric | MSE | MAE | MSE | MAE | MSE | MAE |
| LE+PE | 0.446 | 0.442 | 0.170 | 0.265 | 0.363 | 0.407 |
| LE Only | 0.454 | 0.447 | 0.178 | 0.270 | 0.360 | **0.403** |
| PE Only | 0.721 | 0.582 | 0.862 | 0.765 | 0.432 | 0.456 |
| Sinusoidal PE | 0.446 | 0.442 | 0.170 | 0.265 | 0.363 | 0.407 |
| Learnable PE | 0.443 | **0.440** | **0.161** | **0.257** | 0.363 | 0.406 |
| TUPE | 0.444 | 0.441 | 0.174 | 0.265 | 0.362 | 0.405 |
| ConvSPE | 0.448 | 0.443 | 0.168 | 0.263 | 0.365 | 0.407 |
| NTE-FC | 0.455 | 0.447 | 0.175 | 0.268 | **0.356** | 0.404 |
| NTE-LSTM | 0.452 | 0.445 | 0.167 | 0.262 | 0.361 | 0.410 |
| NTE-Conv1D | 0.467 | 0.455 | 0.173 | 0.268 | 0.366 | 0.411 |
| NTE-BiDConv1D | **0.442** | 0.441 | 0.177 | 0.269 | 0.367 | 0.409 |

Table 2 presents the ablation results of iTransformer under different input configurations. When only positional encoding is used (PE Only), performance drops substantially across datasets, confirming the necessity of value embeddings. By contrast, combining value embeddings with various PE schemes yields generally stable results, with Learnable PE achieving the best performance on ETTh1 and ECL. Importantly, NTE-based front-ends also deliver competitive accuracy: NTE-BiDConv1D achieves the lowest error on ETTh1, NTE-LSTM performs best on ECL, and NTE-FC achieves the best results on Exchange. These findings indicate that iTransformer does not heavily rely on explicit PE, and that simple neural modules such as NTE can serve as effective alternatives for capturing temporal dependencies.

## 5 LIMITATIONS

Although the proposed NTE framework demonstrates strong empirical performance, one limitation remains. All evaluations were conducted exclusively on time-series forecasting tasks, and thus the generalizability of NTE to other domains, such as vision or NLP, remains an open question for future work.

### 5.1 REPRESENTATION ANALYSIS

Figure 4 (a) heatmap presents the Linear CKA similarity among NTE-based methods and the baseline using PE, while (b) displays the Kernel CKA similarity. Note that configurations prefixed with **L+** indicate that a Linear projection layer was applied before the embedding. In both cases, we observe that most NTE variants including Conv1D, DConv1D, BiDConv1D, and FC exhibit very high mutual similarity (above 0.98 in most linear comparisons), even across versions with **L+** and without linear projection. This indicates that the temporal representations learned by NTE remain consistent and robust regardless of the use of the linear projection layer. Configurations using explicit L+PE show noticeably lower similarity to other NTE variants in both linear and kernel spaces. This divergence suggests that PE-based embeddings construct somewhat different representational spaces compared to NTE. Notably, despite this difference, the NTE variants are able to produce similar or even superior performance, as shown in other experiments.

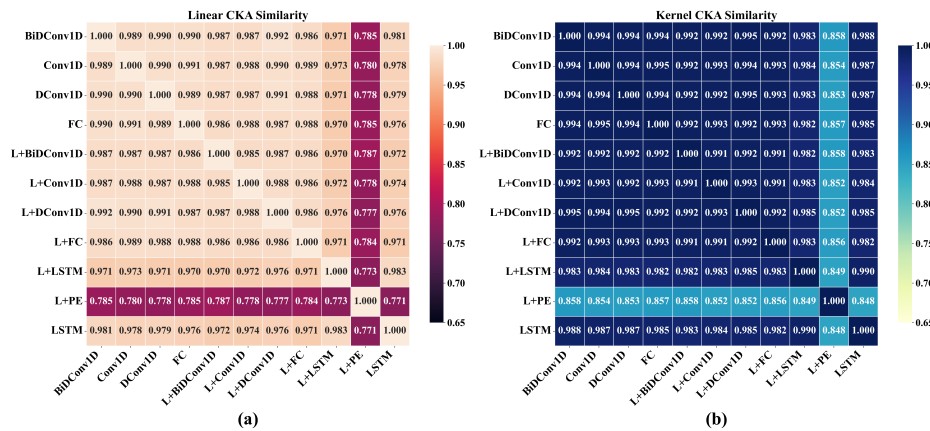

Figure 4: Linear and Kernel CKA similarity heatmaps

These findings support the central claim of this work: temporal dependencies can be effectively internalized through NTE mechanisms without relying on PE, and even without the use of linear projection layers. The fact that structurally diverse NTE modules converge to highly similar representations further suggests that they implicitly capture the necessary temporal structures—highlighting the sufficiency and flexibility of this approach.

## 6 CONCLUSION AND FUTURE WORK

Existing Transformer-based time-series forecasters largely inherited PE from NLP. Yet time-series intrinsically possess a fixed temporal order whose role can differ from token positions in language. Motivated by this gap, we revisited the input interface of time-series Transformers and proposed a NTE layer that replaces the conventional value embedding + positional/time encoding stack. NTE is a shallow neural layer (e.g., Dense/Conv1D/LSTM) that directly encodes per-variable temporal structure, requires no explicit PE, and leaves the downstream Transformer stack unchanged while remaining configurable and plug-and-play.

During our study we also observed that the commonly used linear projection for value embedding is not strictly necessary in all settings. More importantly, consistent gains with NTE appeared when the input is inverted into variable-wise tokens (i.e., a feature-token pipeline); under time-step tokenization, improvements were limited. This offers a structural insight: simplifying and redesigning the input pipeline around a learnable temporal layer can be especially effective when variables—not time steps—form the tokens.

Empirically, on three representative datasets (ETTh1, ECL, Exchange) across three backbones (a vanilla Transformer, a patch-based backbone, and a variable-token backbone), NTE achieves competitive or improved horizon-mean performance (96/192/336/720). We further extend the evaluation to the full set of nine standard datasets in the variable-token setting and observe similar trends. Collectively, these results indicate that explicit positional information is not a prerequisite for strong forecasting performance, particularly when the input is organized as variable-wise tokens, and that the input stack can be streamlined without sacrificing accuracy.

For future work, we plan to broaden backbone coverage (including PE-free/channel-attention and state-space models), study irregular sampling and ultra-long horizons, and explore hybrid designs (e.g., light NTE with optional PE/gating) alongside clearer selection heuristics for NTE variants. We will also provide more systematic efficiency measurements and consider automated design search to tailor NTE to dataset-specific temporal characteristics.

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

# A DETAILED COMPONENTS

## A.1 IMPLEMENTATION DETAILS

All experiments are implemented in PyTorch and conducted on a single NVIDIA RTX 4090 GPU (24GB VRAM) equipped with an Intel Core i9-13900K CPU and 128GB RAM. The software environment is managed with `Anaconda`. For the proposed Neural Temporal Encoding (NTE), we implement five architectural variants: **FC**, **LSTM**, **Conv1D**, **Dilated Conv1D**, and **Bi-Dilated Conv1D**. The convolutional variants use kernel sizes $\{3, 5\}$, and dilations follow the schedules in Table 3. We further vary the repeat count in $\{1, 2, 3\}$ with input length $T = 96$, and evaluate forecasting horizons $S \in \{96, 192, 336, 720\}$. For reproducibility, full implementation details including model settings and training hyperparameters are publicly available in our open-source codebase. All baseline models are re-implemented using the open-source iTransformer repository, following each model's original settings and data configurations. The pseudocode for our NTE-based iTransformer is presented in Algorithm.

Table 3: Dilation schedules for Dilated and Bi-Dilated Conv1D across different forecasting horizons.

| Task | Horizon | NTE Dilated Conv1D Bi-Dilated Conv1D |
|---|---|---|
| Long-term forecasting | 96 | $1, 2, 4, 8$ |
| | 192 | $1, 2, 4, 8, 16$ |
| | 336 | $1, 2, 4, 8, 16, 32$ |
| | 720 | $1, 2, 4, 8, 16, 32, 64$ |
| Short-term forecasting (M4) | 96 | $1, 2, 4$ |
| | 192 | $1, 2$ |
| | 336 | $1, 2$ |
| | 720 | $1, 2$ |

In the Exchange dataset, dilation values of $1, 2, 4$ are used only for horizon 336, while $1, 2, 4, 8$ are used for the other horizons.

## A.2 DATASETS DETAILED

We evaluate the proposed method on publicly available multivariate time-series datatsets that span energy, meteorology, transportation, and finance domains in time-series forecasting research. In the experiments, we adopt same data pre-processing steps and train-validation-test splits as used in iTransformer, maintaining a strict chronological partition to avoid data leakage. The input sequence length is fixed to 96 for all datasets, and the prediction lengths vary among $\{96, 192, 336, 720\}$. A summary of the dataset characteristics is described in Table 4.

**ETT (Electricity Transformer Temperature)** consists of four subsets: ETTh1 and ETTh2 with hourly sampling, and ETTm1 and ETTm2 with 15-minute sampling, each containing seven variables recorded between July 2016 and July 2018.

**ECL (Electricity Consumption Load)** contains hourly electricity usage data from 321 clients over multiple years and is a standard benchmark for demand forecasting.

**Weather** provides 21 meteorological variables collected every 10 minutes from the Max Planck Institute's Weather Station in 2020.

**Traffic** reports hourly occupancy rates measured by 862 road sensors in the San Francisco Bay Area during 2015-2016.

**Solar-Energy** captures solar power production at 10-minute intervals from 137 photovoltaic plants.

**Exchange** consists of daily exchange rates of eight countries against the US dollar from 1990 to 2016 and is commonly used to evaluate long-horizon financial forecasting.

**M4** contains 100,000 time-series with mixed sampling frequencies, including yearly, quarterly, monthly, weekly, daily, and hourly. Forecast horizons range from 6 to 48 steps, depending on frequency, making it a standard benchmark for testing robustness and generalization.

Table 4: Benchmark datasets for multivariate time-series forecasting. Sizes are listed as (train, validation, test).

| Task | Dataset | Sampling Frequency | Dim | Size | Prediction Length |
|------|---------|--------------------|-----|------|-------------------|
| Long-term forecasting | ETTh1, ETTh2 | 1 hour | 7 | (8545, 2881, 2881) | 96, 192, 336, 720 |
| | ETTm1, ETTm2 | 15 minutes | 7 | (34465, 11521, 11521) | 96, 192, 336, 720 |
| | Exchange | Daily | 8 | (5120, 665, 1422) | 96, 192, 336, 720 |
| | Electricity | 1 hour | 321 | (18317, 2633, 5261) | 96, 192, 336, 720 |
| | Weather | 10 minutes | 21 | (36792, 5271, 10540) | 96, 192, 336, 720 |
| | Traffic | 1 hour | 862 | (12185, 1757, 3509) | 96, 192, 336, 720 |
| | Solar-Energy | 10 minutes | 137 | (36601, 5161, 10417) | 96, 192, 336, 720 |
| Short-term forecasting | M4-Yearly | Yearly | 1 | (23000, 0, 23000) | 6 |
| | M4-Quarterly | Quarterly | 1 | (24000, 0, 24000) | 8 |
| | M4-Monthly | Monthly | 1 | (48000, 0, 48000) | 18 |
| | M4-Weakly | Weakly | 1 | (359, 0, 359) | 13 |
| | M4-Daily | Daily | 1 | (4227, 0, 4227) | 14 |
| | M4-Hourly | Hourly | 1 | (414, 0, 414) | 48 |

# B QUANTITATIVE EVALUATION

## B.1 MAIN RESULTS

Table 5: MAE comparison of multivariate LTSF backbones

| Model | | Transformer | Transformer +NTE | iTransformer | iTransformer +NTE | PatchTST | PatchTST +NTE |
|-------|------|-------------|------------------|--------------|-------------------|----------|---------------|
| ETTh1 | 96 | 0.698 | $0.703_{\Delta \uparrow 0.7\%}$ | 0.405 | $0.402_{\Delta \downarrow 0.7\%}$ | 0.419 | $0.401_{\Delta \downarrow 4.3\%}$ |
| | 192 | 0.741 | $0.853_{\Delta \uparrow 15.1\%}$ | 0.436 | $0.431_{\Delta \downarrow 1.1\%}$ | 0.445 | $0.440_{\Delta \downarrow 1.1\%}$ |
| | 336 | 0.799 | $1.014_{\Delta \uparrow 26.9\%}$ | 0.458 | $0.453_{\Delta \downarrow 1.1\%}$ | 0.466 | $0.461_{\Delta \downarrow 1.1\%}$ |
| | 720 | 0.821 | $0.886_{\Delta \uparrow 7.9\%}$ | 0.491 | $0.478_{\Delta \downarrow 2.6\%}$ | 0.488 | $0.495_{\Delta \uparrow 1.4\%}$ |
| | Avg | 0.765 | $0.864_{\Delta \uparrow 12.9\%}$ | 0.447 | $0.441_{\Delta \downarrow 1.3\%}$ | 0.454 | $0.449_{\Delta \downarrow 1.1\%}$ |
| ECL | 96 | 0.403 | $0.515_{\Delta \uparrow 27.8\%}$ | 0.240 | $0.236_{\Delta \downarrow 1.7\%}$ | 0.270 | $0.270_{\Delta \approx 0.0\%}$ |
| | 192 | 0.437 | $0.564_{\Delta \uparrow 29.1\%}$ | 0.253 | $0.250_{\Delta \downarrow 1.2\%}$ | 0.274 | $0.276_{\Delta \uparrow 0.7\%}$ |
| | 336 | 0.512 | $0.779_{\Delta \uparrow 52.1\%}$ | 0.269 | $0.266_{\Delta \downarrow 1.1\%}$ | 0.293 | $0.294_{\Delta \uparrow 0.3\%}$ |
| | 720 | 0.478 | $0.809_{\Delta \uparrow 69.3\%}$ | 0.317 | $0.295_{\Delta \downarrow 6.9\%}$ | 0.324 | $0.326_{\Delta \uparrow 0.6\%}$ |
| | Avg | 0.458 | $0.667_{\Delta \uparrow 45.6\%}$ | 0.270 | $0.262_{\Delta \downarrow 3.0\%}$ | 0.290 | $0.292_{\Delta \uparrow 0.7\%}$ |
| Exchange | 96 | 0.640 | $0.830_{\Delta \uparrow 29.7\%}$ | 0.206 | $0.208_{\Delta \uparrow 1.0\%}$ | 0.205 | $0.205_{\Delta \approx 0.0\%}$ |
| | 192 | 0.842 | $0.818_{\Delta \downarrow 2.9\%}$ | 0.299 | $0.301_{\Delta \uparrow 0.7\%}$ | 0.299 | $0.307_{\Delta \uparrow 2.7\%}$ |
| | 336 | 1.025 | $1.021_{\Delta \downarrow 0.4\%}$ | 0.417 | $0.421_{\Delta \uparrow 1.0\%}$ | 0.397 | $0.422_{\Delta \uparrow 6.3\%}$ |
| | 720 | 1.234 | $0.945_{\Delta \downarrow 23.4\%}$ | 0.691 | $0.685_{\Delta \downarrow 0.9\%}$ | 0.714 | $0.708_{\Delta \downarrow 0.8\%}$ |
| | Avg | 0.935 | $0.904_{\Delta \downarrow 3.3\%}$ | 0.403 | $0.404_{\Delta \uparrow 0.2\%}$ | 0.404 | $0.411_{\Delta \uparrow 1.7\%}$ |

## B.2 SHORT-TERM FORECASTING

Short-term forecasting is assessed with three additional metrics sMAPE (symmetric Mean Absolute Percentage Error), MASE (Mean Absolute Scaled Error), and OWA (Overall Weighted Average). sMAPE is a scale-free percentage error that is less sensitive to unit changes. MASE scales absolute

Table 6: Short-term forecasting on the M4 dataset with iTransformer. We compare the Base architecture against four NTE variants (NTE-FC, NTE-LSTM, NTE-Conv1D, and NTE-BiDConv1D) across frequency groups (Yearly, Quarterly, Monthly, Other) and their overall average. Each entry includes computational cost (FLOPs, Parameters) alongside accuracy results.

| | | | **M4** | | | |
|---|---|---|---|---|---|---|
| Setting | Horizon | FLOPs | Parameters | sMAPE | MASE | OWA |
| Base | Yearly | 2,124,800 | 6,333,970 | 13.731 | 3.101 | 0.81 |
| | Qarterly | 2,109,440 | 6,318,600 | 11.275 | 1.321 | 0.993 |
| | Monthly | 2,117,120 | 6,326,285 | 16.417 | 1.346 | 1.202 |
| | Other | - | - | 5.478 | 3.902 | 1.192 |
| | Average | - | - | 14.018 | 1.871 | 1.006 |
| NTE-FC | Yearly | 2,106,368 | 6,322,182 | 13.572 | 3.05 | 0.799 |
| | Qarterly | 2,109,440 | 6,327,304 | 10.812 | 1.289 | 0.961 |
| | Monthly | 2,124,800 | 6,352,914 | 13.582 | 1.056 | 0.967 |
| | Other | - | - | 5.676 | 4.116 | 1.246 |
| | Average | - | - | 12.52 | 1.724 | 0.912 |
| NTE-LSTM | Yearly | 2,100,224 | 7,392,774 | 17.231 | 3.873 | 1.014 |
| | Qarterly | 2,101,248 | 7,404,040 | 10.663 | 1.272 | 0.948 |
| | Monthly | 2,106,368 | 7,460,370 | 13.63 | 1.061 | 0.971 |
| | Other | - | - | 5.365 | 3.825 | 1.168 |
| | Average | - | - | 13.333 | 1.896 | 0.987 |
| NTE-Conv1D | Yearly | 2,100,224 | 6,327,302 | 13.711 | 3.111 | 0.811 |
| | Qarterly | 2,101,248 | 6,334,472 | 10.609 | 1.266 | 0.943 |
| | Monthly | 2,106,368 | 6,370,322 | 13.563 | 1.055 | 0.966 |
| | Other | - | - | 4.934 | 3.42 | 1.058 |
| | Average | - | - | 12.456 | 1.697 | 0.903 |
| NTE-BiDConv1D | Yearly | 4,209,664 | 8,541,190 | 16.512 | 3.628 | 0.962 |
| | Qarterly | 3,690,496 | 8,010,760 | 13.852 | 1.768 | 1.274 |
| | Monthly | 3,716,096 | 8,169,490 | 16.886 | 1.397 | 1.242 |
| | Other | - | - | 5.499 | 3.949 | 1.201 |
| | Average | - | - | 15.503 | 2.127 | 1.128 |

error by an in-sample seasonal naive baseline, enabling comparison across heterogeneous series. OWA summarizes performance by averaging sMAPE and MASE relative to the same baseline, so values below one indicate overall improvement. Metrics are computed per series and per horizon, then macro-averaged across series and averaged across horizons to yield one score per model. Complete short-term results are reported for iTransformer in Table 6 and for PatchTST in Table 7.

## B.3 COMPARISON WITH BASELINES

Table 8 presents the MSE and MAE results of long-term forecasting experiments for the baselines (iTransformer, RLinear, TimesNet, FEDformer, PatchTST, DLinear, and Autoformer), while Table 9 compares the experimental results of the baselines (Ada-MSHyper, Timemixer++, S-Mamba, SDE-Mamba, and SAMformer).

## B.4 STATISTICAL SIGNIFICANCE ANALYSIS

To evaluate the robustness of our proposed NTE-based model, we conduct each experiment five times using different random seeds and report both the mean and standard deviation of the MSE and MAE scores. As summarized in Table 10, 11, the performance across different datasets and forecasting lengths remains consistently stable, with low standard deviation values observed for all cases. This indicates that the proposed model is not overly sensitive to random initialization and exhibits reliable performance under stochastic training conditions. These findings highlight the robustness of the NTE architecture and its suitability for real-world deployment.

Table 7: Short-term forecasting on the M4 dataset with PatchTST. The Base model and four NTE variants (NTE-FC, NTE-LSTM, NTE-Conv1D, and NTE-BiDConv1D) are evaluated across the same frequency groups (Yearly, Quarterly, Monthly, Other) with an overall average. Reported values include FLOPs, parameter counts, and accuracy outcomes for each setting.

| Setting | Horizon | FLOPs | Parameters | sMAPE | MASE | OWA |
|---|---|---|---|---|---|---|
| | **M4** | | | | | |
| Base | Yearly | 2,108,416 | 6,448,390 | 13.599 | 3.042 | 0.799 |
| | Qarterly | 2,113,536 | 6,453,512 | 11.006 | 1.294 | 0.971 |
| | Monthly | 2,142,208 | 6,482,194 | 14.07 | 1.12 | 1.014 |
| | Other | - | - | 5.953 | 3.817 | 1.228 |
| | Average | - | - | 12.82 | 1.738 | 0.927 |
| NTE-FC | Yearly | 2,108,416 | 6,317,574 | 13.494 | 3.011 | 0.792 |
| | Qarterly | 2,113,536 | 6,322,696 | 10.730 | 1.285 | 0.956 |
| | Monthly | 2,142,208 | 6,351,378 | 14.272 | 1.147 | 1.034 |
| | Other | - | - | 5.785 | 3.746 | 1.199 |
| | Average | - | - | 12.818 | 1.739 | 0.927 |
| NTE-LSTM | Yearly | 2,100,224 | 7,394,310 | 13.676 | 3.11 | 0.81 |
| | Qarterly | 2,105,344 | 7,399,432 | 10.681 | 1.278 | 0.951 |
| | Monthly | 2,134,016 | 7,428,114 | 13.802 | 1.077 | 0.985 |
| | Other | - | - | 5.768 | 3.779 | 1.203 |
| | Average | - | - | 12.622 | 1.728 | 0.917 |
| NTE-Conv1D | Yearly | 2,100,224 | 6,333,449 | 13.475 | 2.992 | 0.789 |
| | Qarterly | 2,105,344 | 6,338,568 | 10.712 | 1.275 | 0.951 |
| | Monthly | 2,134,016 | 6,3672500 | 13.921 | 1.099 | 0.999 |
| | Other | - | - | 6.365 | 4.145 | 1.323 |
| | Average | - | - | 12.67 | 1.729 | 0.919 |
| NTE-BiDConv1D | Yearly | 4,213,760 | 8,575,494 | 14.42 | 3.199 | 0.844 |
| | Qarterly | 3,694,592 | 8,006,152 | 11.174 | 1.335 | 0.994 |
| | Monthly | 3,723,264 | 8,034,834 | 15.206 | 1.211 | 1.096 |
| | Other | - | - | 5.466 | 3.663 | 1.153 |
| | Average | - | - | 13.571 | 1.821 | 0.976 |

Table 12 presents the impact of different NTE structures (e.g., FC, LSTM, Conv1D) and the presence or absence of the linear projection layer on time-series forecasting performance across the ETT, ECL, Weather, Traffic, and Solar-Energy datasets. The results in Table 12 are obtained using a fixed kernel size of 3 and a repeat count of 1. A more comprehensive comparison with various parameter combinations is provided in Appendix D, which demonstrates that component selection has a significant impact on performance.

## B.5 EFFICIENCY ANALYSIS

Tables 13, 14 summarize the averaged efficiency metrics over three datasets (ETTh1, ECL, Exchange) for the vanilla Transformer and iTransformer models, respectively.

Empirically, we find that attaching NTE to a time-tokenized vanilla Transformer leads to degraded forecasting accuracy compared to variants equipped with explicit positional encodings. This outcome suggests that when the backbone itself is order-agnostic, removing positional signals entirely leaves the model unable to recover temporal structure effectively. In contrast, when NTE is applied to feature-tokenized architectures such as iTransformer, performance consistently improves. This indicates that NTE is most effective when temporal encoding is delegated to the preprocessing front end, allowing the Transformer layers to specialize in modeling cross-feature interactions—a separation of concerns that leads to more robust forecasting performance.

Table 8: Performance comparison of various time-series forecasting models and dataset-wise best NTE configurations (MSE / MAE). For **ETTh1** and **ETTh2**, we use a Bi-directional Dilated Convolutional Embedding without linear projection and the Combine method. **ETTm2** shares the same embedding but uses Concat. **ETTm1** adopts a Fully Connected Embedding without projection. **Weather** uses a Combine-based Bi-directional Dilated Convolutional Embedding with projection, while **Traffic** and **Solar-Energy** employ Fully Connected Embeddings with projection.

| Model | iTransformer +NTE(Ours) | | iTransformer | | RLinear | | TimesNet | | FEDformer | | PatchTST | | DLinear | | Autoformer | |
|---|---|---|---|---|---|---|---|---|---|---|---|---|---|---|---|---|
| Metric | MSE | MAE | MSE | MAE | MSE | MAE | MSE | MAE | MSE | MAE | MSE | MAE | MSE | MAE | MSE | MAE |
| **ETTh1** 96 | 0.380 | 0.402 | 0.386 | 0.405 | 0.386 | **0.395** | 0.384 | 0.402 | **0.376** | 0.419 | 0.414 | 0.419 | 0.386 | 0.400 | 0.449 | 0.459 |
| 192 | 0.429 | 0.431 | 0.441 | 0.436 | 0.437 | **0.424** | 0.436 | 0.429 | **0.420** | 0.448 | 0.460 | 0.445 | 0.437 | 0.432 | 0.500 | 0.482 |
| 336 | 0.475 | 0.453 | 0.487 | 0.458 | 0.479 | **0.446** | 0.491 | 0.469 | **0.459** | 0.465 | 0.501 | 0.466 | 0.481 | 0.459 | 0.521 | 0.496 |
| 720 | 0.484 | 0.478 | 0.503 | 0.491 | **0.481** | **0.470** | 0.521 | 0.500 | 0.506 | 0.507 | 0.500 | 0.488 | 0.519 | 0.516 | 0.514 | 0.512 |
| Avg | 0.442 | 0.441 | 0.454 | 0.447 | 0.446 | **0.434** | 0.458 | 0.450 | **0.440** | 0.460 | 0.469 | 0.454 | 0.456 | 0.452 | 0.496 | 0.487 |
| **ETTh2** 96 | 0.295 | 0.347 | 0.297 | 0.349 | **0.288** | **0.338** | 0.340 | 0.374 | 0.358 | 0.397 | 0.302 | 0.348 | 0.333 | 0.387 | 0.346 | 0.388 |
| 192 | 0.378 | 0.399 | 0.380 | 0.400 | **0.374** | **0.390** | 0.402 | 0.414 | 0.429 | 0.439 | 0.388 | 0.400 | 0.477 | 0.476 | 0.456 | 0.452 |
| 336 | 0.418 | 0.430 | 0.428 | 0.432 | **0.415** | **0.426** | 0.452 | 0.452 | 0.496 | 0.487 | 0.426 | 0.433 | 0.594 | 0.541 | 0.482 | 0.486 |
| 720 | 0.422 | 0.443 | 0.427 | 0.445 | **0.420** | **0.440** | 0.462 | 0.468 | 0.463 | 0.474 | 0.431 | 0.446 | 0.831 | 0.657 | 0.515 | 0.511 |
| Avg | 0.378 | 0.405 | 0.383 | 0.407 | **0.374** | **0.398** | 0.414 | 0.427 | 0.437 | 0.449 | 0.387 | 0.407 | 0.559 | 0.515 | 0.450 | 0.459 |
| **ETTm1** 96 | **0.329** | 0.369 | 0.334 | 0.368 | 0.355 | 0.376 | 0.338 | 0.375 | 0.379 | 0.419 | 0.329 | **0.367** | 0.345 | 0.372 | 0.505 | 0.475 |
| 192 | 0.372 | 0.391 | 0.377 | 0.391 | 0.391 | 0.392 | 0.374 | 0.387 | 0.426 | 0.441 | **0.367** | **0.385** | 0.380 | 0.389 | 0.553 | 0.496 |
| 336 | 0.404 | 0.412 | 0.426 | 0.420 | 0.424 | 0.415 | 0.410 | 0.411 | 0.445 | 0.459 | **0.399** | **0.410** | 0.413 | 0.413 | 0.621 | 0.537 |
| 720 | 0.473 | 0.450 | 0.491 | 0.459 | 0.487 | 0.450 | 0.478 | 0.450 | 0.543 | 0.490 | **0.454** | **0.439** | 0.474 | 0.453 | 0.671 | 0.561 |
| Avg | 0.395 | 0.406 | 0.407 | 0.410 | 0.414 | 0.407 | 0.400 | 0.406 | 0.448 | 0.452 | **0.387** | **0.400** | 0.403 | 0.407 | 0.588 | 0.517 |
| **ETTm2** 96 | 0.179 | 0.264 | 0.180 | 0.264 | 0.182 | 0.265 | 0.187 | 0.267 | 0.203 | 0.287 | **0.175** | **0.259** | 0.193 | 0.292 | 0.255 | 0.339 |
| 192 | 0.251 | 0.310 | 0.250 | 0.309 | 0.246 | 0.304 | 0.249 | 0.309 | 0.269 | 0.328 | **0.241** | **0.302** | 0.284 | 0.362 | 0.281 | 0.340 |
| 336 | 0.311 | 0.346 | 0.311 | 0.348 | 0.307 | **0.342** | 0.321 | 0.351 | 0.325 | 0.366 | **0.305** | 0.343 | 0.369 | 0.427 | 0.339 | 0.372 |
| 720 | 0.404 | 0.401 | 0.412 | 0.407 | 0.407 | **0.398** | 0.408 | 0.403 | 0.421 | 0.415 | **0.402** | 0.400 | 0.554 | 0.522 | 0.433 | 0.432 |
| Avg | 0.286 | 0.330 | 0.288 | 0.332 | 0.286 | 0.327 | 0.291 | 0.333 | 0.305 | 0.349 | **0.281** | **0.326** | 0.350 | 0.401 | 0.327 | 0.371 |
| **ECL** 96 | **0.140** | **0.236** | 0.148 | 0.240 | 0.201 | 0.281 | 0.168 | 0.272 | 0.193 | 0.308 | 0.181 | 0.270 | 0.197 | 0.282 | 0.201 | 0.317 |
| 192 | **0.157** | **0.250** | 0.162 | 0.253 | 0.201 | 0.283 | 0.184 | 0.289 | 0.201 | 0.315 | 0.188 | 0.274 | 0.196 | 0.285 | 0.222 | 0.334 |
| 336 | **0.170** | **0.266** | 0.178 | 0.269 | 0.215 | 0.298 | 0.198 | 0.300 | 0.214 | 0.329 | 0.204 | 0.293 | 0.209 | 0.301 | 0.231 | 0.338 |
| 720 | **0.201** | **0.295** | 0.225 | 0.317 | 0.257 | 0.331 | 0.220 | 0.320 | 0.246 | 0.355 | 0.246 | 0.324 | 0.245 | 0.333 | 0.254 | 0.361 |
| Avg | **0.167** | **0.262** | 0.178 | 0.270 | 0.219 | 0.298 | 0.192 | 0.295 | 0.214 | 0.327 | 0.205 | 0.290 | 0.212 | 0.300 | 0.227 | 0.338 |
| **Weather** 96 | **0.168** | **0.212** | 0.174 | 0.214 | 0.192 | 0.232 | 0.172 | 0.220 | 0.217 | 0.296 | 0.177 | 0.218 | 0.196 | 0.255 | 0.266 | 0.336 |
| 192 | **0.217** | 0.257 | 0.221 | 0.254 | 0.240 | 0.271 | 0.219 | 0.261 | 0.276 | 0.336 | 0.225 | 0.259 | 0.237 | 0.296 | 0.307 | 0.367 |
| 336 | **0.269** | **0.296** | 0.278 | 0.296 | 0.292 | 0.307 | 0.280 | 0.306 | 0.339 | 0.380 | 0.278 | 0.297 | 0.283 | 0.335 | 0.359 | 0.395 |
| 720 | 0.349 | **0.347** | 0.358 | 0.347 | 0.364 | 0.353 | 0.365 | 0.359 | 0.403 | 0.428 | 0.354 | 0.348 | **0.345** | 0.381 | 0.419 | 0.428 |
| Avg | **0.251** | **0.278** | 0.258 | 0.278 | 0.272 | 0.291 | 0.259 | 0.287 | 0.309 | 0.360 | 0.259 | 0.281 | 0.265 | 0.317 | 0.338 | 0.382 |
| **Traffic** 96 | 0.398 | 0.272 | **0.395** | **0.268** | 0.649 | 0.389 | 0.593 | 0.321 | 0.587 | 0.366 | 0.462 | 0.295 | 0.650 | 0.396 | 0.613 | 0.388 |
| 192 | 0.421 | 0.283 | **0.417** | 0.276 | 0.601 | 0.366 | 0.617 | 0.336 | 0.604 | 0.373 | 0.466 | 0.296 | 0.598 | 0.370 | 0.616 | 0.382 |
| 336 | 0.433 | 0.289 | **0.433** | 0.283 | 0.609 | 0.369 | 0.629 | 0.336 | 0.621 | 0.383 | 0.482 | 0.304 | 0.605 | 0.373 | 0.622 | 0.337 |
| 720 | **0.464** | 0.305 | 0.467 | 0.302 | 0.647 | 0.387 | 0.640 | 0.350 | 0.626 | 0.382 | 0.514 | 0.322 | 0.645 | 0.394 | 0.660 | 0.408 |
| Avg | 0.429 | 0.287 | **0.428** | 0.282 | 0.626 | 0.378 | 0.620 | 0.336 | 0.610 | 0.376 | 0.481 | 0.304 | 0.625 | 0.383 | 0.628 | 0.379 |
| **Solar Energy** 96 | 0.205 | **0.235** | 0.203 | 0.237 | 0.322 | 0.339 | 0.250 | 0.292 | 0.242 | 0.342 | 0.234 | 0.286 | 0.290 | 0.378 | 0.884 | 0.711 |
| 192 | 0.236 | **0.260** | **0.233** | 0.261 | 0.359 | 0.356 | 0.296 | 0.318 | 0.285 | 0.380 | 0.267 | 0.310 | 0.320 | 0.398 | 0.834 | 0.692 |
| 336 | 0.246 | 0.270 | 0.248 | 0.273 | 0.397 | 0.369 | 0.319 | 0.330 | 0.282 | 0.376 | 0.290 | 0.315 | 0.353 | 0.415 | 0.941 | 0.723 |
| 720 | 0.250 | 0.276 | 0.249 | 0.275 | 0.397 | 0.356 | 0.338 | 0.337 | 0.357 | 0.427 | 0.289 | 0.317 | 0.356 | 0.413 | 0.882 | 0.717 |
| Avg | 0.234 | **0.260** | **0.233** | 0.262 | 0.369 | 0.356 | 0.301 | 0.319 | 0.291 | 0.381 | 0.270 | 0.307 | 0.330 | 0.401 | 0.885 | 0.711 |
| **1ˢᵗ Count** | 12 | 13 | 8 | 9 | 6 | 12 | 0 | 0 | 4 | 0 | 9 | 8 | 1 | 0 | 0 | 0 |

Table 9: Performance comparison of various time-series forecasting models and dataset-wise best NTE configurations (MSE / MAE). For **ETTh1** and **ETTh2**, we use a Bi-directional Dilated Convolutional Embedding without linear projection and the Combine method. **ETTm2** shares the same embedding but uses Concat. **ETTm1** adopts a Fully Connected Embedding without projection. **Weather** uses a Combine-based Bi-directional Dilated Convolutional Embedding with projection, while **Traffic** and **Solar-Energy** employ Fully Connected Embeddings with projection.

| Model | | iTransformer +NTE(Ours) | | Ada MSHyper | | Timemixer++ | | S-Mamba | | SDE-Mamba | | SAMformer | |
|---|---|---|---|---|---|---|---|---|---|---|---|---|---|
| Metric | | MSE | MAE | MSE | MAE | MSE | MAE | MSE | MAE | MSE | MAE | MSE | MAE |
| ETTh1 | 96 | 0.380 | 0.402 | 0.372 | **0.393** | **0.361** | 0.403 | 0.386 | 0.405 | 0.376 | 0.400 | 0.381 | 0.402 |
| | 192 | 0.429 | 0.431 | 0.433 | **0.417** | 0.416 | 0.441 | 0.443 | 0.437 | 0.432 | 0.429 | **0.409** | 0.418 |
| | 336 | 0.475 | 0.453 | **0.422** | 0.433 | 0.430 | 0.434 | 0.489 | 0.468 | 0.477 | 0.437 | 0.423 | **0.425** |
| | 720 | 0.484 | 0.478 | 0.445 | 0.459 | 0.467 | 0.451 | 0.502 | 0.489 | 0.488 | 0.471 | **0.427** | **0.449** |
| | Avg | 0.442 | 0.441 | 0.418 | 0.426 | 0.419 | 0.432 | 0.455 | 0.450 | 0.443 | 0.432 | **0.410** | **0.424** |
| ETTh2 | 96 | 0.295 | 0.347 | 0.283 | 0.332 | **0.276** | **0.328** | 0.296 | 0.348 | 0.288 | 0.340 | 0.295 | 0.358 |
| | 192 | 0.378 | 0.399 | 0.358 | **0.374** | 0.342 | 0.379 | 0.376 | 0.396 | 0.373 | 0.390 | **0.340** | 0.386 |
| | 336 | 0.418 | 0.430 | 0.428 | 0.437 | **0.346** | 0.398 | 0.424 | 0.431 | 0.380 | 0.406 | 0.350 | **0.395** |
| | 720 | 0.422 | 0.443 | 0.413 | 0.432 | 0.392 | **0.415** | 0.426 | 0.444 | 0.412 | 0.432 | **0.391** | 0.428 |
| | Avg | 0.378 | 0.405 | 0.371 | 0.394 | **0.339** | **0.380** | 0.381 | 0.405 | 0.363 | 0.392 | 0.344 | 0.392 |
| ETTm1 | 96 | 0.329 | 0.369 | **0.301** | 0.354 | 0.310 | **0.334** | 0.333 | 0.368 | 0.315 | 0.357 | 0.329 | 0.363 |
| | 192 | 0.372 | 0.391 | **0.345** | 0.375 | 0.348 | **0.362** | 0.376 | 0.390 | 0.360 | 0.383 | 0.353 | 0.378 |
| | 336 | 0.404 | 0.412 | **0.375** | 0.397 | 0.376 | **0.391** | 0.408 | 0.413 | 0.389 | 0.405 | 0.382 | 0.394 |
| | 720 | 0.473 | 0.450 | 0.437 | 0.435 | 0.440 | **0.423** | 0.475 | 0.448 | 0.448 | 0.440 | **0.429** | 0.418 |
| | Avg | 0.395 | 0.406 | **0.365** | 0.390 | 0.369 | **0.378** | 0.398 | 0.405 | 0.378 | 0.394 | 0.373 | 0.388 |
| ETTm2 | 96 | 0.179 | 0.264 | **0.165** | 0.257 | 0.170 | **0.245** | 0.179 | 0.263 | 0.172 | 0.259 | 0.181 | 0.274 |
| | 192 | 0.251 | 0.310 | 0.230 | 0.307 | **0.229** | **0.291** | 0.250 | 0.309 | 0.238 | 0.301 | 0.233 | 0.306 |
| | 336 | 0.311 | 0.346 | **0.282** | **0.328** | 0.303 | 0.343 | 0.312 | 0.349 | 0.300 | 0.340 | 0.285 | 0.338 |
| | 720 | 0.404 | 0.401 | 0.375 | 0.396 | **0.373** | 0.399 | 0.411 | 0.406 | 0.394 | 0.394 | 0.375 | **0.390** |
| | Avg | 0.286 | 0.330 | **0.263** | 0.322 | 0.269 | **0.320** | 0.288 | 0.332 | 0.276 | 0.322 | 0.269 | 0.327 |
| ECL | 96 | 0.140 | 0.236 | **0.135** | 0.238 | **0.135** | **0.222** | 0.139 | 0.235 | 0.146 | 0.244 | 0.155 | 0.252 |
| | 192 | 0.157 | 0.250 | 0.152 | 0.239 | **0.147** | **0.235** | 0.159 | 0.255 | 0.162 | 0.258 | 0.168 | 0.263 |
| | 336 | 0.170 | 0.266 | 0.168 | 0.266 | **0.164** | **0.245** | 0.176 | 0.272 | 0.177 | 0.274 | 0.183 | 0.277 |
| | 720 | **0.201** | **0.295** | 0.212 | 0.293 | 0.212 | 0.310 | 0.204 | 0.298 | 0.202 | 0.297 | 0.219 | 0.306 |
| | Avg | 0.167 | 0.262 | 0.167 | 0.259 | **0.165** | **0.253** | 0.170 | 0.265 | 0.172 | 0.268 | 0.181 | 0.274 |
| Weather | 96 | 0.168 | 0.212 | 0.157 | **0.195** | **0.155** | 0.205 | 0.165 | 0.210 | 0.165 | 0.214 | 0.197 | 0.249 |
| | 192 | 0.217 | 0.257 | 0.218 | 0.259 | **0.201** | **0.245** | 0.214 | 0.252 | 0.214 | 0.255 | 0.235 | 0.277 |
| | 336 | 0.269 | 0.296 | 0.251 | 0.252 | **0.237** | 0.265 | 0.274 | 0.297 | 0.271 | 0.297 | 0.276 | 0.304 |
| | 720 | 0.349 | 0.347 | **0.304** | **0.328** | 0.312 | 0.334 | 0.350 | 0.345 | 0.346 | 0.347 | 0.334 | 0.342 |
| | Avg | 0.251 | 0.278 | 0.232 | **0.259** | **0.226** | 0.262 | 0.251 | 0.276 | 0.249 | 0.278 | 0.261 | 0.293 |
| Traffic | 96 | 0.398 | 0.272 | 0.384 | **0.248** | **0.382** | 0.253 | **0.382** | 0.261 | 0.388 | 0.261 | 0.407 | 0.292 |
| | 192 | 0.421 | 0.283 | 0.401 | **0.258** | 0.402 | **0.258** | **0.396** | 0.267 | 0.411 | 0.271 | 0.415 | 0.294 |
| | 336 | 0.433 | 0.289 | 0.423 | **0.261** | 0.428 | 0.263 | **0.417** | 0.276 | 0.428 | 0.278 | 0.421 | 0.292 |
| | 720 | 0.464 | 0.305 | 0.453 | **0.282** | **0.441** | **0.282** | 0.460 | 0.300 | 0.461 | 0.297 | 0.456 | 0.311 |
| | Avg | 0.429 | 0.287 | 0.415 | **0.262** | **0.414** | 0.264 | **0.414** | 0.276 | 0.422 | 0.276 | 0.425 | 0.297 |
| Solar Energy | 96 | 0.205 | 0.235 | - | - | **0.171** | 0.231 | 0.205 | 0.244 | 0.186 | **0.217** | - | - |
| | 192 | 0.236 | 0.260 | - | - | **0.218** | 0.263 | 0.237 | 0.270 | 0.230 | **0.251** | - | - |
| | 336 | 0.246 | 0.270 | - | - | **0.212** | **0.269** | 0.258 | 0.288 | 0.253 | 0.270 | - | - |
| | 720 | 0.250 | 0.276 | - | - | **0.212** | **0.270** | 0.260 | 0.288 | 0.247 | 0.274 | - | - |
| | Avg | 0.234 | 0.260 | - | - | 0.203 | 0.238 | 0.240 | 0.273 | 0.229 | 0.253 | - | - |
| 1st Count | | 1 | 1 | 10 | 13 | 21 | 20 | 4 | 0 | 0 | 2 | 6 | 5 |

Table 10: Forecasting performance (mean ± std) of the proposed NTE-based model on the ETT dataset (ETTh1, ETTh2, ETTm1, ETTm2) over 5 runs with different random seeds.

| Dataset | ETTh1 | | ETTh2 | | ETTm1 | | ETTm2 | |
|---|---|---|---|---|---|---|---|---|
| Metric | MSE | MAE | MSE | MAE | MSE | MAE | MSE | MAE |
| 96 | $0.379_{\pm 0.000}$ | $0.401_{\pm 0.000}$ | $0.297_{\pm 0.003}$ | $0.348_{\pm 0.003}$ | $0.332_{\pm 0.001}$ | $0.369_{\pm 0.001}$ | $0.181_{\pm 0.003}$ | $0.265_{\pm 0.002}$ |
| 192 | $0.431_{\pm 0.001}$ | $0.431_{\pm 0.000}$ | $0.379_{\pm 0.001}$ | $0.399_{\pm 0.001}$ | $0.374_{\pm 0.000}$ | $0.392_{\pm 0.000}$ | $0.251_{\pm 0.001}$ | $0.310_{\pm 0.001}$ |
| 336 | $0.473_{\pm 0.002}$ | $0.451_{\pm 0.001}$ | $0.418_{\pm 0.003}$ | $0.430_{\pm 0.002}$ | $0.409_{\pm 0.001}$ | $0.415_{\pm 0.000}$ | $0.313_{\pm 0.001}$ | $0.348_{\pm 0.001}$ |
| 720 | $0.489_{\pm 0.003}$ | $0.481_{\pm 0.001}$ | $0.422_{\pm 0.002}$ | $0.443_{\pm 0.001}$ | $0.485_{\pm 0.001}$ | $0.457_{\pm 0.001}$ | $0.407_{\pm 0.000}$ | $0.404_{\pm 0.000}$ |

Table 11: Forecasting performance (mean ± std) of the proposed NTE-based model on ECL, Weather, Traffic, and Solar-Energy datasets over 5 runs with different random seeds.

| Dataset | ECL | | Weather | | Traffic | | Solar-Energy | |
|---|---|---|---|---|---|---|---|---|
| Metric | MSE | MAE | MSE | MAE | MSE | MAE | MSE | MAE |
| 96 | $0.140_{\pm 0.000}$ | $0.235_{\pm 0.000}$ | $0.170_{\pm 0.003}$ | $0.215_{\pm 0.003}$ | $0.399_{\pm 0.001}$ | $0.273_{\pm 0.001}$ | $0.203_{\pm 0.003}$ | $0.233_{\pm 0.002}$ |
| 192 | $0.157_{\pm 0.001}$ | $0.250_{\pm 0.000}$ | $0.217_{\pm 0.001}$ | $0.257_{\pm 0.001}$ | $0.421_{\pm 0.000}$ | $0.283_{\pm 0.000}$ | $0.236_{\pm 0.001}$ | $0.259_{\pm 0.001}$ |
| 336 | $0.172_{\pm 0.002}$ | $0.267_{\pm 0.001}$ | $0.275_{\pm 0.003}$ | $0.299_{\pm 0.002}$ | $0.433_{\pm 0.001}$ | $0.289_{\pm 0.000}$ | $0.247_{\pm 0.001}$ | $0.271_{\pm 0.001}$ |
| 720 | $0.203_{\pm 0.003}$ | $0.296_{\pm 0.001}$ | $0.353_{\pm 0.002}$ | $0.349_{\pm 0.001}$ | $0.465_{\pm 0.001}$ | $0.306_{\pm 0.001}$ | $0.250_{\pm 0.000}$ | $0.276_{\pm 0.000}$ |

Table 12: Performance comparison of different NTE configurations and linear projection removal (ETTh, ECL, Weather, Traffic, Solar-Energy dataset, MSE / MAE average).

| Value Embedding (Linear) | NTE | ETTh1 | | ETTh2 | | ETTm1 | | ETTm2 | | ECL | | Weather | | Traffic | | SolarEnergy | |
|---|---|---|---|---|---|---|---|---|---|---|---|---|---|---|---|---|---|
| | | MSE | MAE | MSE | MAE | MSE | MAE | MSE | MAE | MSE | MAE | MSE | MAE | MSE | MAE | MSE | MAE |
| ✓ | w/o | 0.454 | 0.447 | 0.383 | 0.407 | 0.407 | 0.410 | 0.288 | 0.332 | 0.178 | 0.270 | 0.258 | 0.278 | **0.428** | **0.282** | **0.233** | 0.262 |
| | FC | 0.450 | 0.443 | 0.384 | 0.409 | 0.402 | 0.406 | 0.287 | 0.332 | 0.182 | 0.272 | 0.259 | 0.280 | 0.429 | 0.287 | 0.234 | **0.260** |
| | LSTM | 0.450 | 0.444 | 0.388 | 0.412 | 0.396 | 0.404 | 0.289 | 0.335 | **0.167** | **0.262** | 0.253 | 0.278 | 0.432 | 0.288 | 0.248 | 0.269 |
| | Conv1D | 0.464 | 0.455 | 0.397 | 0.417 | 0.415 | 0.417 | 0.296 | 0.340 | 0.182 | 0.275 | 0.256 | 0.281 | 0.437 | 0.293 | 0.243 | 0.266 |
| | Dilated Conv1D | 0.445 | 0.442 | 0.385 | 0.408 | 0.402 | 0.409 | 0.293 | 0.336 | 0.171 | 0.264 | 0.255 | 0.282 | 1.223 | 0.682 | 0.295 | 0.309 |
| | Bi-Dilated Conv1D (Concat) | 0.459 | 0.452 | 0.388 | 0.410 | 0.396 | 0.418 | 0.287 | **0.330** | 0.174 | 0.270 | 0.255 | 0.281 | 1.456 | 0.795 | 0.595 | 0.499 |
| | Bi-Dilated Conv1D (Combine) | 0.446 | 0.443 | 0.383 | 0.408 | **0.395** | 0.404 | 0.291 | 0.334 | 0.172 | 0.266 | **0.251** | 0.278 | 1.174 | 0.650 | 0.304 | 0.317 |
| w/o | FC | 0.455 | 0.447 | 0.394 | 0.414 | **0.395** | 0.406 | 0.287 | 0.332 | 0.175 | 0.268 | 0.259 | **0.259** | 0.430 | 0.288 | 0.235 | 0.261 |
| | LSTM | 0.456 | 0.446 | 0.390 | 0.414 | 0.396 | 0.405 | 0.296 | 0.338 | **0.167** | **0.262** | 0.253 | 0.278 | 0.429 | 0.285 | 0.243 | 0.267 |
| | Conv1D | 0.467 | 0.455 | 0.398 | 0.419 | 0.417 | 0.418 | 0.296 | 0.339 | 0.173 | 0.268 | 0.257 | 0.282 | 0.439 | 0.295 | 0.238 | 0.266 |
| | Dilated Conv1D | 0.446 | 0.443 | 0.383 | **0.383** | 0.401 | **0.401** | 0.290 | 0.333 | 0.172 | 0.270 | 0.258 | 0.283 | 0.538 | 0.373 | 0.265 | 0.289 |
| | Bi-Dilated Conv1D (Concat) | 0.472 | 0.459 | 0.383 | 0.407 | 0.402 | 0.409 | **0.286** | **0.330** | 0.172 | 0.268 | 0.256 | 0.282 | 1.368 | 0.759 | 0.291 | 0.314 |
| | Bi-Dilated Conv1D (Combine) | **0.442** | **0.441** | **0.378** | 0.405 | 0.396 | 0.404 | 0.292 | 0.335 | **0.177** | 0.269 | 0.254 | 0.279 | 0.538 | 0.374 | 0.273 | 0.295 |

Table 13: Efficiency analysis of the vanilla Transformer under different input designs across ETTh1, ECL, and Exchange. We report the averages of FLOPs, number of parameters, per-sample inference time, end-to-end latency, peak GPU memory usage, and forecasting accuracy (MSE/MAE) over four forecasting horizons (96, 192, 336, 720) for several PE variants (Sinusoidal, Learnable, TUPE, ConvSPE) and NTE-based front-ends (FC, LSTM, Conv1D, BiDConv1D).

| Model / Dataset | Transformer / ETTh1 | | | | | | |
|---|---|---|---|---|---|---|---|
| Metric | FLOPs | Parameters | per-sample inference time (s) | End-to-end inference latency (s) | Peak GPU memory usage (MB) | MSE | MAE |
| *SinuPE* | 4,197,888 | - | 0.34 | 26.43 | 820.96 | 1.010 | 0.797 |
| *LPE* | 4,197,888 | 15,952,647 | 0.34 | 26.79 | 840.78 | 1.018 | 0.800 |
| *TUPE* | 16,780,800 | 26,192,647 | 0.34 | 26.64 | 880.94 | 1.033 | 0.802 |
| *ConvSPE* | 4,197,888 | 11,360,009 | 0.34 | 26.63 | 824.98 | **0.986** | **0.788** |
| *NTE-FC* | 4,205,056 | 8,418,911 | 0.34 | 26.53 | 801.20 | 1.443 | 1.013 |
| *NTE-LSTM* | 4,197,888 | 12,988,935 | 0.35 | 27.24 | 818.37 | 1.126 | 0.864 |
| *NTE-Conv1D* | 4,197,888 | 10,798,599 | 0.34 | 26.80 | 801.40 | 1.330 | 0.952 |
| *NTE-BiDConv1D* | 11,027,968 | 17,868,295 | 0.34 | 27.11 | 856.57 | 1.207 | 0.897 |
| | Transformer / ECL | | | | | | |
| *SinuPE* | 4,194,816 | 10,811,137 | 0.18 | 27.31 | 824.28 | **0.388** | 0.458 |
| *LPE* | 4,194,816 | 15,931,137 | 0.18 | 27.27 | 844.47 | 0.389 | **0.457** |
| *TUPE* | 16,777,728 | 26,171,137 | 0.18 | 27.31 | 885.30 | 0.406 | 0.476 |
| *ConvSPE* | 4,194,816 | 11,338,499 | 0.19 | 28.47 | 828.31 | 0.441 | 0.491 |
| *NTE-FC* | 4,195,840 | 10,853,889 | 0.18 | 27.21 | 805.42 | 1.000 | 0.846 |
| *NTE-LSTM* | 4,195,840 | 12,961,281 | 0.18 | 28.70 | 821.80 | 0.758 | 0.667 |
| *NTE-Conv1D* | 4,195,840 | 10,777,089 | 0.18 | 27.29 | 804.90 | 0.959 | 0.791 |
| *NTE-BiDConv1D* | 11,012,608 | 17,644,033 | 0.18 | 28.27 | 859.50 | 0.913 | 0.744 |
| | Transformer / Exchange | | | | | | |
| *SinuPE* | 4,198,400 | 10,836,232 | 0.74 | 26.26 | 824.28 | 1.503 | 0.935 |
| *LPE* | 4,198,400 | 15,956,232 | 0.74 | 26.19 | 843.26 | 1.773 | 1.067 |
| *TUPE* | 16,781,312 | 26,196,232 | 0.74 | 26.23 | 883.36 | 1.844 | 1.085 |
| *ConvSPE* | 4,198,400 | 11,363,594 | 0.74 | 26.26 | 828.30 | 1.822 | 1.061 |
| *NTE-FC* | 4,206,592 | 10,864,648 | 0.74 | 26.21 | 804.96 | **1.339** | **0.904** |
| *NTE-LSTM* | 4,198,400 | 12,993,544 | 0.75 | 26.49 | 821.79 | 1.470 | 0.973 |
| *NTE-Conv1D* | 4,198,400 | 10,802,184 | 0.73 | 25.84 | 803.88 | 1.540 | 0.984 |
| *NTE-BiDConv1D* | 9,195,520 | 15,981,064 | 0.76 | 26.65 | 845.25 | 1.445 | 0.946 |

Table 14: Efficiency analysis of the iTransformer under different input designs across ETTh1, ECL, and Exchange. We report the averages of FLOPs, number of parameters, per-sample inference time, end-to-end latency, peak GPU memory usage, and forecasting accuracy (MSE/MAE) over four forecasting horizons (96, 192, 336, 720) for several PE variants (Sinusoidal, Learnable, TUPE, ConvSPE) and NTE-based front-ends (FC, LSTM, Conv1D, BiDConv1D).

| Model / Dataset | iTransformer / ETTh1 | | | | | | |
|---|---|---|---|---|---|---|---|
| Metric | FLOPs | Parameters | per-sample inference time (s) | End-to-end inference latency (s) | Peak GPU memory usage (MB) | MSE | MAE |
| *SinuPE* | 634,880 | 936,144 | 0.33 | 26.14 | 31.72 | 0.446 | 0.442 |
| *LPE* | 634,880 | 2,216,144 | 0.33 | 26.30 | 36.60 | 0.443 | **0.440** |
| *TUPE* | 634,880 | 936,144 | 0.33 | 26.04 | 26.84 | 0.444 | 0.441 |
| *ConvSPE* | 634,880 | 977,449 | 0.33 | 26.25 | 32.23 | 0.448 | 0.443 |
| *NTE-FC* | 634,880 | 960,976 | 0.33 | 26.09 | 26.93 | 0.455 | 0.447 |
| *NTE-LSTM* | 610,304 | 1,298,640 | 0.33 | 26.11 | 38.26 | 0.452 | 0.445 |
| *NTE-Conv1D* | 610,304 | 985,040 | 0.33 | 26.24 | 27.21 | 0.467 | 0.455 |
| *NTE-BiDConv1D* | 1,449,984 | 2,590,688 | 0.30 | 26.67 | 41.36 | **0.442** | 0.441 |
| | iTransformer / ECL | | | | | | |
| *SinuPE* | 3,366,912 | 5,154,000 | 0.09 | 28.99 | 285.50 | 0.170 | 0.265 |
| *LPE* | 3,366,912 | 7,714,000 | 0.10 | 28.92 | 295.70 | 0.161 | 0.257 |
| *TUPE* | 3,366,912 | 5,154,000 | 0.10 | 29.19 | 276.01 | 0.174 | 0.265 |
| *ConvSPE* | 3,366,912 | 5,417,681 | 0.10 | 29.92 | 287.52 | 0.168 | 0.263 |
| *NTE-FC* | 3,366,912 | 5,203,664 | 0.10 | 29.77 | 276.01 | 0.175 | 0.268 |
| *NTE-LSTM* | 3,317,760 | 6,403,280 | 0.10 | 31.25 | 285.45 | **0.167** | **0.262** |
| *NTE-Conv1D* | 3,317,760 | 5,251,792 | 0.10 | 29.41 | 276.48 | 0.173 | 0.268 |
| *NTE-BiDConv1D* | 6,823,936 | 10,290,384 | 0.10 | 30.41 | 314.82 | 0.177 | 0.269 |
| | iTransformer / Exchange | | | | | | |
| *SinuPE* | 186,368 | 263,440 | 0.73 | 25.75 | 23.80 | 0.363 | 0.407 |
| *LPE* | 186,368 | 903,440 | 0.73 | 25.74 | 26.24 | 0.363 | 0.406 |
| *TUPE* | 186,368 | 263,440 | 0.73 | 25.78 | 21.36 | 0.362 | 0.405 |
| *ConvSPE* | 186,368 | 280,209 | 0.73 | 25.69 | 23.93 | 0.365 | 0.407 |
| *NTE-FC* | 186,368 | 275,856 | 0.73 | 25.74 | 21.40 | **0.356** | **0.404** |
| *NTE-LSTM* | 174,080 | 379,152 | 0.73 | 25.80 | 32.36 | 0.361 | 0.410 |
| *NTE-Conv1D* | 174,080 | 287,888 | 0.73 | 25.86 | 21.54 | 0.366 | 0.411 |
| *NTE-BiDConv1D* | 354,304 | 721,744 | 0.73 | 25.80 | 24.92 | 0.367 | 0.409 |

# C  VISUALIZATION OF FORECASTING RESULTS

To qualitatively evaluate the time-series forecasting performance of our NTE-based iTransformer, we visualizes its prediction results on **ETT(ETTm2)**, **ECL**, **Weather**, **Traffic** and **Solar-Energy**. All visualizations use input length $T = 96$, kernel size 3, and repeat count 1. Each figure demonstrates the forecasting performance and compares the predicted values against the ground truth over various forecasting horizons $S \in \{96, 192, 336, 720\}$. The x-axis represents the time index, and the y-axis denotes the normalized value of the target variable. The blue lines represent the model predictions, while the orange lines indicate the ground truth values. Figures 5 and 6 present the results using a linear projection, while Figures 7, 8, and 9 show the results without it.

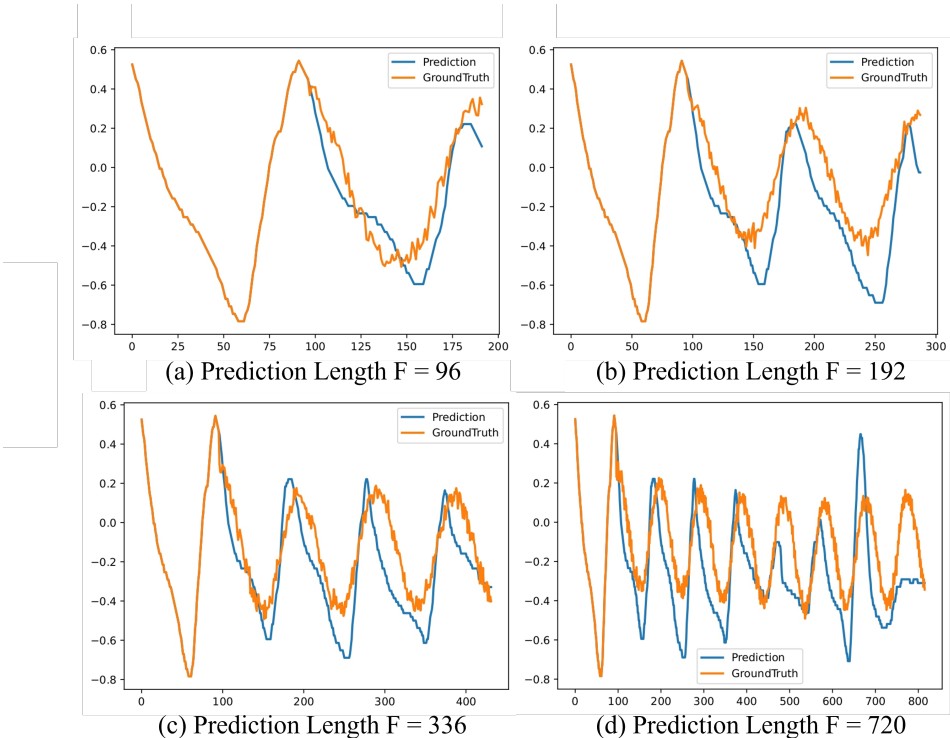

(a) Prediction Length F = 96

(b) Prediction Length F = 192

(c) Prediction Length F = 336

(d) Prediction Length F = 720

Figure 5: Forecasting results on the ETT(ETTm2) dataset across multiple prediction lengths.

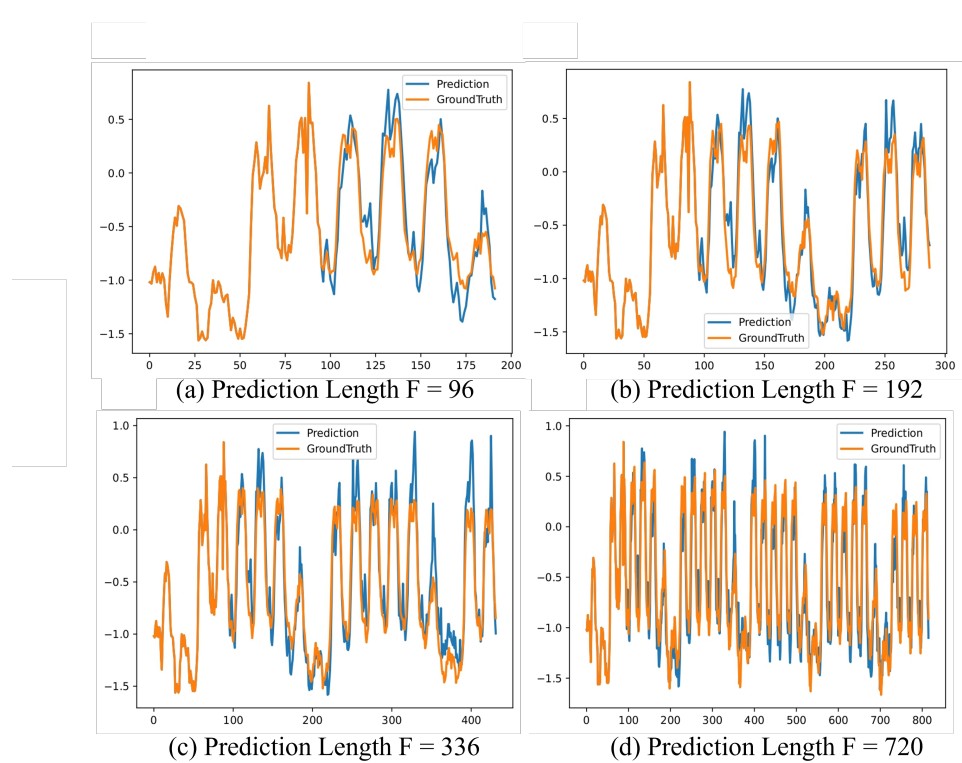

Figure 6: Forecasting results on the ECL dataset across multiple prediction lengths.

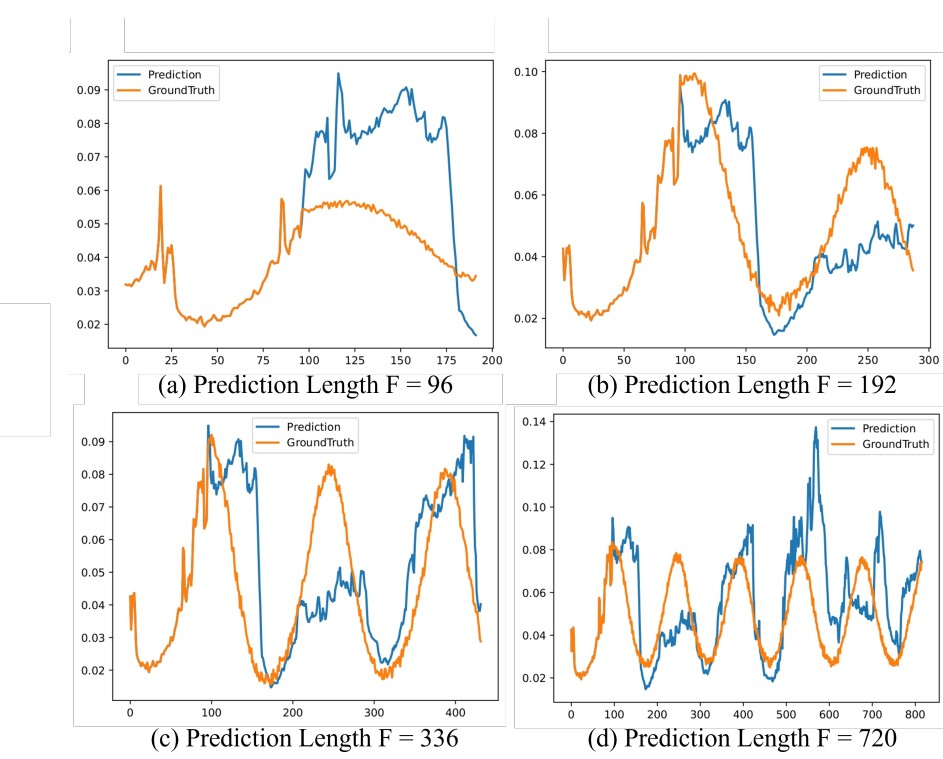

Figure 7: Forecasting results on the Weather dataset across multiple prediction lengths.

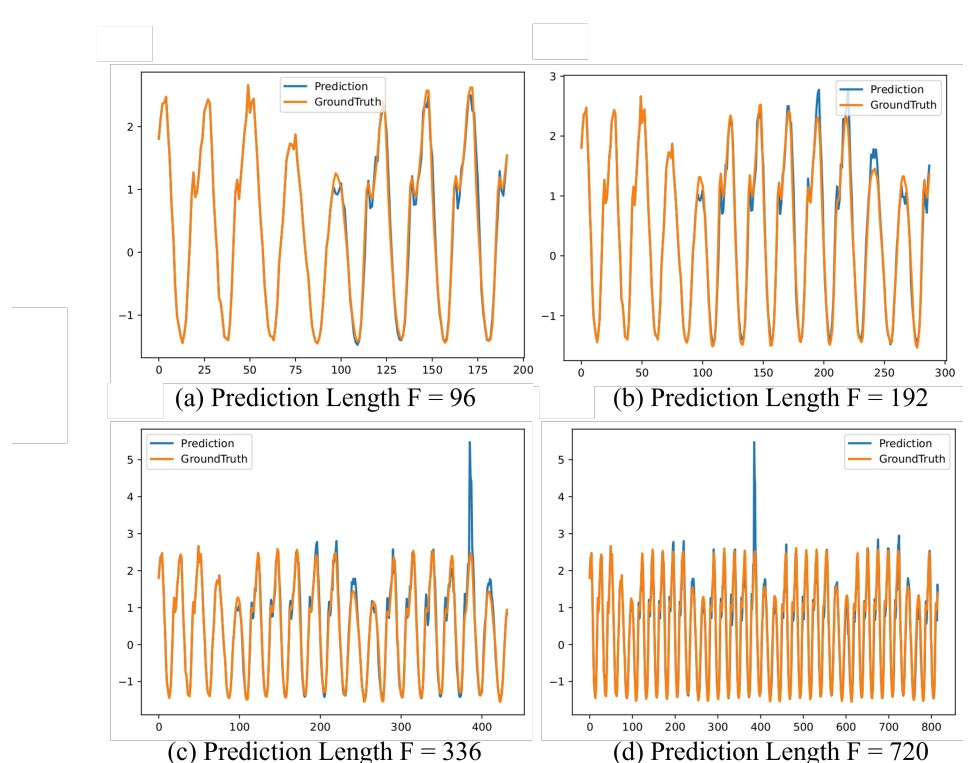

Figure 8: Forecasting results on the Traffic dataset across multiple prediction lengths.

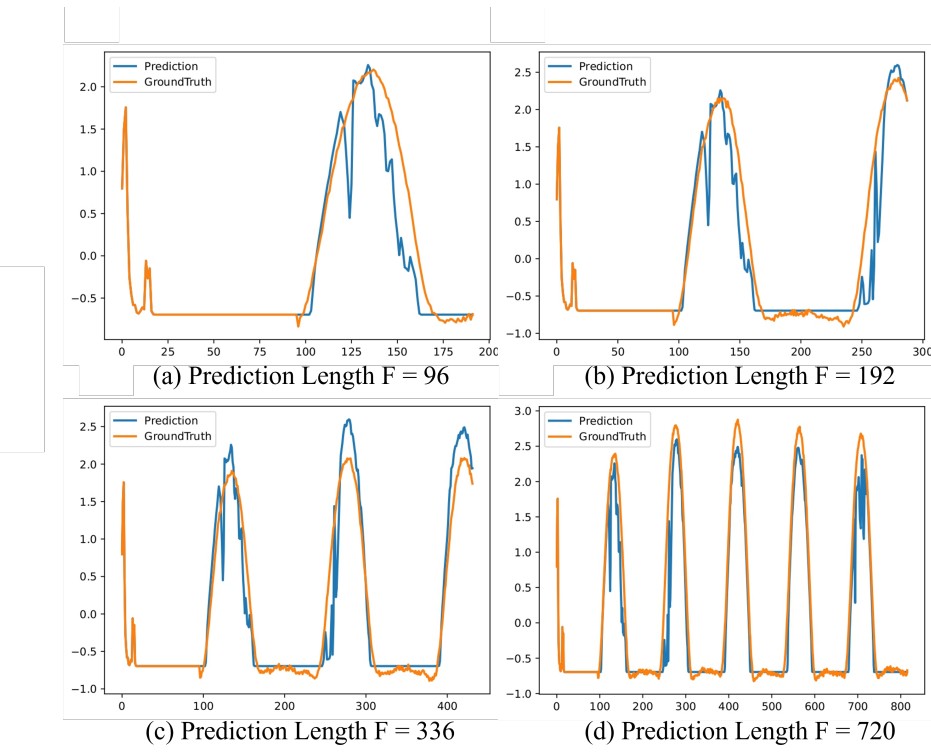

Figure 9: Forecasting results on the Solar-Energy dataset across multiple prediction lengths.

# D COMPREHENSIVE RESULTS ACROSS HYPERPARAMETER CONFIGURATIONS

This section presents the entire experimental results of NTE under various hyperparameter configurations, including different repeat counts and the use or omission of linear projection. All results are reported with a fixed kernel size of 3. * indicates the use of concatenation, while the absence of the marker denotes the use of feature combination.

Table 15: Performance of NTE (kernel size = 3, repeat = 1, without linear projection).

| Model | | FC | | LSTM | | Conv1D | | Dilated Conv1D | | Bi-Dilated Conv1D | | Bi-Dilated Conv1D* | |
|---|---|---|---|---|---|---|---|---|---|---|---|---|---|
| Metric | | MSE | MAE | MSE | MAE | MSE | MAE | MSE | MAE | MSE | MAE | MSE | MAE |
| ETTh1 | 96 | 0.392 | 0.410 | 0.386 | 0.405 | 0.402 | 0.416 | 0.383 | 0.403 | 0.380 | 0.402 | 0.391 | 0.410 |
| | 192 | 0.441 | 0.438 | 0.440 | 0.435 | 0.453 | 0.445 | 0.435 | 0.432 | 0.429 | 0.431 | 0.437 | 0.438 |
| | 336 | 0.482 | 0.457 | 0.489 | 0.459 | 0.497 | 0.466 | 0.481 | 0.456 | 0.475 | 0.453 | 0.484 | 0.463 |
| | 720 | 0.505 | 0.484 | 0.507 | 0.485 | 0.517 | 0.494 | 0.485 | 0.480 | 0.484 | 0.478 | 0.575 | 0.525 |
| | Avg | 0.455 | 0.447 | 0.456 | 0.446 | 0.467 | 0.455 | 0.446 | 0.443 | 0.442 | 0.441 | 0.472 | 0.459 |
| ETTh2 | 96 | 0.302 | 0.351 | 0.306 | 0.358 | 0.308 | 0.359 | 0.302 | 0.302 | 0.295 | 0.347 | 0.295 | 0.347 |
| | 192 | 0.385 | 0.402 | 0.389 | 0.407 | 0.392 | 0.408 | 0.383 | 0.383 | 0.378 | 0.399 | 0.383 | 0.402 |
| | 336 | 0.445 | 0.448 | 0.430 | 0.439 | 0.444 | 0.449 | 0.417 | 0.417 | 0.418 | 0.430 | 0.424 | 0.432 |
| | 720 | 0.442 | 0.454 | 0.435 | 0.452 | 0.449 | 0.459 | 0.431 | 0.431 | 0.422 | 0.443 | 0.430 | 0.446 |
| | Avg | 0.394 | 0.414 | 0.390 | 0.414 | 0.398 | 0.419 | 0.383 | 0.383 | 0.378 | 0.405 | 0.383 | 0.407 |
| ETTm1 | 96 | 0.329 | 0.369 | 0.333 | 0.369 | 0.358 | 0.387 | 0.336 | 0.336 | 0.329 | 0.365 | 0.333 | 0.368 |
| | 192 | 0.372 | 0.391 | 0.373 | 0.390 | 0.392 | 0.402 | 0.377 | 0.377 | 0.371 | 0.389 | 0.376 | 0.392 |
| | 336 | 0.404 | 0.412 | 0.407 | 0.411 | 0.424 | 0.422 | 0.412 | 0.412 | 0.406 | 0.411 | 0.413 | 0.418 |
| | 720 | 0.473 | 0.450 | 0.472 | 0.448 | 0.494 | 0.461 | 0.478 | 0.478 | 0.477 | 0.451 | 0.484 | 0.457 |
| | Avg | 0.395 | 0.406 | 0.396 | 0.405 | 0.417 | 0.418 | 0.401 | 0.401 | 0.396 | 0.404 | 0.402 | 0.409 |
| ETTm2 | 96 | 0.181 | 0.267 | 0.184 | 0.268 | 0.188 | 0.274 | 0.183 | 0.267 | 0.185 | 0.269 | 0.179 | 0.264 |
| | 192 | 0.245 | 0.306 | 0.262 | 0.318 | 0.258 | 0.317 | 0.251 | 0.311 | 0.254 | 0.312 | 0.251 | 0.310 |
| | 336 | 0.308 | 0.346 | 0.327 | 0.358 | 0.322 | 0.355 | 0.315 | 0.351 | 0.317 | 0.352 | 0.311 | 0.346 |
| | 720 | 0.414 | 0.407 | 0.411 | 0.409 | 0.414 | 0.408 | 0.409 | 0.403 | 0.411 | 0.406 | 0.404 | 0.401 |
| | Avg | 0.287 | 0.332 | 0.296 | 0.338 | 0.296 | 0.339 | 0.290 | 0.333 | 0.292 | 0.335 | 0.286 | 0.330 |
| ECL | 96 | 0.149 | 0.242 | 0.140 | 0.236 | 0.146 | 0.243 | 0.144 | 0.239 | 0.144 | 0.239 | 0.144 | 0.242 |
| | 192 | 0.163 | 0.256 | 0.157 | 0.250 | 0.161 | 0.256 | 0.140 | 0.254 | 0.159 | 0.253 | 0.161 | 0.256 |
| | 336 | 0.177 | 0.272 | 0.170 | 0.266 | 0.175 | 0.271 | 0.175 | 0.270 | 0.174 | 0.268 | 0.176 | 0.273 |
| | 720 | 0.211 | 0.302 | 0.201 | 0.295 | 0.210 | 0.303 | 0.229 | 0.316 | 0.230 | 0.316 | 0.207 | 0.302 |
| | Avg | 0.175 | 0.268 | 0.167 | 0.262 | 0.173 | 0.268 | 0.172 | 0.270 | 0.177 | 0.269 | 0.172 | 0.268 |
| Weather | 96 | 0.176 | 0.215 | 0.167 | 0.209 | 0.174 | 0.219 | 0.172 | 0.216 | 0.172 | 0.214 | 0.172 | 0.217 |
| | 192 | 0.222 | 0.256 | 0.220 | 0.257 | 0.222 | 0.259 | 0.222 | 0.260 | 0.215 | 0.256 | 0.219 | 0.259 |
| | 336 | 0.279 | 0.298 | 0.272 | 0.297 | 0.278 | 0.300 | 0.278 | 0.301 | 0.276 | 0.298 | 0.276 | 0.301 |
| | 720 | 0.359 | 0.350 | 0.352 | 0.349 | 0.354 | 0.349 | 0.358 | 0.354 | 0.352 | 0.349 | 0.355 | 0.352 |
| | Avg | 0.259 | 0.280 | 0.253 | 0.278 | 0.257 | 0.282 | 0.258 | 0.283 | 0.254 | 0.279 | 0.256 | 0.282 |
| Traffic | 96 | 0.402 | 0.275 | 0.398 | 0.272 | 0.410 | 0.282 | 0.524 | 0.367 | 0.524 | 0.374 | 1.342 | 0.756 |
| | 192 | 0.421 | 0.282 | 0.419 | 0.279 | 0.429 | 0.289 | 0.521 | 0.365 | 0.513 | 0.362 | 1.403 | 0.795 |
| | 336 | 0.434 | 0.288 | 0.435 | 0.287 | 0.442 | 0.295 | 0.534 | 0.371 | 0.535 | 0.368 | 1.377 | 0.745 |
| | 720 | 0.464 | 0.306 | 0.465 | 0.303 | 0.476 | 0.313 | 0.573 | 0.389 | 0.580 | 0.392 | 1.351 | 0.738 |
| | Avg | 0.430 | 0.288 | 0.429 | 0.285 | 0.439 | 0.295 | 0.538 | 0.373 | 0.538 | 0.374 | 1.368 | 0.759 |
| Solar Energy | 96 | 0.202 | 0.233 | 0.211 | 0.243 | 0.208 | 0.241 | 0.224 | 0.261 | 0.223 | 0.259 | 0.253 | 0.290 |
| | 192 | 0.238 | 0.262 | 0.242 | 0.267 | 0.241 | 0.266 | 0.255 | 0.280 | 0.266 | 0.292 | 0.294 | 0.315 |
| | 336 | 0.248 | 0.272 | 0.262 | 0.280 | 0.245 | 0.276 | 0.300 | 0.313 | 0.302 | 0.315 | 0.317 | 0.334 |
| | 720 | 0.252 | 0.276 | 0.258 | 0.279 | 0.259 | 0.280 | 0.279 | 0.300 | 0.302 | 0.315 | 0.301 | 0.317 |
| | Avg | 0.235 | 0.261 | 0.243 | 0.267 | 0.238 | 0.266 | 0.265 | 0.289 | 0.273 | 0.295 | 0.291 | 0.314 |

Table 16: Performance of NTE (kernel size = 3, repeat = 1, with linear projection).

| Model | | FC | | LSTM | | Conv1D | | Dilated Conv1D | | Bi-Dilated Conv1D | | Bi-Dilated Conv1D[*] | |
|---|---|---|---|---|---|---|---|---|---|---|---|---|---|
| Metric | | MSE | MAE | MSE | MAE | MSE | MAE | MSE | MAE | MSE | MAE | MSE | MAE |
| ETTh1 | 96 | 0.386 | 0.404 | 0.385 | 0.405 | 0.395 | 0.413 | 0.386 | 0.404 | 0.380 | 0.401 | 0.393 | 0.410 |
| | 192 | 0.438 | 0.432 | 0.435 | 0.434 | 0.447 | 0.443 | 0.439 | 0.435 | 0.434 | 0.434 | 0.442 | 0.440 |
| | 336 | 0.482 | 0.455 | 0.483 | 0.456 | 0.510 | 0.471 | 0.477 | 0.453 | 0.478 | 0.454 | 0.506 | 0.474 |
| | 720 | 0.492 | 0.479 | 0.495 | 0.481 | 0.503 | 0.491 | 0.479 | 0.475 | 0.490 | 0.483 | 0.496 | 0.482 |
| | Avg | 0.450 | 0.443 | 0.450 | 0.444 | 0.464 | 0.455 | 0.445 | 0.442 | 0.446 | 0.443 | 0.459 | 0.452 |
| ETTh2 | 96 | 0.298 | 0.349 | 0.304 | 0.354 | 0.306 | 0.358 | 0.302 | 0.350 | 0.299 | 0.349 | 0.307 | 0.353 |
| | 192 | 0.383 | 0.402 | 0.387 | 0.405 | 0.396 | 0.411 | 0.388 | 0.403 | 0.394 | 0.407 | 0.386 | 0.402 |
| | 336 | 0.423 | 0.434 | 0.423 | 0.434 | 0.440 | 0.444 | 0.422 | 0.432 | 0.419 | 0.431 | 0.429 | 0.436 |
| | 720 | 0.433 | 0.450 | 0.438 | 0.453 | 0.446 | 0.456 | 0.428 | 0.446 | 0.420 | 0.443 | 0.431 | 0.448 |
| | Avg | 0.384 | 0.409 | 0.388 | 0.412 | 0.397 | 0.417 | 0.385 | 0.408 | 0.383 | 0.408 | 0.388 | 0.410 |
| ETTm1 | 96 | 0.338 | 0.372 | 0.333 | 0.369 | 0.356 | 0.384 | 0.339 | 0.373 | 0.328 | 0.365 | 0.334 | 0.370 |
| | 192 | 0.378 | 0.391 | 0.373 | 0.389 | 0.390 | 0.402 | 0.376 | 0.392 | 0.371 | 0.388 | 0.376 | 0.393 |
| | 336 | 0.413 | 0.413 | 0.405 | 0.410 | 0.423 | 0.421 | 0.412 | 0.416 | 0.404 | 0.410 | 0.413 | 0.418 |
| | 720 | 0.480 | 0.449 | 0.472 | 0.449 | 0.492 | 0.460 | 0.479 | 0.454 | 0.478 | 0.451 | 0.461 | 0.490 |
| | Avg | 0.402 | 0.406 | 0.396 | 0.404 | 0.415 | 0.417 | 0.402 | 0.409 | 0.395 | 0.404 | 0.396 | 0.418 |
| ETTm2 | 96 | 0.180 | 0.265 | 0.185 | 0.267 | 0.188 | 0.271 | 0.183 | 0.268 | 0.184 | 0.269 | 0.181 | 0.264 |
| | 192 | 0.248 | 0.310 | 0.253 | 0.314 | 0.256 | 0.318 | 0.254 | 0.312 | 0.253 | 0.313 | 0.251 | 0.308 |
| | 336 | 0.310 | 0.349 | 0.314 | 0.352 | 0.323 | 0.358 | 0.315 | 0.351 | 0.318 | 0.352 | 0.309 | 0.346 |
| | 720 | 0.409 | 0.404 | 0.405 | 0.408 | 0.418 | 0.412 | 0.421 | 0.411 | 0.409 | 0.403 | 0.407 | 0.403 |
| | Avg | 0.287 | 0.332 | 0.289 | 0.335 | 0.296 | 0.340 | 0.293 | 0.336 | 0.291 | 0.334 | 0.287 | 0.330 |
| ECL | 96 | 0.148 | 0.241 | 0.140 | 0.236 | 0.146 | 0.241 | 0.143 | 0.238 | 0.142 | 0.238 | 0.145 | 0.242 |
| | 192 | 0.162 | 0.253 | 0.157 | 0.251 | 0.161 | 0.255 | 0.160 | 0.252 | 0.159 | 0.251 | 0.162 | 0.257 |
| | 336 | 0.179 | 0.272 | 0.172 | 0.268 | 0.179 | 0.274 | 0.174 | 0.269 | 0.174 | 0.270 | 0.177 | 0.275 |
| | 720 | 0.237 | 0.322 | 0.199 | 0.293 | 0.242 | 0.328 | 0.205 | 0.298 | 0.211 | 0.304 | 0.210 | 0.304 |
| | Avg | 0.182 | 0.272 | 0.167 | 0.262 | 0.182 | 0.275 | 0.171 | 0.264 | 0.172 | 0.266 | 0.174 | 0.270 |
| Weather | 96 | 0.174 | 0.214 | 0.168 | 0.211 | 0.172 | 0.215 | 0.176 | 0.220 | 0.168 | 0.212 | 0.171 | 0.215 |
| | 192 | 0.224 | 0.258 | 0.216 | 0.255 | 0.219 | 0.257 | 0.214 | 0.255 | 0.217 | 0.257 | 0.219 | 0.258 |
| | 336 | 0.280 | 0.298 | 0.272 | 0.296 | 0.276 | 0.299 | 0.276 | 0.301 | 0.269 | 0.296 | 0.274 | 0.298 |
| | 720 | 0.357 | 0.349 | 0.355 | 0.350 | 0.357 | 0.351 | 0.354 | 0.350 | 0.349 | 0.347 | 0.357 | 0.353 |
| | Avg | 0.259 | 0.280 | 0.253 | 0.278 | 0.256 | 0.281 | 0.255 | 0.282 | 0.251 | 0.278 | 0.255 | 0.281 |
| Traffic | 96 | 0.398 | 0.272 | 0.404 | 0.275 | 0.408 | 0.281 | 1.268 | 0.715 | 1.181 | 0.659 | 1.407 | 0.798 |
| | 192 | 0.421 | 0.283 | 0.423 | 0.283 | 0.427 | 0.287 | 1.174 | 0.646 | 1.317 | 0.707 | 1.410 | 0.774 |
| | 336 | 0.433 | 0.289 | 0.436 | 0.288 | 0.440 | 0.293 | 1.150 | 0.646 | 1.117 | 0.631 | 1.630 | 0.856 |
| | 720 | 0.464 | 0.305 | 0.466 | 0.304 | 0.473 | 0.312 | 1.299 | 0.721 | 1.082 | 0.601 | 1.378 | 0.752 |
| | Avg | 0.429 | 0.287 | 0.432 | 0.288 | 0.437 | 0.293 | 1.223 | 0.682 | 1.174 | 0.650 | 1.456 | 0.795 |
| Solar Energy | 96 | 0.205 | 0.235 | 0.208 | 0.234 | 0.211 | 0.240 | 0.251 | 0.285 | 0.243 | 0.278 | 0.528 | 0.472 |
| | 192 | 0.236 | 0.260 | 0.262 | 0.279 | 0.246 | 0.267 | 0.284 | 0.302 | 0.294 | 0.312 | 0.541 | 0.469 |
| | 336 | 0.246 | 0.270 | 0.262 | 0.279 | 0.257 | 0.279 | 0.339 | 0.332 | 0.314 | 0.323 | 0.691 | 0.539 |
| | 720 | 0.250 | 0.276 | 0.261 | 0.282 | 0.256 | 0.279 | 0.304 | 0.317 | 0.365 | 0.356 | 0.618 | 0.515 |
| | Avg | 0.234 | 0.260 | 0.248 | 0.269 | 0.243 | 0.266 | 0.295 | 0.309 | 0.304 | 0.317 | 0.595 | 0.499 |

Table 17: Performance of NTE (kernel size = 3, repeat = 2, without linear projection).

| Model | Metric | FC | | LSTM | | Conv1D | | Dilated Conv1D | | Bi-Dilated Conv1D | | Bi-Dilated Conv1D* | |
|---|---|---|---|---|---|---|---|---|---|---|---|---|---|
| | | MSE | MAE | MSE | MAE | MSE | MAE | MSE | MAE | MSE | MAE | MSE | MAE |
| ETTh1 | 96 | 0.390 | 0.409 | 0.388 | 0.407 | 0.399 | 0.416 | 0.391 | 0.408 | 0.381 | 0.403 | 0.397 | 0.414 |
| | 192 | 0.442 | 0.438 | 0.442 | 0.437 | 0.452 | 0.446 | 0.444 | 0.439 | 0.432 | 0.432 | 0.447 | 0.444 |
| | 336 | 0.478 | 0.456 | 0.482 | 0.455 | 0.504 | 0.469 | 0.485 | 0.459 | 0.476 | 0.455 | 0.493 | 0.465 |
| | 720 | 0.504 | 0.484 | 0.499 | 0.482 | 0.502 | 0.487 | 0.511 | 0.495 | 0.493 | 0.483 | 0.504 | 0.488 |
| | Avg | 0.454 | 0.447 | 0.453 | 0.445 | 0.464 | 0.455 | 0.458 | 0.450 | 0.446 | 0.443 | 0.460 | 0.453 |
| ETTh2 | 96 | 0.302 | 0.351 | 0.306 | 0.359 | 0.306 | 0.357 | 0.311 | 0.355 | 0.296 | 0.348 | 0.308 | 0.355 |
| | 192 | 0.394 | 0.409 | 0.394 | 0.413 | 0.398 | 0.414 | 0.391 | 0.403 | 0.377 | 0.398 | 0.399 | 0.411 |
| | 336 | 0.450 | 0.450 | 0.431 | 0.441 | 0.435 | 0.442 | 0.435 | 0.438 | 0.425 | 0.434 | 0.428 | 0.434 |
| | 720 | 0.450 | 0.461 | 0.434 | 0.451 | 0.449 | 0.460 | 0.440 | 0.452 | 0.424 | 0.444 | 0.426 | 0.445 |
| | Avg | 0.399 | 0.418 | 0.391 | 0.416 | 0.397 | 0.418 | 0.394 | 0.412 | 0.381 | 0.406 | 0.390 | 0.411 |
| ETTm1 | 96 | 0.331 | 0.371 | 0.333 | 0.370 | 0.352 | 0.383 | 0.336 | 0.371 | 0.330 | 0.366 | 0.340 | 0.373 |
| | 192 | 0.370 | 0.390 | 0.373 | 0.391 | 0.385 | 0.400 | 0.378 | 0.395 | 0.373 | 0.390 | 0.377 | 0.395 |
| | 336 | 0.402 | 0.411 | 0.406 | 0.412 | 0.420 | 0.420 | 0.416 | 0.418 | 0.408 | 0.412 | 0.423 | 0.423 |
| | 720 | 0.476 | 0.452 | 0.476 | 0.451 | 0.489 | 0.460 | 0.484 | 0.456 | 0.487 | 0.455 | 0.492 | 0.462 |
| | Avg | 0.395 | 0.406 | 0.397 | 0.406 | 0.412 | 0.416 | 0.404 | 0.410 | 0.400 | 0.406 | 0.408 | 0.413 |
| ETTm2 | 96 | 0.179 | 0.264 | 0.186 | 0.271 | 0.185 | 0.270 | 0.187 | 0.271 | 0.187 | 0.269 | 0.183 | 0.267 |
| | 192 | 0.243 | 0.305 | 0.264 | 0.320 | 0.253 | 0.312 | 0.251 | 0.312 | 0.259 | 0.315 | 0.249 | 0.308 |
| | 336 | 0.308 | 0.347 | 0.321 | 0.356 | 0.315 | 0.351 | 0.313 | 0.350 | 0.319 | 0.352 | 0.314 | 0.348 |
| | 720 | 0.408 | 0.404 | 0.413 | 0.407 | 0.413 | 0.407 | 0.414 | 0.407 | 0.417 | 0.408 | 0.413 | 0.406 |
| | Avg | 0.285 | 0.330 | 0.296 | 0.339 | 0.292 | 0.335 | 0.291 | 0.335 | 0.296 | 0.336 | 0.290 | 0.332 |
| ECL | 96 | 0.148 | 0.242 | 0.140 | 0.237 | 0.144 | 0.242 | 0.142 | 0.238 | 0.142 | 0.237 | 0.383 | 0.432 |
| | 192 | 0.162 | 0.255 | 0.159 | 0.252 | 0.161 | 0.257 | 0.160 | 0.254 | 0.159 | 0.252 | 0.370 | 0.426 |
| | 336 | 0.179 | 0.276 | 0.171 | 0.268 | 0.177 | 0.275 | 0.177 | 0.274 | 0.212 | 0.307 | 0.395 | 0.446 |
| | 720 | 0.240 | 0.325 | 0.201 | 0.294 | 0.243 | 0.331 | 0.322 | 0.392 | 0.289 | 0.370 | 0.460 | 0.486 |
| | Avg | 0.182 | 0.275 | 0.168 | 0.263 | 0.181 | 0.276 | 0.200 | 0.290 | 0.201 | 0.292 | 0.402 | 0.448 |
| Weather | 96 | 0.178 | 0.218 | 0.169 | 0.211 | 0.169 | 0.212 | 0.173 | 0.220 | 0.163 | 0.209 | 0.173 | 0.218 |
| | 192 | 0.221 | 0.257 | 0.219 | 0.256 | 0.217 | 0.256 | 0.215 | 0.257 | 0.214 | 0.255 | 0.220 | 0.261 |
| | 336 | 0.278 | 0.297 | 0.271 | 0.295 | 0.277 | 0.300 | 0.275 | 0.300 | 0.273 | 0.299 | 0.283 | 0.306 |
| | 720 | 0.357 | 0.349 | 0.355 | 0.350 | 0.353 | 0.350 | 0.355 | 0.351 | 0.353 | 0.350 | 0.358 | 0.354 |
| | Avg | 0.259 | 0.280 | 0.254 | 0.278 | 0.254 | 0.280 | 0.255 | 0.282 | 0.251 | 0.278 | 0.259 | 0.285 |
| Traffic | 96 | 0.402 | 0.275 | 0.413 | 0.280 | 0.420 | 0.288 | 1.235 | 0.702 | 1.276 | 0.718 | 1.339 | 0.721 |
| | 192 | 0.422 | 0.284 | 0.427 | 0.284 | 0.434 | 0.293 | 1.411 | 0.803 | 1.088 | 0.624 | 1.269 | 0.714 |
| | 336 | 0.434 | 0.290 | 0.442 | 0.291 | 0.449 | 0.299 | 1.458 | 0.791 | 1.506 | 0.825 | 1.428 | 0.809 |
| | 720 | 0.468 | 0.308 | 0.473 | 0.309 | 0.483 | 0.318 | 1.366 | 0.742 | 1.328 | 0.730 | 1.450 | 0.809 |
| | Avg | 0.432 | 0.289 | 0.439 | 0.291 | 0.447 | 0.300 | 1.368 | 0.760 | 1.300 | 0.724 | 1.372 | 0.763 |
| Solar Energy | 96 | 0.204 | 0.229 | 0.207 | 0.248 | 0.210 | 0.245 | 0.284 | 0.313 | 0.297 | 0.322 | 0.417 | 0.405 |
| | 192 | 0.233 | 0.255 | 0.245 | 0.271 | 0.248 | 0.270 | 0.326 | 0.339 | 0.307 | 0.312 | 0.482 | 0.430 |
| | 336 | 0.248 | 0.273 | 0.265 | 0.284 | 0.259 | 0.280 | 0.361 | 0.344 | 0.364 | 0.352 | 0.560 | 0.484 |
| | 720 | 0.250 | 0.277 | 0.260 | 0.282 | 0.257 | 0.280 | 0.363 | 0.342 | 0.365 | 0.354 | 0.598 | 0.508 |
| | Avg | 0.234 | 0.259 | 0.244 | 0.271 | 0.244 | 0.269 | 0.334 | 0.335 | 0.333 | 0.335 | 0.514 | 0.457 |

Table 18: Performance of NTE (kernel size = 3, repeat = 2, with linear projection).

| Model | | FC | | LSTM | | Conv1D | | Dilated Conv1D | | Bi-Dilated Conv1D | | Bi-Dilated Conv1D* | |
|---|---|---|---|---|---|---|---|---|---|---|---|---|---|
| Metric | | MSE | MAE | MSE | MAE | MSE | MAE | MSE | MAE | MSE | MAE | MSE | MAE |
| ETTh1 | 96 | 0.387 | 0.406 | 0.387 | 0.407 | 0.394 | 0.414 | 0.391 | 0.408 | 0.392 | 0.410 | 0.402 | 0.416 |
| | 192 | 0.438 | 0.434 | 0.439 | 0.437 | 0.449 | 0.444 | 0.436 | 0.435 | 0.444 | 0.440 | 0.459 | 0.449 |
| | 336 | 0.481 | 0.456 | 0.482 | 0.461 | 0.491 | 0.467 | 0.486 | 0.461 | 0.483 | 0.459 | 0.522 | 0.483 |
| | 720 | 0.489 | 0.478 | 0.498 | 0.486 | 0.502 | 0.488 | 0.504 | 0.493 | 0.496 | 0.485 | 0.524 | 0.500 |
| | Avg | 0.449 | 0.444 | 0.452 | 0.448 | 0.459 | 0.453 | 0.454 | 0.449 | 0.454 | 0.449 | 0.477 | 0.462 |
| ETTh2 | 96 | 0.299 | 0.350 | 0.301 | 0.354 | 0.302 | 0.353 | 0.302 | 0.353 | 0.187 | 0.269 | 0.321 | 0.365 |
| | 192 | 0.373 | 0.400 | 0.386 | 0.405 | 0.385 | 0.403 | 0.398 | 0.408 | 0.254 | 0.316 | 0.394 | 0.407 |
| | 336 | 0.423 | 0.433 | 0.429 | 0.441 | 0.440 | 0.446 | 0.442 | 0.444 | 0.423 | 0.431 | 0.423 | 0.432 |
| | 720 | 0.433 | 0.450 | 0.435 | 0.452 | 0.453 | 0.463 | 0.432 | 0.449 | 0.424 | 0.443 | 0.440 | 0.453 |
| | Avg | 0.382 | 0.408 | 0.388 | 0.413 | 0.395 | 0.416 | 0.394 | 0.414 | 0.322 | 0.365 | 0.395 | 0.414 |
| ETTm1 | 96 | 0.343 | 0.375 | 0.332 | 0.368 | 0.347 | 0.380 | 0.336 | 0.371 | 0.337 | 0.372 | 0.338 | 0.372 |
| | 192 | 0.378 | 0.391 | 0.375 | 0.391 | 0.382 | 0.398 | 0.380 | 0.396 | 0.378 | 0.394 | 0.379 | 0.396 |
| | 336 | 0.413 | 0.413 | 0.412 | 0.415 | 0.416 | 0.419 | 0.418 | 0.420 | 0.410 | 0.415 | 0.419 | 0.420 |
| | 720 | 0.480 | 0.451 | 0.481 | 0.453 | 0.489 | 0.459 | 0.489 | 0.460 | 0.483 | 0.456 | 0.501 | 0.466 |
| | Avg | 0.404 | 0.408 | 0.400 | 0.407 | 0.409 | 0.414 | 0.406 | 0.412 | 0.402 | 0.409 | 0.409 | 0.414 |
| ETTm2 | 96 | 0.182 | 0.264 | 0.187 | 0.270 | 0.184 | 0.268 | 0.186 | 0.271 | 0.187 | 0.269 | 0.184 | 0.267 |
| | 192 | 0.245 | 0.306 | 0.262 | 0.317 | 0.254 | 0.312 | 0.254 | 0.314 | 0.256 | 0.314 | 0.252 | 0.310 |
| | 336 | 0.304 | 0.345 | 0.321 | 0.355 | 0.317 | 0.351 | 0.314 | 0.351 | 0.314 | 0.351 | 0.318 | 0.350 |
| | 720 | 0.404 | 0.400 | 0.410 | 0.406 | 0.412 | 0.406 | 0.417 | 0.408 | 0.415 | 0.407 | 0.411 | 0.404 |
| | Avg | 0.284 | 0.329 | 0.295 | 0.337 | 0.292 | 0.334 | 0.293 | 0.336 | 0.293 | 0.335 | 0.291 | 0.333 |
| ECL | 96 | 0.148 | 0.242 | 0.141 | 0.237 | 0.144 | 0.242 | 0.142 | 0.237 | 0.181 | 0.278 | 0.471 | 0.499 |
| | 192 | 0.162 | 0.255 | 0.157 | 0.250 | 0.164 | 0.260 | 0.186 | 0.281 | 0.191 | 0.289 | 0.547 | 0.546 |
| | 336 | 0.178 | 0.274 | 0.170 | 0.267 | 0.177 | 0.276 | 0.293 | 0.378 | 0.318 | 0.393 | 0.568 | 0.557 |
| | 720 | 0.221 | 0.310 | 0.203 | 0.298 | 0.216 | 0.309 | 0.461 | 0.490 | 0.561 | 0.551 | 0.609 | 0.581 |
| | Avg | 0.177 | 0.270 | 0.168 | 0.263 | 0.175 | 0.272 | 0.271 | 0.347 | 0.313 | 0.378 | 0.549 | 0.546 |
| Weather | 96 | 0.172 | 0.212 | 0.168 | 0.210 | 0.168 | 0.212 | 0.168 | 0.215 | 0.165 | 0.211 | 0.170 | 0.216 |
| | 192 | 0.222 | 0.256 | 0.217 | 0.255 | 0.217 | 0.255 | 0.217 | 0.258 | 0.219 | 0.260 | 0.216 | 0.256 |
| | 336 | 0.278 | 0.297 | 0.271 | 0.295 | 0.273 | 0.298 | 0.278 | 0.303 | 0.279 | 0.302 | 0.277 | 0.301 |
| | 720 | 0.356 | 0.349 | 0.354 | 0.350 | 0.356 | 0.352 | 0.356 | 0.351 | 0.362 | 0.354 | 0.358 | 0.351 |
| | Avg | 0.257 | 0.279 | 0.253 | 0.278 | 0.254 | 0.279 | 0.255 | 0.282 | 0.256 | 0.282 | 0.255 | 0.281 |
| Traffic | 96 | 0.405 | 0.278 | 0.413 | 0.282 | 0.555 | 0.376 | 1.306 | 0.727 | 1.282 | 0.721 | 1.365 | 0.762 |
| | 192 | 0.425 | 0.286 | 0.425 | 0.283 | 0.442 | 0.298 | 1.343 | 0.748 | 1.412 | 0.803 | 1.434 | 0.811 |
| | 336 | 0.436 | 0.292 | 0.440 | 0.291 | 0.457 | 0.305 | 1.300 | 0.734 | 1.276 | 0.712 | 1.369 | 0.771 |
| | 720 | 0.474 | 0.309 | 0.472 | 0.308 | 0.493 | 0.324 | 1.282 | 0.719 | 1.284 | 0.718 | 1.450 | 0.810 |
| | Avg | 0.435 | 0.291 | 0.438 | 0.291 | 0.487 | 0.326 | 1.308 | 0.732 | 1.314 | 0.739 | 1.405 | 0.789 |
| Solar Energy | 96 | 0.206 | 0.233 | 0.222 | 0.238 | 0.218 | 0.251 | 0.284 | 0.320 | 0.287 | 0.323 | 0.554 | 0.479 |
| | 192 | 0.239 | 0.261 | 0.252 | 0.271 | 0.254 | 0.276 | 0.347 | 0.350 | 0.321 | 0.338 | 0.479 | 0.437 |
| | 336 | 0.250 | 0.274 | 0.264 | 0.283 | 0.273 | 0.290 | 0.423 | 0.389 | 0.364 | 0.359 | 0.434 | 0.407 |
| | 720 | 0.249 | 0.276 | 0.266 | 0.285 | 0.283 | 0.298 | 0.467 | 0.415 | 0.368 | 0.352 | 0.544 | 0.467 |
| | Avg | 0.236 | 0.261 | 0.251 | 0.269 | 0.257 | 0.279 | 0.380 | 0.369 | 0.335 | 0.343 | 0.503 | 0.448 |

Table 19: Performance of NTE (kernel size = 3, repeat = 3, without linear projection).

| Model | | FC | | LSTM | | Conv1D | | Dilated Conv1D | | Bi-Dilated Conv1D | | Bi-Dilated Conv1D[*] | |
|---|---|---|---|---|---|---|---|---|---|---|---|---|---|---|
| Metric | | MSE | MAE | MSE | MAE | MSE | MAE | MSE | MAE | MSE | MAE | MSE | MAE |
| ETTh1 | 96 | 0.391 | 0.411 | 0.401 | 0.416 | 0.400 | 0.418 | 0.394 | 0.410 | 0.385 | 0.404 | 0.403 | 0.416 |
| | 192 | 0.443 | 0.439 | 0.454 | 0.446 | 0.454 | 0.448 | 0.445 | 0.441 | 0.443 | 0.439 | 0.456 | 0.450 |
| | 336 | 0.470 | 0.454 | 0.476 | 0.458 | 0.496 | 0.467 | 0.484 | 0.461 | 0.482 | 0.457 | 0.515 | 0.480 |
| | 720 | 0.496 | 0.480 | 0.545 | 0.513 | 0.515 | 0.493 | 0.494 | 0.486 | 0.497 | 0.486 | 0.518 | 0.494 |
| | Avg | 0.450 | 0.446 | 0.469 | 0.458 | 0.466 | 0.457 | 0.454 | 0.450 | 0.452 | 0.447 | 0.473 | 0.460 |
| ETTh2 | 96 | 0.299 | 0.350 | 0.316 | 0.366 | 0.306 | 0.357 | 0.312 | 0.357 | 0.304 | 0.353 | 0.317 | 0.363 |
| | 192 | 0.381 | 0.401 | 0.395 | 0.412 | 0.386 | 0.407 | 0.386 | 0.401 | 0.374 | 0.399 | 0.400 | 0.411 |
| | 336 | 0.436 | 0.440 | 0.433 | 0.445 | 0.423 | 0.435 | 0.432 | 0.437 | 0.433 | 0.438 | 0.435 | 0.436 |
| | 720 | 0.455 | 0.464 | 0.447 | 0.460 | 0.448 | 0.460 | 0.447 | 0.456 | 0.427 | 0.447 | 0.446 | 0.452 |
| | Avg | 0.393 | 0.414 | 0.398 | 0.421 | 0.391 | 0.415 | 0.394 | 0.413 | 0.385 | 0.409 | 0.400 | 0.416 |
| ETTm1 | 96 | 0.336 | 0.375 | 0.338 | 0.373 | 0.352 | 0.383 | 0.336 | 0.371 | 0.330 | 0.366 | 0.337 | 0.371 |
| | 192 | 0.374 | 0.394 | 0.378 | 0.395 | 0.383 | 0.400 | 0.378 | 0.395 | 0.371 | 0.390 | 0.382 | 0.399 |
| | 336 | 0.406 | 0.415 | 0.413 | 0.417 | 0.418 | 0.421 | 0.418 | 0.420 | 0.405 | 0.420 | 0.428 | 0.426 |
| | 720 | 0.473 | 0.452 | 0.484 | 0.456 | 0.488 | 0.460 | 0.486 | 0.458 | 0.485 | 0.455 | 0.500 | 0.465 |
| | Avg | 0.397 | 0.409 | 0.403 | 0.410 | 0.410 | 0.416 | 0.405 | 0.411 | 0.398 | 0.408 | 0.412 | 0.415 |
| ETTm2 | 96 | 0.180 | 0.266 | 0.195 | 0.281 | 0.186 | 0.271 | 0.188 | 0.270 | 0.190 | 0.272 | 0.184 | 0.267 |
| | 192 | 0.243 | 0.304 | 0.263 | 0.323 | 0.254 | 0.315 | 0.254 | 0.315 | 0.258 | 0.314 | 0.254 | 0.312 |
| | 336 | 0.304 | 0.343 | 0.326 | 0.360 | 0.314 | 0.351 | 0.317 | 0.354 | 0.316 | 0.350 | 0.311 | 0.346 |
| | 720 | 0.407 | 0.403 | 0.419 | 0.412 | 0.413 | 0.406 | 0.414 | 0.407 | 0.415 | 0.408 | 0.413 | 0.406 |
| | Avg | 0.284 | 0.329 | 0.301 | 0.344 | 0.292 | 0.336 | 0.293 | 0.337 | 0.295 | 0.336 | 0.291 | 0.333 |
| ECL | 96 | 0.150 | 0.244 | 0.141 | 0.238 | 0.145 | 0.245 | 0.484 | 0.506 | 0.355 | 0.410 | 0.547 | 0.550 |
| | 192 | 0.163 | 0.257 | 0.158 | 0.253 | 0.162 | 0.260 | 0.577 | 0.570 | 0.585 | 0.576 | 0.492 | 0.515 |
| | 336 | 0.179 | 0.276 | 0.174 | 0.270 | 0.180 | 0.279 | 0.408 | 0.451 | 0.446 | 0.478 | 0.472 | 0.498 |
| | 720 | 0.214 | 0.307 | 0.203 | 0.298 | 0.209 | 0.304 | 0.519 | 0.528 | 0.469 | 0.493 | 0.716 | 0.638 |
| | Avg | 0.177 | 0.271 | 0.169 | 0.265 | 0.174 | 0.272 | 0.497 | 0.514 | 0.464 | 0.489 | 0.557 | 0.550 |
| Weather | 96 | 0.173 | 0.213 | 0.171 | 0.215 | 0.172 | 0.216 | 0.172 | 0.221 | 0.167 | 0.211 | 0.509 | 0.523 |
| | 192 | 0.220 | 0.255 | 0.214 | 0.254 | 0.220 | 0.260 | 0.232 | 0.271 | 0.227 | 0.268 | 0.221 | 0.261 |
| | 336 | 0.279 | 0.298 | 0.274 | 0.299 | 0.275 | 0.301 | 0.284 | 0.306 | 0.287 | 0.308 | 0.282 | 0.304 |
| | 720 | 0.356 | 0.349 | 0.354 | 0.349 | 0.358 | 0.354 | 0.364 | 0.355 | 0.361 | 0.354 | 0.368 | 0.358 |
| | Avg | 0.257 | 0.279 | 0.253 | 0.279 | 0.256 | 0.283 | 0.263 | 0.288 | 0.261 | 0.285 | 0.345 | 0.362 |
| Traffic | 96 | 0.406 | 0.279 | 1.282 | 0.717 | 0.612 | 0.414 | 1.175 | 0.661 | 1.409 | 0.800 | 1.244 | 0.706 |
| | 192 | 0.427 | 0.288 | 0.437 | 0.292 | 0.457 | 0.308 | 1.349 | 0.741 | 1.319 | 0.743 | 1.354 | 0.775 |
| | 336 | 0.438 | 0.294 | 0.453 | 0.300 | 0.474 | 0.315 | 1.277 | 0.715 | 1.343 | 0.741 | 1.293 | 0.726 |
| | 720 | 0.473 | 0.312 | 0.485 | 0.317 | 0.513 | 0.335 | 1.450 | 0.809 | 1.241 | 0.688 | 1.450 | 0.810 |
| | Avg | 0.436 | 0.293 | 0.664 | 0.407 | 0.514 | 0.343 | 1.313 | 0.732 | 1.328 | 0.743 | 1.335 | 0.754 |
| Solar Energy | 96 | 0.208 | 0.240 | 0.212 | 0.248 | 0.213 | 0.253 | 0.482 | 0.436 | 0.514 | 0.451 | 0.580 | 0.497 |
| | 192 | 0.233 | 0.258 | 0.252 | 0.273 | 0.263 | 0.285 | 0.353 | 0.360 | 0.539 | 0.486 | 0.543 | 0.469 |
| | 336 | 0.254 | 0.279 | 0.269 | 0.287 | 0.275 | 0.291 | 0.427 | 0.397 | 0.473 | 0.419 | 0.543 | 0.469 |
| | 720 | 0.256 | 0.282 | 0.263 | 0.285 | 0.268 | 0.288 | 0.450 | 0.418 | 0.508 | 0.438 | 0.430 | 0.399 |
| | Avg | 0.238 | 0.265 | 0.249 | 0.273 | 0.255 | 0.279 | 0.428 | 0.403 | 0.509 | 0.449 | 0.524 | 0.459 |

Table 20: Performance of NTE (kernel size = 3, repeat = 3, with linear projection).

| Model | | FC | | LSTM | | Conv1D | | Dilated Conv1D | | Bi-Dilated Conv1D | | Bi-Dilated Conv1D* | |
|---|---|---|---|---|---|---|---|---|---|---|---|---|---|---|
| Metric | | MSE | MAE | MSE | MAE | MSE | MAE | MSE | MAE | MSE | MAE | MSE | MAE |
| ETTh1 | 96 | 0.388 | 0.407 | 0.393 | 0.411 | 0.400 | 0.417 | 0.390 | 0.408 | 0.398 | 0.414 | 0.398 | 0.415 |
| | 192 | 0.443 | 0.438 | 0.450 | 0.442 | 0.455 | 0.449 | 0.449 | 0.444 | 0.455 | 0.448 | 0.464 | 0.453 |
| | 336 | 0.492 | 0.462 | 0.484 | 0.457 | 0.521 | 0.474 | 0.497 | 0.468 | 0.493 | 0.465 | 0.528 | 0.486 |
| | 720 | 0.495 | 0.479 | 0.512 | 0.494 | 0.520 | 0.494 | 0.500 | 0.487 | 0.505 | 0.491 | 0.522 | 0.495 |
| | Avg | 0.455 | 0.447 | 0.460 | 0.451 | 0.474 | 0.459 | 0.459 | 0.452 | 0.463 | 0.455 | 0.478 | 0.462 |
| ETTh2 | 96 | 0.299 | 0.349 | 0.317 | 0.367 | 0.308 | 0.359 | 0.319 | 0.361 | 0.319 | 0.361 | 0.316 | 0.361 |
| | 192 | 0.374 | 0.397 | 0.397 | 0.412 | 0.382 | 0.404 | 0.405 | 0.413 | 0.392 | 0.405 | 0.414 | 0.419 |
| | 336 | 0.425 | 0.437 | 0.434 | 0.445 | 0.427 | 0.437 | 0.430 | 0.435 | 0.419 | 0.429 | 0.417 | 0.434 |
| | 720 | 0.432 | 0.449 | 0.444 | 0.459 | 0.460 | 0.467 | 0.433 | 0.449 | 0.429 | 0.445 | 0.455 | 0.461 |
| | Avg | 0.383 | 0.408 | 0.398 | 0.421 | 0.394 | 0.417 | 0.397 | 0.415 | 0.390 | 0.410 | 0.401 | 0.419 |
| ETTm1 | 96 | 0.350 | 0.380 | 0.336 | 0.372 | 0.348 | 0.381 | 0.347 | 0.381 | 0.337 | 0.372 | 0.337 | 0.372 |
| | 192 | 0.377 | 0.392 | 0.379 | 0.395 | 0.382 | 0.400 | 0.382 | 0.400 | 0.380 | 0.396 | 0.383 | 0.399 |
| | 336 | 0.411 | 0.413 | 0.420 | 0.420 | 0.417 | 0.421 | 0.417 | 0.421 | 0.417 | 0.419 | 0.432 | 0.428 |
| | 720 | 0.481 | 0.452 | 0.497 | 0.462 | 0.492 | 0.462 | 0.492 | 0.462 | 0.484 | 0.457 | 0.506 | 0.469 |
| | Avg | 0.405 | 0.409 | 0.408 | 0.412 | 0.410 | 0.416 | 0.410 | 0.416 | 0.405 | 0.411 | 0.415 | 0.417 |
| ETTm2 | 96 | 0.182 | 0.268 | 0.192 | 0.278 | 0.187 | 0.271 | 0.186 | 0.271 | 0.187 | 0.269 | 0.187 | 0.270 |
| | 192 | 0.247 | 0.309 | 0.270 | 0.327 | 0.256 | 0.315 | 0.253 | 0.313 | 0.254 | 0.316 | 0.255 | 0.313 |
| | 336 | 0.305 | 0.345 | 0.320 | 0.356 | 0.318 | 0.352 | 0.320 | 0.354 | 0.315 | 0.352 | 0.315 | 0.348 |
| | 720 | 0.405 | 0.403 | 0.420 | 0.412 | 0.417 | 0.409 | 0.416 | 0.408 | 0.420 | 0.411 | 0.412 | 0.405 |
| | Avg | 0.285 | 0.331 | 0.301 | 0.343 | 0.295 | 0.337 | 0.294 | 0.337 | 0.294 | 0.337 | 0.292 | 0.334 |
| ECL | 96 | 0.149 | 0.244 | 0.140 | 0.237 | 0.147 | 0.247 | 0.509 | 0.523 | 0.484 | 0.509 | 0.627 | 0.612 |
| | 192 | 0.164 | 0.258 | 0.159 | 0.254 | 0.164 | 0.262 | 0.609 | 0.601 | 0.545 | 0.550 | 0.541 | 0.540 |
| | 336 | 0.179 | 0.276 | 0.171 | 0.269 | 0.179 | 0.278 | 0.519 | 0.533 | 0.436 | 0.473 | 0.536 | 0.542 |
| | 720 | 0.215 | 0.307 | 0.200 | 0.296 | 0.209 | 0.304 | 0.548 | 0.546 | 0.724 | 0.669 | 0.687 | 0.638 |
| | Avg | 0.177 | 0.271 | 0.168 | 0.264 | 0.175 | 0.273 | 0.546 | 0.551 | 0.547 | 0.550 | 0.598 | 0.583 |
| Weather | 96 | 0.173 | 0.213 | 0.170 | 0.211 | 0.169 | 0.216 | 0.178 | 0.224 | 0.183 | 0.229 | 0.176 | 0.224 |
| | 192 | 0.220 | 0.254 | 0.217 | 0.257 | 0.216 | 0.257 | 0.235 | 0.273 | 0.233 | 0.271 | 0.225 | 0.264 |
| | 336 | 0.279 | 0.298 | 0.274 | 0.298 | 0.275 | 0.301 | 0.292 | 0.312 | 0.288 | 0.308 | 0.300 | 0.318 |
| | 720 | 0.357 | 0.349 | 0.356 | 0.352 | 0.357 | 0.354 | 0.364 | 0.356 | 0.362 | 0.355 | 0.369 | 0.359 |
| | Avg | 0.257 | 0.279 | 0.254 | 0.280 | 0.254 | 0.282 | 0.267 | 0.291 | 0.267 | 0.291 | 0.268 | 0.291 |
| Traffic | 96 | 0.204 | 0.233 | 0.223 | 0.255 | 0.263 | 0.297 | 0.678 | 0.541 | 0.590 | 0.526 | 0.590 | 0.526 |
| | 192 | 0.238 | 0.263 | 0.244 | 0.273 | 0.299 | 0.317 | 0.552 | 0.476 | 0.964 | 0.587 | 0.718 | 0.581 |
| | 336 | 0.251 | 0.275 | 0.281 | 0.293 | 0.333 | 0.338 | 0.548 | 0.446 | 0.601 | 0.495 | 0.601 | 0.495 |
| | 720 | 0.252 | 0.279 | 0.270 | 0.288 | 0.333 | 0.337 | 0.398 | 0.375 | 0.693 | 0.547 | 0.693 | 0.547 |
| | Avg | 0.236 | 0.263 | 0.255 | 0.277 | 0.307 | 0.322 | 0.544 | 0.460 | 0.712 | 0.539 | 0.651 | 0.537 |
| Solar Energy | 96 | 0.410 | 0.283 | 1.352 | 0.757 | 1.278 | 0.716 | 1.277 | 0.718 | 1.268 | 0.720 | 1.409 | 0.801 |
| | 192 | 0.430 | 0.291 | 0.442 | 0.296 | 1.236 | 0.710 | 1.412 | 0.804 | 1.300 | 0.732 | 1.412 | 0.803 |
| | 336 | 0.444 | 0.298 | 0.459 | 0.304 | 1.083 | 0.598 | 1.367 | 0.767 | 1.300 | 0.727 | 1.337 | 0.754 |
| | 720 | 0.479 | 0.316 | 0.495 | 0.324 | 0.864 | 0.521 | 1.346 | 0.731 | 1.450 | 0.810 | 1.450 | 0.809 |
| | Avg | 0.441 | 0.297 | 0.687 | 0.420 | 1.115 | 0.636 | 1.351 | 0.755 | 1.330 | 0.747 | 1.402 | 0.792 |

Table 21: Performance of NTE (kernel size = 5, repeat = 1, without linear projection).

| Model | | Conv1D | | Dilated Conv1D | | Bi-Dilated Conv1D | | Bi-Dilated Conv1D[*] | |
|---|---|---|---|---|---|---|---|---|---|
| Metric | | MSE | MAE | MSE | MAE | MSE | MAE | MSE | MAE |
| ETTh1 | 96 | 0.400 | 0.417 | 0.389 | 0.406 | 0.378 | 0.400 | 0.392 | 0.410 |
| | 192 | 0.452 | 0.446 | 0.437 | 0.433 | 0.429 | 0.430 | 0.445 | 0.441 |
| | 336 | 0.497 | 0.467 | 0.480 | 0.456 | 0.477 | 0.453 | 0.502 | 0.466 |
| | 720 | 0.542 | 0.504 | 0.494 | 0.485 | 0.491 | 0.483 | 0.497 | 0.481 |
| | Avg | 0.473 | 0.459 | 0.450 | 0.445 | 0.444 | 0.442 | 0.459 | 0.450 |
| ETTh2 | 96 | 0.309 | 0.359 | 0.312 | 0.358 | 0.297 | 0.349 | 0.309 | 0.354 |
| | 192 | 0.389 | 0.407 | 0.388 | 0.402 | 0.380 | 0.399 | 0.385 | 0.401 |
| | 336 | 0.422 | 0.434 | 0.428 | 0.436 | 0.430 | 0.437 | 0.424 | 0.431 |
| | 720 | 0.428 | 0.448 | 0.427 | 0.446 | 0.414 | 0.440 | 0.443 | 0.454 |
| | Avg | 0.387 | 0.412 | 0.389 | 0.411 | 0.380 | 0.307 | 0.390 | 0.410 |
| ETTm1 | 96 | 0.351 | 0.383 | 0.336 | 0.371 | 0.327 | 0.365 | 0.332 | 0.369 |
| | 192 | 0.385 | 0.400 | 0.378 | 0.394 | 0.368 | 0.387 | 0.376 | 0.394 |
| | 336 | 0.421 | 0.422 | 0.411 | 0.415 | 0.408 | 0.412 | 0.414 | 0.418 |
| | 720 | 0.489 | 0.458 | 0.481 | 0.453 | 0.477 | 0.450 | 0.487 | 0.458 |
| | Avg | 0.412 | 0.416 | 0.402 | 0.408 | 0.395 | 0.404 | 0.402 | 0.410 |
| ETTm2 | 96 | 0.188 | 0.273 | 0.183 | 0.269 | 0.185 | 0.266 | 0.182 | 0.266 |
| | 192 | 0.258 | 0.317 | 0.252 | 0.312 | 0.257 | 0.313 | 0.251 | 0.309 |
| | 336 | 0.322 | 0.356 | 0.318 | 0.354 | 0.313 | 0.349 | 0.313 | 0.349 |
| | 720 | 0.417 | 0.411 | 0.411 | 0.405 | 0.413 | 0.406 | 0.417 | 0.408 |
| | Avg | 0.296 | 0.339 | 0.291 | 0.335 | 0.292 | 0.334 | 0.291 | 0.333 |
| ECL | 96 | 0.180 | 0.226 | 0.171 | 0.218 | 0.168 | 0.213 | 0.171 | 0.215 |
| | 192 | 0.221 | 0.259 | 0.222 | 0.261 | 0.217 | 0.258 | 0.219 | 0.260 |
| | 336 | 0.279 | 0.301 | 0.276 | 0.300 | 0.275 | 0.299 | 0.277 | 0.303 |
| | 720 | 0.356 | 0.350 | 0.355 | 0.351 | 0.352 | 0.350 | 0.356 | 0.352 |
| | Avg | 0.259 | 0.284 | 0.256 | 0.283 | 0.253 | 0.280 | 0.256 | 0.283 |
| Weather | 96 | 0.145 | 0.243 | 0.142 | 0.237 | 0.142 | 0.238 | 0.144 | 0.242 |
| | 192 | 0.161 | 0.256 | 0.158 | 0.253 | 0.158 | 0.253 | 0.161 | 0.257 |
| | 336 | 0.178 | 0.274 | 0.174 | 0.270 | 0.174 | 0.269 | 0.176 | 0.274 |
| | 720 | 0.215 | 0.306 | 0.203 | 0.296 | 0.205 | 0.298 | 0.208 | 0.305 |
| | Avg | 0.175 | 0.270 | 0.169 | 0.264 | 0.170 | 0.265 | 0.172 | 0.270 |
| Traffic | 96 | 0.419 | 0.287 | 0.554 | 0.389 | 0.556 | 0.391 | 1.306 | 0.733 |
| | 192 | 0.434 | 0.292 | 0.519 | 0.362 | 0.529 | 0.369 | 1.411 | 0.803 |
| | 336 | 0.449 | 0.299 | 0.531 | 0.362 | 0.538 | 0.372 | 1.427 | 0.806 |
| | 720 | 0.484 | 0.318 | 0.628 | 0.419 | 0.740 | 0.476 | 1.616 | 0.865 |
| | Avg | 0.447 | 0.299 | 0.558 | 0.383 | 0.591 | 0.402 | 1.440 | 0.802 |
| Solar Energy | 96 | 0.213 | 0.247 | 0.239 | 0.272 | 0.235 | 0.274 | 0.353 | 0.375 |
| | 192 | 0.243 | 0.269 | 0.294 | 0.306 | 0.295 | 0.309 | 0.331 | 0.345 |
| | 336 | 0.264 | 0.284 | 0.306 | 0.317 | 0.308 | 0.318 | 0.332 | 0.343 |
| | 720 | 0.263 | 0.283 | 0.313 | 0.319 | 0.321 | 0.321 | 0.382 | 0.368 |
| | Avg | 0.246 | 0.271 | 0.288 | 0.304 | 0.290 | 0.306 | 0.350 | 0.358 |

Table 22: Performance of NTE (kernel size = 5, repeat = 1, with linear projection).

| Model | Metric | Conv1D | | Dilated Conv1D | | Bi-Dilated Conv1D | | Bi-Dilated Conv1D* | |
|---|---|---|---|---|---|---|---|---|---|
| | | MSE | MAE | MSE | MAE | MSE | MAE | MSE | MAE |
| ETTh1 | 96 | 0.392 | 0.412 | 0.390 | 0.407 | 0.389 | 0.407 | 0.396 | 0.413 |
| | 192 | 0.449 | 0.444 | 0.439 | 0.434 | 0.440 | 0.436 | 0.216 | 0.257 |
| | 336 | 0.510 | 0.473 | 0.480 | 0.456 | 0.483 | 0.458 | 0.273 | 0.298 |
| | 720 | 0.518 | 0.495 | 0.481 | 0.477 | 0.499 | 0.488 | 0.505 | 0.488 |
| | Avg | 0.465 | 0.455 | 0.455 | 0.449 | 0.452 | 0.447 | 0.466 | 0.456 |
| ETTh2 | 96 | 0.304 | 0.356 | 0.320 | 0.361 | 0.312 | 0.358 | 0.317 | 0.360 |
| | 192 | 0.383 | 0.403 | 0.393 | 0.405 | 0.384 | 0.400 | 0.388 | 0.402 |
| | 336 | 0.423 | 0.433 | 0.431 | 0.437 | 0.426 | 0.433 | 0.428 | 0.435 |
| | 720 | 0.427 | 0.447 | 0.427 | 0.445 | 0.420 | 0.442 | 0.417 | 0.438 |
| | Avg | 0.384 | 0.410 | 0.393 | 0.412 | 0.386 | 0.408 | 0.388 | 0.409 |
| ETTm1 | 96 | 0.347 | 0.381 | 0.336 | 0.371 | 0.334 | 0.371 | 0.338 | 0.372 |
| | 192 | 0.383 | 0.399 | 0.379 | 0.395 | 0.375 | 0.393 | 0.376 | 0.394 |
| | 336 | 0.417 | 0.419 | 0.412 | 0.416 | 0.412 | 0.416 | 0.412 | 0.416 |
| | 720 | 0.491 | 0.460 | 0.479 | 0.453 | 0.479 | 0.453 | 0.489 | 0.460 |
| | Avg | 0.410 | 0.415 | 0.402 | 0.409 | 0.400 | 0.408 | 0.404 | 0.411 |
| ETTm2 | 96 | 0.187 | 0.271 | 0.185 | 0.270 | 0.184 | 0.267 | 0.186 | 0.268 |
| | 192 | 0.254 | 0.315 | 0.251 | 0.312 | 0.250 | 0.311 | 0.251 | 0.309 |
| | 336 | 0.325 | 0.357 | 0.316 | 0.351 | 0.315 | 0.350 | 0.310 | 0.346 |
| | 720 | 0.416 | 0.410 | 0.410 | 0.403 | 0.418 | 0.409 | 0.407 | 0.402 |
| | Avg | 0.296 | 0.338 | 0.291 | 0.334 | 0.292 | 0.334 | 0.289 | 0.331 |
| ECL | 96 | 0.143 | 0.240 | 0.142 | 0.238 | 0.143 | 0.238 | 0.156 | 0.256 |
| | 192 | 0.161 | 0.256 | 0.159 | 0.254 | 0.158 | 0.252 | 0.163 | 0.258 |
| | 336 | 0.178 | 0.275 | 0.173 | 0.269 | 0.173 | 0.269 | 0.177 | 0.276 |
| | 720 | 0.242 | 0.329 | 0.209 | 0.302 | 0.207 | 0.301 | 0.260 | 0.347 |
| | Avg | 0.181 | 0.275 | 0.171 | 0.266 | 0.170 | 0.265 | 0.189 | 0.284 |
| Weather | 96 | 0.174 | 0.217 | 0.176 | 0.222 | 0.165 | 0.211 | 0.171 | 0.217 |
| | 192 | 0.218 | 0.257 | 0.218 | 0.258 | 0.216 | 0.257 | 0.218 | 0.258 |
| | 336 | 0.275 | 0.298 | 0.274 | 0.298 | 0.273 | 0.299 | 0.276 | 0.300 |
| | 720 | 0.354 | 0.350 | 0.355 | 0.351 | 0.355 | 0.350 | 0.360 | 0.355 |
| | Avg | 0.255 | 0.281 | 0.256 | 0.282 | 0.252 | 0.279 | 0.256 | 0.283 |
| Traffic | 96 | 0.422 | 0.290 | 1.322 | 0.741 | 1.272 | 0.717 | 1.411 | 0.803 |
| | 192 | 0.432 | 0.290 | 1.272 | 0.718 | 1.163 | 0.671 | 1.414 | 0.804 |
| | 336 | 0.449 | 0.299 | 0.857 | 0.527 | 0.843 | 0.519 | 1.428 | 0.806 |
| | 720 | 0.483 | 0.316 | 0.878 | 0.525 | 0.906 | 0.549 | 1.450 | 0.810 |
| | Avg | 0.447 | 0.299 | 1.082 | 0.628 | 1.046 | 0.614 | 1.426 | 0.806 |
| Solar Energy | 96 | 0.219 | 0.251 | 0.272 | 0.298 | 0.261 | 0.290 | 0.472 | 0.423 |
| | 192 | 0.254 | 0.274 | 0.305 | 0.313 | 0.306 | 0.313 | 0.720 | 0.566 |
| | 336 | 0.276 | 0.290 | 0.345 | 0.333 | 0.349 | 0.339 | 0.712 | 0.528 |
| | 720 | 0.277 | 0.292 | 0.344 | 0.339 | 0.343 | 0.331 | 0.926 | 0.568 |
| | Avg | 0.257 | 0.277 | 0.317 | 0.321 | 0.315 | 0.318 | 0.708 | 0.521 |

Table 23: Performance of NTE (kernel size = 5, repeat = 2, without linear projection).

| Model | Metric | Conv1D | | Dilated Conv1D | | Bi-Dilated Conv1D | | Bi-Dilated Conv1D* | |
|---|---|---|---|---|---|---|---|---|---|
| | | MSE | MAE | MSE | MAE | MSE | MAE | MSE | MAE |
| ETTh1 | 96 | 0.396 | 0.415 | 0.394 | 0.409 | 0.382 | 0.403 | 0.400 | 0.415 |
| | 192 | 0.450 | 0.446 | 0.443 | 0.438 | 0.449 | 0.442 | 0.450 | 0.446 |
| | 336 | 0.489 | 0.463 | 0.480 | 0.458 | 0.481 | 0.457 | 0.513 | 0.474 |
| | 720 | 0.523 | 0.495 | 0.502 | 0.489 | 0.496 | 0.486 | 0.502 | 0.489 |
| | Avg | 0.465 | 0.455 | 0.455 | 0.449 | 0.452 | 0.447 | 0.466 | 0.456 |
| ETTh2 | 96 | 0.308 | 0.358 | 0.321 | 0.363 | 0.304 | 0.353 | 0.314 | 0.358 |
| | 192 | 0.389 | 0.405 | 0.404 | 0.410 | 0.390 | 0.402 | 0.399 | 0.408 |
| | 336 | 0.425 | 0.433 | 0.435 | 0.439 | 0.431 | 0.439 | 0.445 | 0.445 |
| | 720 | 0.427 | 0.444 | 0.444 | 0.454 | 0.430 | 0.448 | 0.440 | 0.451 |
| | Avg | 0.387 | 0.410 | 0.401 | 0.417 | 0.389 | 0.411 | 0.400 | 0.416 |
| ETTm1 | 96 | 0.340 | 0.377 | 0.336 | 0.371 | 0.326 | 0.364 | 0.338 | 0.372 |
| | 192 | 0.379 | 0.398 | 0.380 | 0.396 | 0.371 | 0.388 | 0.381 | 0.398 |
| | 336 | 0.415 | 0.419 | 0.417 | 0.419 | 0.408 | 0.413 | 0.422 | 0.422 |
| | 720 | 0.485 | 0.457 | 0.491 | 0.460 | 0.486 | 0.454 | 0.492 | 0.463 |
| | Avg | 0.405 | 0.413 | 0.406 | 0.412 | 0.398 | 0.405 | 0.408 | 0.414 |
| ETTm2 | 96 | 0.188 | 0.271 | 0.185 | 0.269 | 0.187 | 0.268 | 0.182 | 0.265 |
| | 192 | 0.256 | 0.315 | 0.251 | 0.311 | 0.260 | 0.315 | 0.253 | 0.310 |
| | 336 | 0.316 | 0.352 | 0.321 | 0.354 | 0.324 | 0.356 | 0.315 | 0.351 |
| | 720 | 0.418 | 0.408 | 0.413 | 0.406 | 0.421 | 0.410 | 0.407 | 0.403 |
| | Avg | 0.295 | 0.337 | 0.293 | 0.335 | 0.298 | 0.337 | 0.289 | 0.332 |
| ECL | 96 | 0.143 | 0.242 | 0.454 | 0.485 | 0.142 | 0.238 | 0.454 | 0.485 |
| | 192 | 0.160 | 0.257 | 0.395 | 0.444 | 0.184 | 0.280 | 0.395 | 0.444 |
| | 336 | 0.176 | 0.275 | 0.397 | 0.446 | 0.234 | 0.329 | 0.396 | 0.445 |
| | 720 | 0.212 | 0.306 | 0.469 | 0.493 | 0.346 | 0.409 | 0.469 | 0.493 |
| | Avg | 0.173 | 0.270 | 0.429 | 0.467 | 0.227 | 0.314 | 0.429 | 0.467 |
| Weather | 96 | 0.168 | 0.214 | 0.170 | 0.216 | 0.167 | 0.211 | 0.174 | 0.222 |
| | 192 | 0.214 | 0.256 | 0.214 | 0.257 | 0.215 | 0.256 | 0.221 | 0.262 |
| | 336 | 0.277 | 0.302 | 0.276 | 0.300 | 0.272 | 0.298 | 0.282 | 0.305 |
| | 720 | 0.353 | 0.350 | 0.355 | 0.351 | 0.354 | 0.351 | 0.359 | 0.355 |
| | Avg | 0.253 | 0.281 | 0.254 | 0.281 | 0.252 | 0.279 | 0.259 | 0.286 |
| Traffic | 96 | 0.438 | 0.302 | 1.311 | 0.730 | 1.408 | 0.801 | 1.358 | 0.754 |
| | 192 | 0.447 | 0.301 | 1.220 | 0.691 | 1.152 | 0.648 | 1.298 | 0.733 |
| | 336 | 0.461 | 0.307 | 1.283 | 0.720 | 1.290 | 0.717 | 1.391 | 0.767 |
| | 720 | 0.498 | 0.325 | 1.027 | 0.583 | 1.307 | 0.721 | 1.354 | 0.737 |
| | Avg | 0.461 | 0.309 | 1.210 | 0.681 | 1.289 | 0.722 | 1.350 | 0.748 |
| Solar Energy | 96 | 0.217 | 0.251 | 0.315 | 0.340 | 0.304 | 0.333 | 0.639 | 0.529 |
| | 192 | 0.257 | 0.279 | 0.318 | 0.334 | 0.326 | 0.340 | 0.606 | 0.490 |
| | 336 | 0.276 | 0.292 | 0.381 | 0.363 | 0.367 | 0.357 | 0.549 | 0.467 |
| | 720 | 0.269 | 0.287 | 0.370 | 0.354 | 0.387 | 0.369 | 0.576 | 0.491 |
| | Avg | 0.255 | 0.277 | 0.346 | 0.348 | 0.346 | 0.350 | 0.593 | 0.494 |

Table 24: Performance of NTE (kernel size = 5, repeat = 2, with linear projection).

| Model | Metric | Conv1D | | Dilated Conv1D | | Bi-Dilated Conv1D | | Bi-Dilated Conv1D* | |
|---|---|---|---|---|---|---|---|---|---|
| | | MSE | MAE | MSE | MAE | MSE | MAE | MSE | MAE |
| ETTh1 | 96 | 0.393 | 0.414 | 0.390 | 0.407 | 0.390 | 0.409 | 0.399 | 0.415 |
| | 192 | 0.450 | 0.446 | 0.442 | 0.437 | 0.448 | 0.442 | 0.452 | 0.447 |
| | 336 | 0.487 | 0.463 | 0.485 | 0.461 | 0.486 | 0.462 | 0.492 | 0.469 |
| | 720 | 0.490 | 0.483 | 0.501 | 0.488 | 0.510 | 0.494 | 0.508 | 0.490 |
| | Avg | 0.455 | 0.452 | 0.455 | 0.448 | 0.459 | 0.452 | 0.463 | 0.455 |
| ETTh2 | 96 | 0.304 | 0.356 | 0.312 | 0.357 | 0.309 | 0.357 | 0.324 | 0.365 |
| | 192 | 0.385 | 0.404 | 0.391 | 0.405 | 0.393 | 0.406 | 0.415 | 0.416 |
| | 336 | 0.423 | 0.432 | 0.433 | 0.438 | 0.428 | 0.434 | 0.436 | 0.438 |
| | 720 | 0.426 | 0.445 | 0.430 | 0.447 | 0.428 | 0.445 | 0.437 | 0.448 |
| | Avg | 0.385 | 0.409 | 0.392 | 0.412 | 0.390 | 0.411 | 0.403 | 0.417 |
| ETTm1 | 96 | 0.337 | 0.375 | 0.337 | 0.372 | 0.337 | 0.372 | 0.339 | 0.373 |
| | 192 | 0.378 | 0.397 | 0.380 | 0.396 | 0.379 | 0.396 | 0.381 | 0.397 |
| | 336 | 0.414 | 0.419 | 0.415 | 0.418 | 0.416 | 0.418 | 0.423 | 0.423 |
| | 720 | 0.493 | 0.461 | 0.487 | 0.458 | 0.487 | 0.458 | - | - |
| | Avg | 0.406 | 0.413 | 0.405 | 0.411 | 0.405 | 0.411 | 0.381 | 0.398 |
| ETTm2 | 96 | 0.188 | 0.270 | 0.186 | 0.270 | 0.186 | 0.268 | 0.184 | 0.267 |
| | 192 | 0.257 | 0.315 | 0.253 | 0.312 | 0.250 | 0.310 | 0.252 | 0.310 |
| | 336 | 0.314 | 0.350 | 0.315 | 0.353 | 0.315 | 0.351 | 0.311 | 0.347 |
| | 720 | 0.415 | 0.407 | 0.413 | 0.406 | 0.415 | 0.407 | 0.416 | 0.406 |
| | Avg | 0.294 | 0.336 | 0.292 | 0.335 | 0.292 | 0.334 | 0.291 | 0.333 |
| ECL | 96 | 0.143 | 0.243 | 0.390 | 0.439 | 0.278 | 0.355 | 0.603 | 0.581 |
| | 192 | 0.161 | 0.259 | 0.323 | 0.390 | 0.549 | 0.560 | 0.495 | 0.512 |
| | 336 | 0.178 | 0.277 | 0.393 | 0.442 | 0.387 | 0.435 | 0.468 | 0.497 |
| | 720 | 0.244 | 0.331 | 0.457 | 0.487 | 0.471 | 0.498 | 0.493 | 0.512 |
| | Avg | 0.182 | 0.278 | 0.391 | 0.440 | 0.421 | 0.462 | 0.515 | 0.526 |
| Weather | 96 | 0.168 | 0.215 | 0.166 | 0.213 | 0.169 | 0.217 | 0.171 | 0.217 |
| | 192 | 0.216 | 0.257 | 0.216 | 0.257 | 0.221 | 0.262 | 0.220 | 0.260 |
| | 336 | 0.216 | 0.247 | 0.278 | 0.302 | 0.281 | 0.303 | 0.276 | 0.299 |
| | 720 | 0.275 | 0.301 | 0.359 | 0.353 | 0.358 | 0.352 | 0.360 | 0.353 |
| | Avg | 0.219 | 0.255 | 0.255 | 0.281 | 0.257 | 0.284 | 0.257 | 0.282 |
| Traffic | 96 | 1.321 | 0.749 | 1.289 | 0.724 | 1.273 | 0.719 | 1.273 | 0.715 |
| | 192 | 0.938 | 0.557 | 1.287 | 0.724 | 1.275 | 0.715 | 1.441 | 0.786 |
| | 336 | 0.982 | 0.566 | 1.277 | 0.719 | 1.260 | 0.713 | 1.428 | 0.806 |
| | 720 | 0.996 | 0.569 | 1.450 | 0.810 | 1.321 | 0.728 | 1.450 | 0.809 |
| | Avg | 1.059 | 0.610 | 1.326 | 0.744 | 1.282 | 0.719 | 1.398 | 0.779 |
| Solar Energy | 96 | 0.245 | 0.274 | 0.375 | 0.378 | 0.280 | 0.320 | 0.560 | 0.470 |
| | 192 | 0.279 | 0.296 | 0.372 | 0.391 | 0.481 | 0.428 | 0.631 | 0.512 |
| | 336 | 0.305 | 0.316 | 0.390 | 0.382 | 0.440 | 0.409 | 0.547 | 0.469 |
| | 720 | 0.305 | 0.311 | 0.377 | 0.358 | - | - | 0.451 | 0.399 |
| | Avg | 0.284 | 0.299 | 0.379 | 0.377 | 0.400 | 0.386 | 0.547 | 0.463 |

