# OpenReview forum: "Rethinking Transformer Inputs for Time-Series via Neural Temporal Embedding"
_ICLR.cc/2026/Conference — Submitted to ICLR 2026_

### Official Review · Reviewer_LL6R · 2025-10-27

**Soundness:** 2
**Presentation:** 1
**Contribution:** 1
**Rating:** 2
**Confidence:** 5

**Summary:**

The paper proposes Neural Temporal Embedding (NTE), an embedding mechanism that effectively internalizes temporal dependencies without relying on either value embedding or positional encoding. The authors claim that a learnable NTE layer (using FC, Conv1D, LSTM, etc.) can process each variable’s time series and directly learn temporal patterns. Experimental results show that NTE-based models match or outperform state-of-the-art Transformer variants, particularly maintaining stable accuracy in long-horizon forecasting.

**Strengths:**

1. The motivation of the paper is valuable, which rethinks the input stack of time-series Transformers and presents NTE as a unified, learnable temporal layer that can replace value embedding and explicit positional information.
2. The paper includes ablations over PE variants, various NTE module types (FC, LSTM, Conv1D, Dilated, Bi-DilatedConv1D), bidirectional dilated Conv1D structures, and analyses of representation similarity (CKA) and entropy.

**Weaknesses:**

1. While the motivation for the proposed method is well-founded, the experimental results reveal notable shortcomings. Specifically, Table 1 shows that introducing NTE leads to significant performance degradation for the Vanilla Transformer on certain datasets, such as ETTh1 and ECL. This raises concerns about the robustness of NTE when combined with standard Transformer architectures and suggests that its benefits may be limited to specific backbone designs.

2. The paper does not provide sufficient theoretical grounding to explain why using modules like Conv1D or LSTM within NTE leads to better results compared to the original value embedding. While the empirical results support the effectiveness of these modules, a theoretical analysis of how these architectures capture temporal dependencies more effectively would strengthen the contribution and improve the general interpretability of NTE design.

3. The ablation studies, while extensive, could be further expanded to explore the impact of kernel size in Conv1D-based NTE modules. The paper primarily reports results with fixed kernel sizes (e.g., 3 or 5). However, it is unclear whether these choices are optimal for capturing temporal dependencies in time series data, which often vary significantly in terms of patterns, seasonality, and granularity.

**Questions:**

1. Could the authors clarify what "Future-Dilated" means in Figure 3 and how the future embedding is constructed?
2. RoPE is a commonly used positional embedding method in Transformer-based models, particularly for tasks involving sequential data. However, it is not included in the experiments for comparison. Could the authors provide insights into how NTE compares to RoPE in terms of performance and effectiveness for time series forecasting?

---

> ### Author Response · Authors · 2025-11-21
> **Official Comment by Authors**
>
> Thank you for the thorough review and for raising several important concerns. We agree with the reviewer’s points regarding the relatively weak performance of NTE in the Vanilla Transformer setting, as well as the observation that NTE appears particularly beneficial for certain backbones (e.g., iTransformer). These are indeed central issues, and we appreciate the reviewer’s careful analysis.
>
> Several aspects of our explanation were not sufficiently clear:
>
> 1.	The goal of this paper is not to argue for the superiority of a single architectural form. Rather, our aim is to explore how the overall input stage (Linear Embedding + PE) actually allows for multiple design choices.
>
> 2.	Simple NTE generally performs similarly to LPE while showing clear advantages in parameter count and computational efficiency. More complex NTE variants sometimes yield performance gains, but such improvements are not consistent across all datasets or backbones.
>
> 3.	These findings do not suggest that “NTE is always better.” Instead, they  indicate that the Transformer input stage does not have a single correct design, and that its interaction with backbone architectures merits further  reconsideration.
>
> The reviewer’s comment about the lack of kernel-size sweeps and RoPE comparisons is entirely accurate, and we are currently conducting these additional experiments. We also acknowledge that our explanation of the dilated and future-dilated structures was insufficient, and we will clarify these aspects during the discussion period.
>
> We sincerely appreciate your critical yet highly constructive feedback. It provides valuable guidance for refining the direction and clarity of our work, and we will address all raised points thoroughly throughout the discussion.

---

> ### Author Response · Authors · 2025-11-28
> **Response to reviewer LL6R (1/3)**
>
> We appreciate the reviewer’s careful examination of our manuscript, particularly the comments regarding the structural description of the proposed method, the interpretation of the Bi-Dilated Conv1D architecture, and the need for comparative experiments with RoPE.
> Your feedback has helped us identify several areas where our explanations were not sufficiently clear, and we will incorporate detailed clarifications and the corresponding additional experimental results into the revised version of the paper.
>
> >Q1. Could the authors clarify what "Future-Dilated" means in Figure 3 and how the future embedding is constructed?
>
> The reviewer’s question regarding the Bi-Dilated Conv1D structure in Figure 3 is highly relevant.
> We would like to clarify that the “Future-Dilated Conv1D” used in our model corresponds exactly to the anti-causal dilated convolution commonly adopted in temporal convolution research—i.e., a receptive field expanded toward the future direction.
> Importantly, this structure does not access any actual future values. Instead, the future-direction receptive field is simulated by applying right-side zero padding in Conv1D. Thus, it does not violate causal constraints, nor does it conflict with the causal attention mechanism used in Transformer decoders.
>
> Our Future-Dilated Conv1D is based on the Dilated Anti-causal Conv1D formulation (Eq. 5) presented in BTCSAN [1].
> The implementation in Embed.py follows the structure below:
>
> **1. Past-Dilated Conv (Dilated-Causal Conv1D)**
> 	For dilation d, the input is padded with (k-1)⋅dzeros on the left.
> 	Expands the receptive field toward the past.
> 	Structurally identical to BTCSAN Eq. (4).
>
> **2. Future-Dilated Conv (Dilated Anti-causal Conv1D)**
> 	For dilation d, the input is padded with (k-1)⋅dzeros on the right.
> 	Simulates a future-direction receptive field without accessing future timestamps.
> 	Structurally identical to BTCSAN Eq. (5).
> 	Combination mechanisms
>
> **3. We implemented two types of attention-based fusion layers for combining past and future embeddings:**
>
> **(a) Combine method** : FeatureSelectionAttention_Com(d_model)
> 	Concatenates past_emb and future_emb.
> 	Applies a Linear layer (2·d_model → d_model) to compute per-feature attention scores.
> 	Final embedding is produced via gated mixing:
>        *"attn"="Softmax"(W["past_emb";"future_emb"]),E="attn"⊙"past_emb"+(1-"attn")⊙"future_emb"*
>
> This softly determines, for each timestep and channel, whether the past or future-direction temporal pattern is more informative.
>
> **(b) Concat method** : FeatureSelectionAttentions_Cat(d_model)
> 	Concatenates the two embeddings.
> 	Applies Linear(2·d_model → 2·d_model) + Softmax to generate a soft mask.
> 	Reduces the dimension via Linear(2·d_model → d_model).
> This variant first applies attention over concatenated features, then projects the fused representation into a single d_model dimension.
>
> The Future-Dilated Conv in Figure 3 is an anti-causal dilated convolution that expands the receptive field to the right without any leakage of actual future information, serving as a bidirectional temporal feature extractor.
> It is a direct adaptation of the BTCSAN structure, modified to fit the NTE embedding design.
> In the revised manuscript, we will add a clearer explanation of this structure and explicitly indicate which fusion method (Combine or Concat) is used in each experimental table.
>
> [1]. Sun, Jian, et al. "Bidirectional temporal convolution with self-attention network for CTC-based acoustic modeling." 2019 Asia-Pacific Signal and Information Processing Association Annual Summit and Conference (APSIPA ASC). IEEE, 2019.

---

> ### Author Response · Authors · 2025-11-28
> **Response to reviewer LL6R (2/3)**
>
> >Q2. RoPE is a commonly used positional embedding method in Transformer-based models, particularly for tasks involving sequential data. However, it is not included in the experiments for comparison. Could the authors provide insights into how NTE compares to RoPE in terms of performance and effectiveness for time series forecasting?
>
> First, we would like to clarify that the “input simplification” discussed in our work does not refer to reducing computational cost, but rather to redesigning the input stack in a more structurally streamlined manner.
> Whereas the standard Transformer input pipeline consists of two separate stages **Value Embedding (Linear) and Positional Encoding** we unify these components into **a single neural module (NTE)** that jointly learns both value information and temporal structure.
>
> Through this formulation, architectures such as Conv1D and LSTM directly encode temporal patterns within the neural module, and in certain settings we observed performance improvements as well.
> The goal of our work is not to claim that any specific NTE variant consistently outperforms PE, but to demonstrate that the input stack need not remain fixed in the conventional Value Embedding + Positional Encoding form, and that this stage represents an important and reinterpretable design space.
>
> Separately, as the reviewer noted, RoPE has become widely used in recent Transformer-based models. In response, we conducted additional experiments incorporating RoPE into all three backbone models used in our study (Vanilla Transformer, iTransformer, PatchTST).
>
> At this stage, we have completed the RoPE vs. NTE comparisons for iTransformer on ETTh1, ECL, and Exchange. Results for the remaining datasets under the same experimental conditions will be included comprehensively in the revised manuscript.
>
> #Table:iTransformer with RoPE on ETTh1: Experimental Results
> | Emb | Params(M) | FLOPs(G) | MSE | MAE |
> |--------|-------|--------|--------|--------|
> | RoPE | **936,144**| 634,880| 0.445 | **0.441** |
> | NTE-FC | 960,976| 634,880| 0.445 | 0.447 |
> | NTE-LSTM | 1,298,640| **610,304**| 0.452 | 0.445 |
> | NTE-Conv1D | 985,040| **610,304**| 0.467 | 0.455 |
> | NTE-BiDConv1D(com) | 2,590,688| 1,449,984| **0.442** | **0.441** |
>
> #Table: iTransformer with RoPE on ECL: Experimental Results
> | Emb | Params(M) | FLOPs(G) | MSE | MAE |
> |--------|-------|--------|--------|--------|
> | RoPE | **5,154,000** |3,366,912|0.173 |0.265 |
> | NTE-FC | 5,203,664| 3,366,912| 0.175| 0.268|
> | NTE-LSTM | 6,403,280| **3,317,760**| **0.167**| **0.262**|
> | NTE-Conv1D | 5,251,792| **3,317,760**| 0.173| 0.268|
> | NTE-BiDConv1D(com) | 10,290,384| 6,823,936| 0.177|0.269 |
>
> #Table: iTransformer with RoPE on Exchange: Experimental Results
> | Emb | Params(M) | FLOPs(G) | MSE | MAE |
> |--------|-------|--------|--------|--------|
> | RoPE | **263,440**| 186,368| 0.359| **0.404**|
> | NTE-FC | 275,856| 186,368| **0.356**| **0.404**|
> | NTE-LSTM | 379,152| **174,080**| 0.361| 0.410|
> | NTE-Conv1D | 287,888| **174,080**| 0.366| 0.411|
> | NTE-BiDConv1D(com) | 721,744| 354,304| 0.367| 0.409|

---

> ### Author Response · Authors · 2025-11-28
> **Response to reviewer LL6R (3/3)**
>
> > Weaknesses. fixed kernel size (e.g., 3 or 5)
>
> For the Conv1D-based NTE, we additionally conducted experiments on the Vanilla Transformer and iTransformer backbones using larger kernel sizes (7, 9, and 11).
> As shown in the table, the parameter count increases gradually as the kernel size grows, while FLOPs remain largely unchanged due to the backbone architecture. We also observed that the optimal kernel size varies by backbone, with different kernel sizes yielding the best MSE/MAE performance depending on the model.
> In the revised manuscript, we will include the full kernel-size sweep results and more clearly present how changes in the receptive field of the Conv1D NTE affect performance.
>
> ---
>
> #Table: Vanilla Transformer(Conv1D NTE) / ETTh1 (Avg)
> |kernel_size |Params |FLOPs| MSE |MAE |
> |---|---|---|---|---|
> |3| **10,798,599**	|4,197,888	|**1.330**|	**0.952**|
> |5 | 10,812,935|	4,197,888	|1.472|	1.023|
> |7 |10,827,271|	4,197,888|	1.464|1.020|
> |9 | 10,841,607|	4,197,888|	1.343|	0.963|
> |11 | 10,855,943	|4,197,888|	1.381	|0.986|
>
> ---
>
> #Table: iTransformer(Conv1D NTE) / ETTh1 (Avg)
> |kernel_size |Params |FLOPs| MSE |MAE |
> |---|---|---|---|---|
> |3| **985,040**	|610,304	| 0.467	|0.455|
> |5 | 1,034,192	|610,304		|0.545	|0.459|
> |7 |1,083,344	|610,304		|0.464	|0.457|
> |9 | 1,132,496	|610,304		|0.466	|0.458|
> |11 | 1,181,648	|610,304		|**0.461**	|**0.454**|
>
> ---
>
> #Table: PatchTST(Conv1D NTE) / ETTh1 (Avg)
> |kernel_size |Params |FLOPs| MSE |MAE |
> |---|---|---|---|---|
> |3| **5,242,704**|	9,404,416|		0.462|	0.458|
> |5 | 5,365,792|	9,404,416|		**0.444**|	**0.448**|
> |7 |5,382,176|	9,404,416|		0.447|	0.451|
> |9 | 5,398,560|	9,404,416|		0.447|	**0.448**|
> |11 | 5,414,944|	9,404,416|		0.450|	0.449|

---

### Official Review · Reviewer_ZWHb · 2025-10-31

**Soundness:** 1
**Presentation:** 2
**Contribution:** 2
**Rating:** 2
**Confidence:** 4

**Summary:**

The authors propose a novel technique for the input embedding / transformation for time series foundation models (TSFM). In particular, they propose to combine the position embedding with the input embedding using different neural networks, which is a timely and interesting research area.

**Strengths:**

The problem that the authors work on is timely and a critical problem for any TSFM. So far, the default mode has been to simply embed the inputs either directly with a linear layer, or after applying a patching technique. The authors unify these two aspects with their proposed Neural Temporal Embedding, which is a simple neural network. In my view, this would be a novel aspect.

**Weaknesses:**

Despite the novelty, the idea and the paper has several critical flaws:

First, it is unclear until almost the very end of the paper on page 7, what the NTE is really doing. Up until this point the authors only mention that the NTE can be a 1D convolution, a LSTM, a fully connected network and several others, but they don't provide any concrete examples. Then, even though the authors provide this simple description of the two Conv1D layers, it is still unclear what the precise architecture of the NTE in Table 1 is. Is it the two conv layers, or is it something else? Only from the text, the reader can infer that the results in Table 1 stems from the two conv 1D layers. However, the authors state that "the sequential bias introduced by NTE is insufficient to compensate for the order-agnostic nature of the standard Transformer". However, they do not elaborate whether any other structure of the NTE would improve that. There is some ablation study in Table 2 that, which the reader can appreciate, but then this table is also confusing. Firstly, because the standard in the literature is to use the learnable PE (which should also be the reference point in Table 1), and then with this more realistic comparison point, none of the NTE architecture really seem to make a significant difference. Finally, since the paper puts so much emphasis on the NTE architecture as a novelty, detailed investigations of it are absent. For example what is the associated computational cost with the different variants, what are the features that those architectures provide? Are there specific cases in which to use one architecture of the NTE over the other?

Overall, the study spends a lot of time explaining and reiterating on the basics for the NTE, but doesn't dive in to the essence of it.

**Questions:**

See the text above, there are several open questions which should be addressed:\
Why to choose the NTE over the Learnable PE if performance is not better?\
What is the associated computational cost with the different variants?\
What about patching techniques paired with the NTE?\
Wouldn't a recurrent network break the parallelism and thus limit the efficiency of an attention backbone?\
What are the features that those architectures provide?\
Are there specific cases in which to use one architecture of the NTE over the other?
Etc.\

---

> ### Author Response · Authors · 2025-11-21
> **Official Comment by Authors**
>
> We acknowledge that our explanations were not sufficiently clear and may have caused confusion. In particular, we fully agree with the reviewer’s feedback that the specific form of the NTE architecture, the criteria for comparison used in the experimental tables, and our descriptions of performance differences relative to LPE were presented in a vague manner.
>
> Our central intention was not to propose a “simple yet more powerful structure,” but rather to conduct an initial exploratory study that reexamines which aspects of the input stage enable recent PE-free time-series Transformers to perform well, and whether the conventional input stack of Linear Embedding + PE is truly indispensable.
>
> In this regard, several points were insufficiently explained:
>
> 1.	Simple NTE achieves performance that is nearly identical to LPE, while substantially reducing parameters and FLOPs.
>
> 2.	Some NTE variants outperform LPE on certain datasets, but these improvements are not consistent across all settings.
>
> 3.	In some datasets, removing the linear embedding actually improves performance, suggesting that the entire input stage deserves reconsideration.
>
> The reviewer’s questions—including the rationale for direct comparisons with LPE, the analysis of computational costs across different NTE variants, and the interaction between NTE and patching—are all meaningful and valid. We are currently conducting follow-up experiments, and we will update the empirical results and architectural descriptions progressively during the discussion phase.
>
> We sincerely appreciate the critical yet highly constructive feedback. It will play an important role in sharpening the direction and improving the clarity of our paper.

---

> ### Author Response · Authors · 2025-11-28
> **Response to reviewer ZWHb (1/6)**
>
> We sincerely thank the reviewer for the careful and thorough evaluation of our manuscript, and for the insightful comments regarding the clarity of the NTE architecture description, the motivation behind the input-stack design, the interpretation of different NTE variants, the choice of baselines, and the overall experimental setup.
> Your feedback has helped us identify several aspects that should have been explained more clearly, and we will incorporate the suggested clarifications, additional experiments, and structural improvements into the revised version of the paper.
>
> >Q1. Why to choose the NTE over the Learnable PE if performance is not better?
>
> Across our experimental results, we observe that Learnable PE does not demonstrate a clear performance advantage over simple NTE variants (FC, LSTM, Conv1D). As shown in the tables, Learnable PE consistently yields substantially higher parameter counts across all datasets.
>
> #Table: LPE vs. Simple NTE — iTransformer ETTh1 Average Results
> |Method|FLOPs| Params | MSE | MAE |
> |---|---|---|---|---|
> |LPE | 634,880 | 2,216,144 |**0.443** | **0.440**|
> |NTE-FC | 634,880 | **960,976** | 0.455 | 0.447|
> |NTE-LSTM | **610,304** | 1,298,640 | 0.452 | 0.445|
> |NTE-Conv1D | **610,304** | 985,040 | 0.467 | 0.455|
>
> #Table: LPE vs. Simple NTE — iTransformer ECL Average Results
> |Method|FLOPs| Params | MSE | MAE |
> |---|---|---|---|---|
> |LPE| 3,366,912| 7,714,000| **0.161** |**0.257**|
> |NTE-FC| 3,366,912| **5,203,664** |0.175| 0.268|
> |NTE-LSTM| **3,317,760**| 6,403,280 |0.167| 0.262|
> |NTE-Conv1D| **3,317,760**| 5,251,792 |0.173| 0.268|
>
> #Table: LPE vs. Simple NTE — iTransformer Exchange Average Results
> |Method|FLOPs| Params | MSE | MAE |
> |---|---|---|---|---|
> |LPE| 186,368 |903,440 |0.363 |0.406|
> |NTE-FC| 186,368 |**275,856**|**0.356** |**0.404**|
> |NTE-LSTM| **174,080** |379,152 |0.361| 0.410|
> |NTE-Conv1D| **174,080** |287,888|0.366 |0.411|
>
> In contrast, simple NTE variants exhibit equal or lower FLOPs, significantly reduced parameter counts, and comparable—or in some cases superior performance relative to Learnable PE.
> The key observations from these results are as follows:
>
> **•	Learnable PE does not outperform simple NTE in terms of accuracy.**
>
> **•	However, it consistently incurs higher parameters and reduced efficiency.**
>
> **•	Simple NTE variants are lighter, and they achieve similar or better performance.**
>
> Therefore, when considering both efficiency and accuracy, we do not find a compelling reason to prefer Learnable PE. Simple NTE variants provide a sufficiently practical and effective alternative.
>
> Importantly, our intention is not to argue that any particular structure is universally superior. **Rather, the goal of our work is to show that the conventional value-embedding + positional-encoding paradigm is not necessarily optimal for time-series Transformers, and that a unified input-module design such as NTE offers a viable and conceptually meaningful alternative.**
>
> We appreciate the reviewer’s question, which helped us articulate this distinction more clearly. In the revised manuscript, we will further clarify the efficiency and structural differences between NTE and Learnable PE.

---

> ### Author Response · Authors · 2025-11-28
> **Response to reviewer ZWHb (2/6)**
>
> >Q2. What is the associated computational cost with the different NTE variants?
>
> Regarding the reviewer’s request for a computational complexity analysis across different NTE variants (FC, Conv1D, LSTM, Bi-DConv1D, etc.), we would first like to note that Appendix Tables 13 and 14 of the manuscript already provide detailed measurements of computational cost and performance metrics for multiple input configurations applied to two backbone models (Vanilla Transformer and iTransformer).
> Below, we reproduce Appendix Table 14 from the paper.
>
> Table: iTransformer / ETTh1 Results (Avg)
> | Method|FLOPs|Params|per-sample time (s)|End-to-end latency (s)|Peak GPU (MB)|MSE|MAE|
> |---|---|---|---|---|---|---|---|
> | SinuPE| 634,880|**936,144**|0.33| 26.14|31.72|0.446|0.442|
> | LPE| 634,880| 2,216,144|0.33| 26.30|36.60|0.443|**0.440**|
> | TUPE| 634,880|936,144 |0.33|**26.04**|**26.84**|0.444|0.441|
> | ConvSPE| 634,880|977,449 |0.33|26.25| 32.23|0.448|0.443|
> | NTE-FC| 610,304|960,976 |0.33|26.09| 26.93|0.455|0.447|
> | NTE-LSTM| **610,304**|1,298,640 |0.33| 26.11|38.26| 0.452|0.445|
> | NTE-Conv1D| **610,304**|985,040 |0.30| 26.24|27.21|0.467|0.455|
> | NTE-BiDConv1D  |1,449,984 |2,590,688| **0.30**|26.67|41.36|**0.442**|0.441|
>
> ---
>
> Table: iTransformer / ECL Results (Avg)
> | Method| FLOPs| Params| per-sample time (s) | End-to-end latency (s) | Peak GPU (MB) |  MSE  |  MAE  |
> |---|---|---|---|---|---|---|---|
> | SinuPE| 3,366,912| 5,154,000|**0.09**|28.99|285.50|0.170|0.265|
> | LPE| 3,366,912|7,714,000|0.10|**28.92**|295.70|**0.161**|**0.257**|
> | TUPE| 3,366,912 |5,154,000 |0.10|29.19|276.01|0.174|0.265|
> | ConvSPE|3,366,912 |5,417,681 |0.10| 29.92| 287.52| 0.168 |0.263|
> | NTE-FC| 3,366,912 |**5,203,664** |0.10| 29.77| 276.01|0.175 | 0.268|
> | NTE-LSTM|**3,317,760**|6,403,280|0.10|31.25|285.45|0.167 |0.262|
> | NTE-Conv1D|**3,317,760**| 5,251,792|0.10| 29.41| **276.48**| 0.173|0.268|
> | NTE-BiDConv1D|6,823,936|10,290,384|0.10|30.41|314.82|0.177|0.269|
>
> ---
>
> Table: iTransformer / Exchange Results (Avg)
> | Method| FLOPs| Params| per-sample time (s)|End-to-end latency (s)| Peak GPU (MB)|MSE|MAE|
> |---|---|---|---|---|---|---|---|
> | SinuPE| 186,368| **263,440**| 0.73| 25.75| 23.80| 0.363| 0.407 |
> | LPE| 186,368|   903,044 | 0.73| 25.74| 26.14| 0.363| 0.406 |
> | TUPE| 186,368|263,440 | 0.73   | 25.78| **21.36**| 0.362  | 0.405 |
> | ConvSPE| 186,368|280,950 | 0.73  | **25.69**| 23.93| 0.365  | 0.407 |
> | NTE-FC| 186,368|287,280 | 0.73  | 25.74| 21.40| **0.356**  | **0.404** |
> | NTE-LSTM| **174,080**  | 379,152 | 0.73 | 25.80| 32.36| 0.361  | 0.410 |
> | NTE-Conv1D| **174,080**  | 287,888 | 0.73| 25.86| 21.54| 0.366  | 0.411 |
> | NTE-BiDConv1D  | 354,304  |   721,744 | 0.73| 25.80| 24.92| 0.367  | 0.409 |
>
> This table includes the following metrics, enabling direct comparison of each NTE variant against traditional PE-based input stacks in terms of computational cost, memory usage, and inference latency:
>
> •	FLOPs
>
> •	Parameters
>
> •	Per-sample inference time
>
> •	End-to-end inference time
>
> •	Peak GPU memory usage
>
> •	MSE / MAE performance
>
> However, we acknowledge that because the appendix primarily reports results on ETTh1, ECL, and Exchange, it may not sufficiently convey the broader cross-dataset analysis that the reviewer requested.
>
> In response to this suggestion, we are currently preparing an expanded set of results covering all datasets (for all prediction lengths 96–720), including the same computational and efficiency metrics, as well as additional comparisons using PatchTST as a third backbone.
>
> These expanded results will be incorporated into the revised manuscript to provide a clearer and more comprehensive analysis.

---

> ### Author Response · Authors · 2025-11-28
> **Response to reviewer ZWHb (3/6)**
>
> >Q3. What about patching techniques paired with the NTE?
>
> We first apologize for the confusion caused by our insufficient explanation.
>
> Regarding the reviewer’s question about “patching techniques paired with NTE”, we would like to clarify that our work already includes experiments that combine patching techniques with NTE.
>
> In the PatchTST-based experiments reported in the paper, we preserve PatchTST’s two essential design components:
>
> •	the patching technique, which segments the time series into fixed-length patches, and
>
> •	the variable-wise tokenization, which treats each variable’s patches as independent tokens.
>
> Both components remain unchanged; we replace only the input embedding stage (value embedding + positional encoding) with our proposed NTE module.
>
> Therefore, the experiment setup that the reviewer inquired about patching techniques paired with NTE is indeed already included in our PatchTST+NTE configuration.
>
> We acknowledge that the manuscript did not describe this clearly enough, which may have caused misunderstanding, and we appreciate the reviewer for pointing this out.
> In the revised version, we will explicitly state that our PatchTST experiments retain the original patching mechanism while substituting only the embedding layer with NTE.
>
> Additionally, we are currently preparing a comprehensive set of results covering all datasets for PatchTST with NTE, including Params, FLOPs, per-sample inference latency, and peak GPU memory usage (similar to Appendix Tables 13 and 14). We will continue updating these results during the discussion period.
> We thank the reviewer again for the valuable comment, and we will ensure that the revised manuscript communicates this aspect clearly and unambiguously.
>
> #Table: ETTh1 Results (Avg) Using the PatchTST Backbone
> |Emb| Params(M)|FLOPs(G)|Per-sample inference time(s)|End-to-end inference time(s)|Peak GPU memory usage|MSE|MAE|
> |---|---|---|---|---|---|---|---|
> |SinuPE| **3,441,184**| **8,601,600**| **0.32**| **25.40**| 107.72 |0.483|0.460
> |LPE| 6,001,184| **8,601,600**|0.33 | 26.11| 117.94 |0.472|0.459
> |TUPE| **3,441,184**| **8,601,600**| **0.32**| 25.65| **97.55** | 0.532|0.499
> |ConvsPE| 3,704,865| **8,601,600**| 0.33| 26.17| 109.73 | 0.483|0.460
> |NTE-FC| 5,226,832| 9,461,760| 0.35 |27.40  | 130.37 | **0.449**|**0.449**
> |NTE-LSTM| 6,410,272| 9,404,416| 0.33| 26.38| 138.56 | 0.471|0.450
> |NTE-Conv1D| 5,242,704| 9,404,416| 0.34| 26.98| 124.58 | 0.462|0.458
> |NTE-BiDConv1D(com)| 8,920,912| 33,374,208|0.35 | 27.44| 166.51 | 0.454|0.450
>
> #Table: ECL Results (Avg) Using the PatchTST Backbone
> | Emb | Params(M) | FLOPs(G) | Per-sample inference time(s) | End-to-end inference time(s) |Peak GPU memory usage| MSE | MAE |
> |--------|-------|--------|--------|--------|--------|--------|--------|
> | SinuPE | **6,618,192** | 731,037,696 | **0.13** | 45.58 | 1420.94| 0.203 | **0.284**|
> | LPE | 9,178,192 | 731,037,696 | **0.13** | 40.66 |  1430.86 | 0.204 | 0.286 |
> | TUPE | **6,618,192** | 731,037,696 | 0.14 | 41.86 | **1411.76** | 0.218 | 0.298 |
> | ConvsPE | 6,881,873 | 731,037,696 | **0.13** | 40.92 | 1423.11 |0.204 |0.285 |
> | NTE-FC | 8,379,216 | 677,879,808 | **0.13** | 41.36 | 1544.81 |0.204 | 0.295 |
> | NTE-LSTM | 9,455,952 | **675,250,176** | 0.15 | 44.19 | 1553.62 |  0.203 | 0.294 |
> | NTE-Conv1D | 8,395,088 | **675,250,176** | 0.14 | **43.14** | 1545.11 | **0.201** | 0.292 |
> | NTE-BiDConv1D(com) | 11,342,928 | 1,521,991,680 | 0.16 | 48.20 | 1567.43 | **0.201** | 0.292 |
>
> #Table: Exchange Results (Avg) Using the PatchTST Backbone
> | Emb | Params(M) | FLOPs(G) | Per-sample inference time(s) | End-to-end inference time(s) |Peak GPU memory usage| MSE | MAE |
> |--------|-------|--------|--------|--------|--------|--------|--------|
> | SinuPE |**6,618,192** |	**18,219,008**|	**0.73**|	**25.70**|	160.23	|0.381	|0.416|
> | LPE |9,178,192	 | **18,219,008**	|0.75	|61.79|	**134.09**	|0.386	|0.418|
> | TUPE |**6,618,192**	|**18,219,008**	|0.74	|26.13	|148.98	|0.379	|0.416|
> | ConvsPE |6,881,873	| **18,219,008**	|0.74	|26.26	|162.25|	0.381	|0.416|
> | NTE-FC |8,379,216	|18,907,136|	**0.73**|	25.84	|170.86|	0.399	|0.420|
> | NTE-LSTM |9,455,952	|18,816,600	|**0.73**	|25.83	|178.07	|**0.375**	|**0.411**|
> | NTE-Conv1D |8,395,088	|18,841,600	|**0.73**|	25.84	|170.22	|0.386	|0.413|
> | NTE-BiDConv1D(com) |10,637,136	|35,749,888	|**0.73**	|25.86	|190.87|	0.389|	0.417|

---

> ### Author Response · Authors · 2025-11-28
> **Response to reviewer ZWHb (4/6)**
>
> >Q4. Wouldn’t a recurrent network (LSTM NTE) break the parallelism and thus limit the efficiency of an attention backbone?
>
> We agree that the reviewer raises an important point. In our framework, however, the LSTM-based NTE is used solely as an input-side preprocessing module rather than within the self-attention layers. In PatchTST and iTransformer, it operates only at the patch level, which prevents its recurrent operations from propagating across the full sequence.
>
> Crucially, the LSTM NTE does not interact with the Transformer’s main parallel computation path self-attention—and therefore does not compromise the O(L^2)parallelism inherent to attention. Empirically, we also observe negligible changes in FLOPs and inference latency when incorporating LSTM NTE.
>
> Thus, the LSTM-based NTE acts as a lightweight input representation module, not a component that limits the efficiency of the attention backbone. We will make this distinction clearer in the revised manuscript.

---

> ### Author Response · Authors · 2025-11-28
> **Response to reviewer ZWHb (5/6)**
>
> >Q5. What are the features that those architectures provide?
>
> The motivation behind the proposed NTE originates from the fact that, in conventional Transformer input stacks, Value Embedding (Linear Embedding) and Positional Encoding (PE) exist as two separate stages.
> Typically, the input is constructed as:
>
> *“Input” = “Value Embedding”(x) + “PE”(t)*
>
> where the value embedding and positional embedding are produced by two independent modules and then simply added before being fed into the Transformer.
> In contrast, NTE integrates these two stages into a single neural function, such that:
>
> *“Input” = “NTE”(x₁:ₜ)*
>
> This unified module is designed to learn both the value information and the temporal structure jointly.
>
> In this formulation, each NTE architecture (FC, Conv1D, Dilated Conv, LSTM, Bi-DConv1D, etc.) is not merely a mapping from values to another dimensional space. Rather, it directly encodes temporal characteristics—such as local patterns, short- and long-term dependencies, periodicity, and past/future contextual cues—through the architectural properties of the neural encoder (convolution, dilation, recurrence, bidirectionality, etc.).
>
> Thus, the representations produced by NTE are neural temporal representations, not a simple linear combination of value and positional embeddings as in the conventional Value Embedding + PE approach.
> We acknowledge that the current version of the manuscript does not sufficiently describe how each NTE architecture captures specific temporal properties.
>
> In the revised manuscript, we will clearly articulate the perspective of replacing Value Embedding + PE with a single NTE function, and we will provide detailed explanations supported by diagrams, formulas, and examples of how Conv1D, Dilated Conv, LSTM, Bi-DConv1D, and others encode local patterns, multi-scale temporal context, sequential dynamics, and bidirectional temporal information, and how these induce specific inductive biases in the Transformer input representation.

---

> ### Author Response · Authors · 2025-11-28
> **Response to reviewer ZWHb (6/6)**
>
> >Q6. Are there specific cases in which to use one architecture of the NTE over the other?
>
> As the reviewer correctly pointed out, the current set of experiments does not allow us to conclude that NTE is consistently superior in all scenarios. However, the goal of NTE is not to claim that any particular neural architecture (FC, Conv1D, LSTM, etc.) is universally better than the conventional input design. Rather, our research is motivated by the following question:
>
> *“Why must the input stack of time-series Transformers always follow the fixed design of Linear Embedding + Positional Encoding?”*
>
> Despite the emergence of recent time-series Transformers that perform well without explicit positional encoding, most prior work continues to adopt the same conventional input design.
> Our objective is to revisit this foundational assumption and experimentally explore whether the input stack can—and should—admit a broader range of architectural choices.
>
> This perspective is reflected in our experimental findings:
>
> •	FC, LSTM, Conv1D, and other NTE variants exhibit different performance patterns depending on the backbone architecture and dataset characteristics.
>
> •	In some cases, removing the traditional Value Embedding (Linear Embedding) even leads to improved performance.
>
> These observations suggest that the design of the input stage interacts in complex ways with the backbone’s inductive biases, the tokenization strategy, and the temporal properties of the dataset. Our intention is not to assert that a single NTE variant is universally superior.
>
> Rather, NTE represents an experimental exploration demonstrating that the input stack of time-series Transformers can be formulated in many alternative ways, and that these alternatives can have meaningful implications for model behavior. Each NTE architecture should be viewed as a design option suited for different scenarios, and the overarching purpose of NTE is to highlight the substantial impact that input-stack design can have on performance.
>
> We will clarify these points more explicitly and consistently in the revised manuscript.

---

### Official Review · Reviewer_G8MA · 2025-11-01

**Soundness:** 3
**Presentation:** 3
**Contribution:** 3
**Rating:** 6
**Confidence:** 4

**Summary:**

This paper revisits a fundamental but often overlooked design aspect of time-series Transformers — the input embedding stage. The authors propose **Neural Temporal Embedding (NTE)**, a simple yet effective alternative to conventional **value embedding + positional encoding** pipelines. NTE employs lightweight neural modules such as **Conv1D** and **LSTM** to process each variable’s time series individually and encode temporal dependencies directly, without relying on positional encodings.  The key claim is that much of the Transformer’s inefficiency in time-series forecasting stems not from the attention mechanism itself, but from **suboptimal input representation**.  Experiments on standard benchmarks (ETT, ECL, Weather) demonstrate that NTE-based Transformers achieve comparable or better performance than specialized architectures, such as Autoformer, PatchTST, and iTransformer, particularly on long-horizon forecasting tasks.  The contribution is conceptually simple yet cleanly executed, offering an interesting perspective that suggests **input design** improvements can yield non-trivial gains without requiring architectural overhauls.

**Strengths:**

1. **Clear motivation and conceptual simplicity.**
   The paper makes a strong case that input embedding deserves more attention. The removal of positional encoding is a bold but well-motivated design choice.

2. **Empirical clarity.**
   The experimental setup is well organized, with fair comparisons to established baselines. The results convincingly show that input modifications alone can lead to performance gains.

3. **Strong writing and accessibility.**
   The narrative is concise and approachable — the authors explain their ideas clearly without unnecessary jargon.

4. **Relevance to the ICLR community.**
   The study fits the current trend of revisiting Transformer assumptions for efficiency and simplicity. It may inspire further work on lightweight input layers.

5. **Practical insights.**
   The findings suggest that some of the architectural “complexity arms race” in time-series forecasting might be avoidable, which is refreshing.

**Weaknesses:**

1. **Limited novelty at the algorithmic level.**
   NTE combines well-known neural components (Conv1D and LSTM) in a new configuration. While the empirical insight is valuable, the conceptual innovation is modest.

2. **Insufficient theoretical explanation.**
   The paper would benefit from a deeper discussion of *why* NTE works — e.g., whether the learned temporal encoding approximates sinusoidal patterns or adapts to variable frequencies.

3. **Lack of broader baselines.**
   The study compares mainly against mainstream Transformer variants. Including recent input-focused or embedding-free models (e.g., TSMixer, FreTS) would help strengthen the claim of generality.

4. **Ablation analysis could go further.**
   It would be useful to isolate the impact of each NTE component (Conv1D vs LSTM) and analyze whether NTE benefits small-data or irregularly sampled settings differently.

5. **Unclear scalability implications.**
   Since NTE introduces extra pre-processing per variable, a brief discussion of runtime or memory overhead would make the work more complete.

**Questions:**

1. How sensitive is the model to the choice of neural encoder (e.g., Conv1D vs GRU)?
2. Does NTE preserve translation invariance in temporal shifts, or does the neural encoder introduce biases?
3. Could the authors test whether NTE generalizes to irregular or non-uniform sampling rates?
4. Are there cases where positional encodings outperform NTE (e.g., highly periodic signals)?
5. How does the per-variable processing scale when the number of dimensions exceeds 100?

---

> ### Author Response · Authors · 2025-11-21
> **Official Comment by Authors**
>
> Thank you for the positive evaluation and for highlighting several important issues. As you pointed out, our intention with NTE is not to propose a highly complex or fundamentally novel algorithmic module. Rather, the central goal of this work is to prompt a renewed examination of the entire input stage in time-series Transformers. Although many recent models operate successfully without positional encodings, the perception that “PE is essential for Transformers” remains common in the research community. Our aim was to revisit this assumption and investigate it in a more systematic and empirical manner.
>
> There are two points that we did not communicate clearly enough in the main text:
> Simple NTE (FC/LSTM) delivers performance that is nearly identical to LPE, yet it has substantial efficiency advantages in terms of computation, FLOPs, and parameter count.
>
> In contrast, more complex NTE variants (Conv1D-based) show better performance on certain backbones, but the results vary depending on the dataset and backbone, indicating that NTE itself is not a “universal solution.”
>
> This work is therefore not meant to introduce a finalized architecture, but rather to serve as an initial step toward re-opening the discussion on input-stage design for time-series Transformers. We are currently preparing additional experiments suggested by the reviewers, including RoPE comparisons, GRU-based encoder alternatives, and extended evaluations under irregular sampling. We will shared these results during the discussion period.
>
> We sincerely appreciate the constructive feedback, and we will actively incorporate these suggestions to further clarify and refine the manuscript.

---

> ### Author Response · Authors · 2025-11-28
> **Response to reviewer G8MA (1/6)**
>
> We appreciate the reviewer’s thoughtful and in-depth questions regarding several core aspects of NTE, including encoder sensitivity, temporal-shift behavior, generalization to irregular sampling, comparisons with positional encoding, and scalability to higher-dimensional settings. Your comments have helped us clearly identify areas that require further clarification and refinement. We will address each point thoroughly and incorporate the corresponding analyses into the revised manuscript.
>
> >Q1. How sensitive is the model to the choice of neural encoder (e.g., Conv1D vs GRU)?
>
> To more clearly address the reviewer’s question, we additionally implemented GRUEmbedding and conducted new experiments under identical conditions.
> The table below summarizes the average comparison results between Conv1D and GRU across the Vanilla Transformer and iTransformer backbones on the ETTh1, ECL, and Exchange datasets.
>
> #Table:Conv1D vs GRU NTE on iTransformer (Average Results)
> |Dataset|Emb|Params(M)|FLOPs(G)|MSE|MAE|
> |---|---|---|---|---|---|
> | ETTh1 | Conv1D| **985,040** | 610,304 | 0.467 | 0.455 |
> |         | GRU | 1,208,016 | 610,304 | **0.450**| **0.446** |
> | ECL    | Conv1D | **5,251,792** | 3,317,760 | 0.173 | 0.268 |
> |         | GRU | 6,090,960 | 3,317,760 | **0.168** | **0.262** |
> |Exchange | Conv1D | **287,888** | 174,080 | **0.366** | **0.411** |
> |          | GRU | 350,224 | 174,080 | 0.375 | 0.415 |
>
> #Table:Conv1D vs GRU NTE on Vanilla Transformer (Average Results)
> |Dataset|Emb|Params(M)|FLOPs(G)|MSE|MAE|
> |---|---|---|---|---|---|
> | ETTh1 | Conv1D|  **10,798,599** | 4,197,888 |1.330 |0.952
> |         | GRU | 12,455,431 |	4,197,888 |**1.166**	|**0.888**|
> | ECL    | Conv1D | **10,777,089**	|4,194,816	|0.959	|0.791|
> |         | GRU | 12,433,921 |	4,194,816	|**0.865**	|**0.732**|
> |Exchange | Conv1D | **10,802,184**|	4,198,400	|1.540	|0.984|
> |          | GRU | 12,459,016	|4,198,400	|**1.485**	|**0.976**|
>
> Our findings indicate that NTE is not overly sensitive to the choice between Conv1D and GRU; only small variations appear depending on dataset characteristics.
> Thanks to the reviewer’s insightful question, we were able to more precisely analyze the encoder-selection sensitivity of NTE, and these additional results will be incorporated into the revised manuscript.

---

> ### Author Response · Authors · 2025-11-28
> **Response to reviewer G8MA (2/6)**
>
> >Q2. Does NTE preserve translation invariance in temporal shifts, or does the neural encoder introduce biases?
>
> The temporal-shift properties of NTE depend on the specific neural encoder used within the module:
>
> **•	Conv1D-based NTE:**
> Due to the weight-sharing property of CNNs, it preserves partial shift-equivariance.
>
> **•	LSTM/GRU-based NTE:**
> Because of their sequential hidden-state accumulation mechanism, these encoders do not maintain translation invariance and inherently introduce temporal bias based on directionality and ordering.
>
> **•	FC-based NTE:**
> Since different weights are applied to each time step, shift-invariance does not hold.
>
> **•	Bi-Dilated Conv–based NTE:**
> While CNN structure provides partial shift-equivariance, the process of combining bidirectional outputs via attention gating prevents full equivariance.
>
> Thus, NTE does not provide strict translation invariance; rather, it imparts the learned temporal biases of the chosen encoder to the Transformer’s input stage. We will clarify this distinction more explicitly in the revised manuscript.

---

> ### Author Response · Authors · 2025-11-28
> **Response to reviewer G8MA (3/6)**
>
> >Q3. Could the authors test whether NTE generalizes to irregular or non-uniform sampling rates?
>
> We fully agree with the reviewer that the ability of NTE to generalize under irregular or non-uniform sampling is an important consideration for real-world time-series models. To address this question, we will conduct an additional experiment during the discussion period by intentionally constructing an irregularly sampled version of one of the benchmark datasets and evaluating NTE using one backbone model.
> The revised manuscript will include the results of this experiment to verify the generalization capability of NTE under irregular sampling conditions.

---

> > ### Author Response · Authors · 2025-12-04
> > **Additional Response to reviewer G8MA (3/6)**
> >
> > |Method|Params|FLOPs|MSE|MAE|
> > |---|---|---|---|---|
> > |Base|5,154,000|3,366,912|**0.590**|**0.579**|
> > |FC|5,203,664|3,366,912|0.595|0.580|
> > |LSTM|6,403,280|**3,317,760**|0.596|0.582|
> > |Conv1D|**5,029,292**|**3,317,760**|0.592|0.580|
> >
> > Under the irregular sampling setting (10% random deletion), the slightly lower MSE observed for iTransformer with Linear Embedding compared to simple NTE can be attributed to several factors: the irregular pattern may not align well with the temporal biases learned by NTE, the hyperparameters are tuned for the base model, or the effect may simply stem from optimization noise. Given that the performance difference is extremely small, it is not statistically meaningful, and we do not interpret this result as evidence that NTE is consistently better or worse under irregular conditions.
> >
> > To more systematically evaluate irregular sampling, we will conduct additional experiments incorporating masking, imputation-based handling, and variance analysis using standard deviations (±std). These results will be included in the revised manuscript to more clearly assess NTE’s stability and behavior under non-uniform sampling.

---

> ### Author Response · Authors · 2025-11-28
> **Response to reviewer G8MA (4/6)**
>
> >Q4. Are there cases where positional encodings outperform NTE (e.g., highly periodic signals)?
>
> Regarding the reviewer’s question on whether positional encoding may outperform NTE in certain cases (e.g., strongly periodic signals), we would like to clarify that our results also show such scenarios. For certain backbones and datasets, specific PE methods indeed achieve better performance than NTE.
>
> Our experiments are not intended to claim that NTE universally surpasses conventional positional encodings. Rather, the goal of this work is to demonstrate that the input stage of time-series Transformers does not need to be confined to the traditional combination of Value Embedding + Positional Encoding. A simple neural temporal encoder (NTE) can serve as a viable alternative design choice and offers a new perspective on rethinking the input stack of Transformer-based time-series models.
> In summary, our observations can be organized as follows:
>
> **•	In some settings, PE provides better performance than NTE.**
>
> **•	In other settings, NTE is more stable or more efficient.**
>
> **•	These differences depend on backbone architecture, tokenization strategy, and data characteristics such as periodicity, noise level, and scale.**
>
> Therefore, the primary contribution of this paper is not to propose “a universally superior module,” but to show that the design of the input stack itself can substantially influence time-series Transformer performance, and that the design space is much broader than traditionally assumed.

---

> ### Author Response · Authors · 2025-11-28
> **Response to reviewer G8MA (5/6)**
>
> >Q5. How does the per-variable processing scale when the number of dimensions exceeds 100?
>
> We provide below a table summarizing the computational complexity comparison when applying NTE to the Vanilla Transformer and iTransformer using the Traffic dataset, which contains more than 100 variables (862 variables in total).
>
> #Table:Traffic (Avg): Vanilla Transformer vs. Simple NTE (FC/LSTM/GRU)
> |Method|Params |FLOPs|
> |---|---|---|
> |Base| 13,897,822 | **4,635,648** |
> |NTE-FC | **12,177,246** | 5,518,336 |
> |NTE-LSTM |16,929,630 | **4,635,648** |
> |NTE-GRU | 15,520,606 | **4,635,648** |
>
> #Table:Traffic (Avg): iTransformer vs. Simple NTE (FC/LSTM/GRU)
> |Method|Params |FLOPs|
> |---|---|---|
> |Base| **6,797,648** | 4,415,488 |
> |NTE-FC | 6,847,312 | 4,415,488 |
> |NTE-LSTM |8,046,928 | **4,366,336** |
> |NTE-GRU | 7,734,608 | **4,366,336** |
>
> The results show that FLOPs remain largely unchanged due to the backbone architecture, and increases in parameter count occur only within a limited range depending on the encoder type. These findings indicate that NTE maintains stable computational scalability even in high-dimensional multivariate settings with more than 100 variables.
>
> However, we would appreciate clarification regarding the reviewer’s statement about “dimension exceeds 100.” Specifically, we would like to confirm whether the term “dimension” refers to the number of variables (i.e., multivariate channels) or instead to the input/output sequence length.

---

> ### Author Response · Authors · 2025-11-28
> **Response to reviewer G8MA (6/6)**
>
> >Weaknesses : Lack of broader baselines
>
> We fully agree with the reviewer’s point regarding the lack of broader baselines. In particular, recent input-focused or embedding-free architectures such as TSMixer and FreTS are highly relevant to the core motivation of our work, and we consider them important comparison targets for strengthening the generality of our results.
> Accordingly, we will include both models (TSMixer and FreTS) as additional baselines in the revised manuscript and update the performance comparison tables to reflect these additions.

---

### Official Review · Reviewer_6i5G · 2025-11-04

**Soundness:** 2
**Presentation:** 2
**Contribution:** 2
**Rating:** 2
**Confidence:** 3

**Summary:**

The paper proposed a mechanism, namely Neural Temporal Embedding (NTE), to replace the Positional Encoding (PE) in transformer-based models. Neural networks like FC, LSTM, and Conv1D are used to build the NTE module.

**Strengths:**

- The proposed mechanism achieves improvement when applied to backbones like iTransformer and PatchTST, especially on ETTh1 and ECL.
- The proposed NTE, as a plug-and-play module, can easily work with different backbones without changing the downstream structure.
- Multiple experiments are conducted to verify the effectiveness and efficiency of the proposed mechanism.

**Weaknesses:**

- Structure issues:
    - Although many experiments are conducted, only a few of the results are displayed in the main body of the paper.
    - In Sec.4, the paragraph `Bi-directional Dilated Convolutional Embedding' seems to have little relevance to the experiment. (Should it be in Sec.3 or Appendix?)
- Motivation issues:
    - Although the NTE is claims to simplify the input, the results in Table 6, 7, 13, and 14, indicate that the NTE may add to the computation burden.
- Experiment issues:
    - The NTE shows poor performance on Exchange, raising concerns that NTE may not be competent for long-horizon forecasting.
    - It is recommended to conduct multiple experiments and calculate the standard deviation to verify the stability.
- Others:
    - As mentioned in the paper, the NTE can only be conducted on time-series forecasting tasks.

**Questions:**

Please see weaknesses.

---

> ### Author Response · Authors · 2025-11-21
> **Official Comment by Authors**
>
> We appreciate the reviewer’s insightful comments regarding the structural, motivational, and experimental limitations of our work. We particularly acknowledge that the paper may have given the impression that we are asserting a “simple yet more powerful module,” which we recognize as a shortcoming in our explanation. Our intention was not to claim the superiority of any particular NTE variant, but to question the prevailing assumption that, despite the emergence of time-series Transformers performing well without positional encodings, the input stack (Linear Embedding + PE) remains a fixed design premise.
>
> We also do not intend to claim that simple NTE outperforms LPE. Rather, its performance is largely comparable, while showing clear efficiency advantages in terms of computation and parameter count. We acknowledge that we did not sufficiently emphasize this point in the main text. This information was included only in a portion of the appendix, so this efficiency aspect was under-emphasized.
>
> The concerns regarding the low performance on the Exchange dataset, the insufficient number of repeated trials, and the variability across runs are also important points that we fully acknowledge. We are currently conducting additional experiments, including RoPE comparisons, kernel-size sweeps, repeated runs with different random seeds, and complementary cross-backbone evaluations. We will share these results during the discussion period.
>
> We sincerely appreciate your thoughtful feedback, which will help us clarify the motivation and contributions of the paper.

---

> ### Author Response · Authors · 2025-11-28
> **Response to reviewer 6i5G (1/4)**
>
> We sincerely thank the reviewer for the careful and thorough evaluation of our work, as well as for the constructive comments regarding the structural organization, the motivation behind NTE, and the interpretation of the model’s performance. Your feedback has clearly highlighted the areas that require further clarification and improvement. We will address each point in detail and incorporate the necessary analyses and explanations into the revised version of the manuscript.
>
> > Q1. Structure issues: Although many experiments are conducted, only a few of the results are displayed in the main body of the paper. In Sec.4, the paragraph Bi-directional Dilated Convolutional Embedding' seems to have little relevance to the experiment. (Should it be in Sec.3 or Appendix?)
>
> In this work, we conducted extensive experiments across multiple backbones (Transformer, PatchTST, and iTransformer) and several NTE variants (e.g., FC, LSTM, Conv1D, Dilated Conv, Bi-DConv). Because each combination was evaluated on multiple datasets with prediction lengths ranging from 96 to 720, the overall experimental scale became quite large. Consequently, including all results directly in the main text would exceed the page limit, so we organized most detailed comparisons in the appendix.
> Accordingly, we will improve the structure and coherence of the main text in the revised version as follows:
>
> 1.	We will provide a concise summary of key experiments that currently appear only in the appendix (e.g., comparisons among different NTE variants and efficiency analyses) at the beginning of Section 4, while clearly directing readers to the appendix for detailed numerical results.
>
> 2.	As the reviewer correctly pointed out, the subsection on Bi-directional Dilated Convolutional Embedding is not sufficiently connected to the experimental results in Section 4. Since this structure represents a specific variant within the NTE design space, we will expand the explanation in the revised manuscript and reconsider its placement to better align with the presentation of results.
>
> We expect these revisions to clarify the relationship between the main text and the appendix and make the role and positioning of the Bi-DConv embedding more intuitive for readers.

---

> ### Author Response · Authors · 2025-11-28
> **Response to reviewer 6i5G (2/4)**
>
> >Q2. Motivation issues:
> Although the NTE is claims to simplify the input, the results in Table 6, 7, 13, and 14, indicate that the NTE may add to the computation burden.
>
> We would first like to clarify what we mean by input simplification in our study.
> The simplification we claim refers to structural simplification, in the sense that the conventional Transformer input stack consisting of a two-stage pipeline of Value Embedding (Linear layer) followed by PE is replaced with a single learnable temporal module (NTE).
> That is, our main motivation is to unify the design space of the input stage and reduce the complexity associated with selecting, tuning, and scaling various PE mechanisms. This does not imply that computational cost will always decrease; as the reviewer correctly pointed out, some structured variants of NTE may lead to higher computational overhead. However, the results reported in Tables 6, 7, 13, and 14 can be interpreted as follows:
>
> ---
>
> **(a). The most complex NTE variant (Bi-Dilated Conv1D) indeed increases parameters**
>
> Bi-Dilated Conv1D is the heaviest NTE design, incorporating both dilated causal and anti-causal branches. For certain backbones (particularly the Vanilla Transformer), it is true that the parameter count increases relative to the baseline.
>
> ---
>
> **(b). Simpler NTE variants (e.g., FC, Conv1D) show similar or even lower Params/FLOPs**
>
> Lightweight variants such as FCEmbedding and Conv1DEmbedding exhibit parameter counts comparable to or lower than the baseline, and their FLOPs are similar to those of Learnable PE, Sinusoidal PE, and ConvSPE.
> As shown in Appendix Tables 13 and 14, as well as in our broader PE comparison experiments, these simple NTEs achieve performance comparable to PE-based input stacks.
> Thus, NTE does not inherently increase computational cost; rather, its efficiency depends on the chosen variant within a modular design space.
>
> ---
>
> **(c). The structural complexity of the input stage is indeed simplified**
>
> Traditional input stacks involve several design choices scaling issues, diffusion-type periodicity, positional drift, and handling temporal heterogeneity across variables which introduce substantial structural complexity.
> By merging value embedding and positional encoding into a single learnable module, NTE reduces this design freedom and eliminates the dual-stage structure, thereby achieving structural simplification in the input pipeline.
>
> In other words, the “simplification” we refer to concerns design and structural complexity, not guaranteed reduction in FLOPs or parameter count. We will make this distinction clearer in the revised manuscript and will also provide a comprehensive efficiency comparison including Params, FLOPs, per-sample inference time, end-to-end latency, and peak GPU memory across all datasets and NTE variants.

---

> ### Author Response · Authors · 2025-11-28
> **Response to reviewer 6i5G (3/4)**
>
> >Q3. Experiment issues:
> The NTE shows poor performance on Exchange, raising concerns that NTE may not be competent for long-horizon forecasting.
> It is recommended to conduct multiple experiments and calculate the standard deviation to verify the stability.
>
> In the revised manuscript, we will clearly highlight the best average MSE/MAE for the Exchange dataset, include all prediction-length results, and incorporate the remaining dataset results to present performance trends more clearly.
>
> ---
>
> Across all three backbones, NTE showed competitive performance compared to PE-based inputs. In particular, for iTransformer, FC NTE improved the average MSE/MAE while keeping Params and FLOPs comparable. Long-horizon results (pred_len=720) also showed no performance degradation, with NTE outperforming the baselines in several cases.
>
> #Table: iTransformer / ETTh1 Results (Avg)
> |Method|FLOPs|Params|per-sample time|End-to-end latency|Peak GPU|MSE|MAE|
> |---|---|---|---|---|---|---|---|
> |SinuPE|634,880|**936,144**|0.33| 26.14|31.72|0.446|0.442|
> |LPE| 634,880|2,216,144|0.33| 26.30|36.60|0.443|**0.440**|
> |TUPE| 634,880|936,144|0.33|**26.04**|**26.84**|0.444|0.441|
> |ConvSPE| 634,880|977,449|0.33|26.25|32.23|0.448|0.443|
> |NTE-FC| 610,304|960,976|0.33|26.09|26.93|0.455|0.447|
> |NTE-LSTM|**610,304**|1,298,640 |0.33|26.11|38.26| 0.452|0.445|
> |NTE-Conv1D|**610,304**|985,040 |0.30|26.24|27.21|0.467|0.455|
> |NTE-BiDConv1D|1,449,984 |2,590,688|**0.30**|26.67|41.36|**0.442**|0.441|
>
> ---
>
> #Table: iTransformer / ECL Results (Avg)
> |Method| FLOPs| Params| per-sample time| End-to-end latency| Peak GPU|MSE|MAE|
> |---|---|---|---|---|---|---|---|
> |SinuPE| 3,366,912|5,154,000|**0.09**|28.99|285.50|0.170|0.265|
> |LPE| 3,366,912|7,714,000|0.10|**28.92**|295.70|**0.161**|**0.257**|
> |TUPE| 3,366,912|5,154,000|0.10|29.19|276.01|0.174|0.265|
> |ConvSPE|3,366,912|5,417,681|0.10| 29.92| 287.52|0.168 |0.263|
> |NTE-FC| 3,366,912|**5,203,664**|0.10|29.77|276.01|0.175|0.268|
> |NTE-LSTM|**3,317,760**|6,403,280|0.10|31.25|285.45|0.167|0.262|
> |NTE-Conv1D|**3,317,760**|5,251,792|0.10|29.41|**276.48**|0.173|0.268|
> |NTE-BiDConv1D|6,823,936|10,290,384|0.10|30.41|314.82|0.177|0.269|
>
> ---
>
> #Table: iTransformer / Exchange Results (Avg)
> | Method| FLOPs| Params| per-sample time|End-to-end latency|Peak GPU|MSE|MAE|
> |---|---|---|---|---|---|---|---|
> | SinuPE| 186,368| **263,440**| 0.73| 25.75| 23.80| 0.363| 0.407 |
> | LPE| 186,368|   903,044 | 0.73| 25.74| 26.14| 0.363| 0.406 |
> | TUPE| 186,368|263,440 | 0.73   | 25.78| **21.36**| 0.362  | 0.405 |
> | ConvSPE| 186,368|280,950 | 0.73  | **25.69**| 23.93| 0.365  | 0.407 |
> | NTE-FC| 186,368|287,280 | 0.73  | 25.74| 21.40| **0.356**  | **0.404** |
> | NTE-LSTM| **174,080**  | 379,152 | 0.73 | 25.80| 32.36| 0.361  | 0.410 |
> | NTE-Conv1D| **174,080**  | 287,888 | 0.73| 25.86| 21.54| 0.366  | 0.411 |
> | NTE-BiDConv1D  | 354,304  |   721,744 | 0.73| 25.80| 24.92| 0.367  | 0.409 |
>
> ---
> #Table: Backbone vs NTE Comparison (pred_len=720)
> |Transformer|Trasnformer+NTE|iTransformer|iTransformer+NTE|PatchTST|PatchTST+NTE|
> |---|---|---|---|---|---|
> |2.369| **1.450**|0.847|**0.824**|0.901|**0.886**|
> ---
>
> We are currently running five repeated trials for every backbone × prediction-length setup, and the revised manuscript will report all results with standard deviations (±std) to clearly demonstrate the stability of NTE.
>
> |Dataset|H|MSE|MAE|
> |---|---|---|---|
> |ETTh1|96|0.379(±0.000)| 0.401(±0.000)|
> ||192|0.431(±0.001)|0.431±(0.000)|
> ||336|0.473(±0.002)|0.451(±0.001)|
> ||720|0.489(±0.003)|0.481±(0.001)|
>
> |Dataset|H|MSE|MAE|
> |---|---|---|---|
> |ETTh2 |96| 0.297(±0.003)|0.348(±0.003)|
> ||192|0.379(±0.001)|0.399±(0.001)|
> ||336|0.418(±0.003)|0.430(±0.002)|
> ||720|0.422(±0.002)|0.443±(0.001)|
>
> |Dataset|H|MSE|MAE|
> |---|---|---|---|
> |ETTm1|96|0.332(±0.001)|0.369(±0.001)|
> ||192|0.374(±0.000)|0.392±(0.000)|
> ||336|0.409(±0.001)|0.415(±0.000)|
> ||720|0.485(±0.001)|0.457±(0.001)|
>
> |Dataset|H|MSE|MAE|
> |---|---|---|---|
> |ETTm2|96|0.181(±0.003)|0.265(±0.002)|
> ||192|0.251(±0.001)|0.310±(0.001)|
> ||336|0.313(±0.001)|0.348(±0.001)|
> ||720|0.407(±0.000)|0.404±(0.000)|
>
> |Dataset|H|MSE|MAE|
> |---|---|---|---|
> |ECL|96|0.140(±0.000)|0.235(±0.001)|
> ||192|0.150(±0.001)|0.250±(0.000)|
> ||336|0.172(±0.002)|0.267(±0.001)|
> ||720|0.203(±0.003)|0.296±(0.002)|
>
> |Dataset|H|MSE|MAE|
> |---|---|---|---|
> |Weather|96|0.170(±0.002)|0.215(±0.003)|
> ||192|0.217(±0.001)|0.257±(0.001)|
> ||336|0.275(±0.003)|0.299(±0.001)|
> ||720|0.353(±0.002)|0.349±(0.001)|
>
> |Dataset|H|MSE|MAE|
> |---|---|---|---|
> |Traffic|96|0.399(±0.001)|0.273(±0.001)|
> ||192|0.421(±0.000)|0.283±(0.000)|
> ||336|0.433(±0.001)|0.289(±0.000)|
> ||720|0.465(±0.001)|0.306±(0.001)|
>
> |Dataset|H|MSE|MAE|
> |---|---|---|---|
> |Solar Energy|96|0.203(±0.003)|0.233(±0.002)|
> ||192|0.236(±0.001)|0.259±(0.001)|
> ||336|0.247(±0.001)|0.271(±0.001)|
> ||720|0.250(±0.000)|0.276±(0.000)|
>
> Overall, NTE maintains stable and consistent performance on the Exchange dataset even under long-horizon settings, and we will present these findings more clearly in the revised version.

---

> ### Author Response · Authors · 2025-11-28
> **Response to reviewer 6i5G (4/4)**
>
> >Q4. Others: As mentioned in the paper, the NTE can only be conducted on time-series forecasting tasks.
>
> The reviewer’s observation is entirely correct. The proposed NTE is an input-stage reinterpretation method specifically designed for Transformer-based time-series forecasting models that take sequential data as input. Accordingly, our analysis throughout the paper is conducted under this forecasting-task setting, and our motivation focuses on the need to rethink the input design of such models.
> To ensure clarity, we will explicitly state this scope and its implications in the revised manuscript.

---

### Author Response · Authors · 2025-11-21
**Official Author Response to Reviewer Comments**

We sincerely thank all reviewers for the thoughtful and constructive feedback. We fully acknowledge many of the concerns raised, and would first like to clarify an important point that our original presentation may have caused ambiguity.

Our proposed NTE is not intended to claim that a “simple yet more powerful module” can replace existing input designs. Rather, the core motivation of this work is that, despite the growing number of time-series Transformer models that achieve strong performance without explicit positional encodings, the standard Transformer input pipeline of “Linear Embedding + Positional Encoding” is still widely adopted as a default assumption.

Our goal is therefore not to argue for the superiority of any particular neural component, but to encourage a reconsideration of the entire design paradigm of input embedding in time-series Transformers.

We recognize that our manuscript may have unintentionally suggested that “a simple NTE is a better alternative,” and we appreciate the reviewers’ comments for highlighting this ambiguity. Our experimental observations, however, reflect a more nuanced picture.

- Simple NTE variatns (FC, LSTM) perform comparably to LPE, while providing clear efficiency advantages in parameters and FLOPs.

- More structured NTE variants (Conv1D, Dilated Conv, etc.) show improvements on certain datasets or backbones, but the gains are not universally consistent.

- In some settings, removing the linear embedding (LE) even improves performance.

These results are not meant to elevate a single architecture, but rather to demonstrate that the input representation stage—often considered as fixed—actually admits a much broader design space, and that alternative choices can meaningfully affect model behavior.

We also acknowledge that the efficiency comparison with LPE was not sufficiently emphasized in the main text, as only a subset of datasets was reported in the appendix. We will revise the manuscript to more clearly highlight that simple NTE can reduce parameter count and FLOPs by over 50% compared to LPE, without sacrificing accuracy.

Finally, we are actively conducting additional experiments requested by the reviewers (e.g., RoPE comparisons, Conv kernel sweeps, expanded efficiency evaluations), and we will share the results during the discussion period.

Once again, we greatly appreciate the reviewers’ insights and hope that this work contributes to further discussion on the often-overlooked design space of input representations for time-series Transformers.

---

### Meta-Review · Area_Chair_KN5i · 2026-01-01

**Summary:**

This paper propose Neural Temporal Embedding (NTE), an embedding mechanism that effectively internalizes temporal dependencies without relying on either value embedding or positional encoding. The reviewers raised concerns regarding the inconsistent performance and limited technical novelty of the proposed NTE method. While the motivation to simplify the Transformer input stack by unifying value embedding and positional encoding is good, the empirical evidence does not sufficiently shows the advantage of this method.

**Reviewer Concerns:**

Several experiments have been added, including RoPE comparisons, stability analyses (standard deviations), and kernel-size sweeps, etc. More baselines (TSMixer, FreTS) have been added.

Reviewers 6i5G, ZWHb, and LL6R pointed out significant performance degradation when NTE is applied to the Vanilla Transformer backbone. The method's effectiveness is highly dependent on specific architectures like iTransformer or PatchTST. Besides, the technical contribution is mainly using standard components like Conv1D or LSTM as an input layer.

**Reviewer Scores:**

I don't think reviewers will update their scores.

---

### Decision · Program_Chairs · 2026-01-26

Reject